# Polyphenols of the Mediterranean Diet and Their Metabolites in the Prevention of Colorectal Cancer

**DOI:** 10.3390/molecules26123483

**Published:** 2021-06-08

**Authors:** Aline Yammine, Amira Namsi, Dominique Vervandier-Fasseur, John J. Mackrill, Gérard Lizard, Norbert Latruffe

**Affiliations:** 1Team Bio-PeroxIL, “Biochemistry of the Peroxisome, Inflammation and Lipid Metabolism” (EA7270), University of Bourgogne Franche-Comté, Inserm, 21000 Dijon, France; alineyammine5@gmail.com (A.Y.); amira.namsi@gmail.com (A.N.); gerard.lizard@u-bourgogne.fr (G.L.); 2Team OCS, Institute of Molecular Chemistry of University of Burgundy (ICMUB UMR CNRS 6302), University of Bourgogne Franche-Comté, 21000 Dijon, France; dominique.vervandier-fasseur@u-bourgogne.fr; 3Department of Physiology, University College Cork, BioScience Institute, College Road, T12 YT20 Cork, Ireland; J.Mackrill@ucc.ie

**Keywords:** intestinal cancer, dietary polyphenols, Mediterranean diet, resveratrol, curcumin, quercetin, rutin, apigenin, EGCG, microbiota, polyphenol nanoformulation

## Abstract

The Mediterranean diet is a central element of a healthy lifestyle, where polyphenols play a key role due to their anti-oxidant properties, and for some of them, as nutripharmacological compounds capable of preventing a number of diseases, including cancer. Due to the high prevalence of intestinal cancer (ranking second in causing morbidity and mortality), this review is focused on the beneficial effects of selected dietary phytophenols, largely present in Mediterranean cooking: apigenin, curcumin, epigallocatechin gallate, quercetin-rutine, and resveratrol. The role of the Mediterranean diet in the prevention of colorectal cancer and future perspectives are discussed in terms of food polyphenol content, the effectiveness, the plasma level, and the importance of other factors, such as the polyphenol metabolites and the influence of the microbiome. Perspectives are discussed in terms of microbiome-dependency of the brain-second brain axis. The emergence of polyphenol formulations may strengthen the efficiency of the Mediterranean diet in the prevention of cancer.

## 1. Introduction

The Mediterranean diet is associated with the Mediterranean climate, which is temperate and characterized by hot and dry summers, and mild and humid winters. However, interestingly, the Mediterranean climate and diet are not restricted to Mediterranean Sea area but are also found in similar regions of the world located between 30 and 40° of latitude. For instance, the western coasts of continents like America (California, mid-Chili), the Cape region in South Africa, and Australia (south and west). In all of these countries, food and beverages are quite comparable and provide good health to the populations as illustrated by the exceptional longevity of people living on Sardinia and Crete, and those of the Okinawa population. The Mediterranean diet has beneficial effects on health thanks to foods rich in polyphenols; the most abundant phytochemicals in the plant kingdom present in fruit, vegetables, wine, and honey (Table 1) [1].

The Mediterranean diet has age-cultural and nutritional aspects that contribute to wellbeing. In 2010, the UNESCO world organization added the Mediterranean diet to the list of intangible cultural heritage, where cooking is associated with the consumption of beverages (wine and/or infusions/tea) with respect to the beliefs and traditions of each community. Mediterranean cooking is becoming more and more popular, providing food microcomponents including polyphenols, vitamins, fiber, vegetables, olive oil, fishes rich in polyunsaturated fatty (ω3), and oligo-elements supplied by fruits and beverages. Besides fruits and vegetables, red wine provides additional unique polyphenols with strong anti-oxidant properties. These include resveratrol, procyanidins, and monophenols, including hydroxytyrosol and tyrosol. The Mediterranean diet is represented by a food pyramid proposed by Willett W.C. et al. [3] based on food groups, consumed weekly or less frequently.

There are about 8000 different dietary polyphenols, divided into two major classes: (i) Flavonoids which include other subclasses based on their chemical structure such as (flavanols, isoflavones, flavanones, flavonones, and anthocyanidins) and (ii) non-flavonoids divided into different groups (phenolic acids, stilbenes, and lignans; tannins are flavonoid polymers) [4]. The Table 2 illustrates the polyphenol profiles of some fruits and vegetables of the Mediterranean diet that have important anti-oxidant potential and a high polyphenol content [2].

We can observe that quercetin, rutin, and glucoside derivatives are from the major polyphenols present in grapes, strawberries, raspberries, beans, tomatoes, and celery. There is considerable evidence from in vitro studies, animal models, and clinical studies supporting the concept that several polyphenols have powerful anti-oxidant [5] and anti-tumor properties (cell cycle delay, apoptosis induction, metastasis prevention) [6].

This review concerns: (a) The benefits of the Mediterranean diet based on cohorts and case-control studies for prevention of cancer, cardiovascular risk, and cognition; (b) polyphenols as major disease-preventing components and analyses of the roles of these molecules; (c) the metabolism of polyphenols and the importance of the microbiome in generating active compounds; (d) the intestine antitumor efficiency and the mechanisms of these polyphenols and the use of nanoformulations; (e) the future developments of the Mediterranean diet, which is growing over the world, and the interest in nanoformulation supplements based on this diet.

## 2. Benefits of the Mediterranean Diet in Preventing or Treating Cancer

Taking France as an example, since 2004 cancer (152,708 death) is ahead of cardiovascular disease (147,323 death) as the leading cause of mortality. Interestingly, numerous works illustrate the benefits of the Mediterranean diet in preventing cancer and other important illnesses.

In cancer, the analysis of an Italian cohort of breast cancer suggests that the adoption of a strict Mediterranean diet is associated with a decrease of 30–50% of the incidence of its development. In addition, relapse and mortality in breast cancer is decreased in patients that have a daily exercise of at least 30 min/day [7]. A case-control study performed in Catania, Italy, on 338 patients diagnosed with colorectal cancer (CRC) compared to 676 subjects with no apparent clinical syndrome, showed a significant association between the adoption of a strict Mediterranean diet and a decrease of the relative risk of developing CRC, with an odds ratio of 0.46 [8]. In head and neck cancer, analysis of a sample of 168 subjects (100 controls and 68 cases) showed that medium-high adherence to the Mediterranean diet was associated with a lower risk of developing head and neck cancer (OR, 0.48 [95% CI: 0.20–1.07], *p* = 0.052) [9]. Very recently, Melander et al., [10] studied the prevalence of cardiovascular disease and cancer in Mediterranean and the north European middle-aged populations. They showed the odds ratio for cancer in Malmö (Sweden) versus Cilento (Italy) was 2.78 (1.81–4.27) (*p* < 0.001) in inhabitants adjusted for age and sex.

Besides the anti-cancer activity of the Mediterranean diet, other benefits have been reported. A meta-analysis on 18 studies (cohort of 4,172,412 subjects), demonstrated that consumption of the Mediterranean diet led to an 8% reduction in general mortality and a 10% decrease in the risk of development of cardiovascular illness [11]. Knowing that decreased telomere length at the chromosome ends and increased shortening rate of telomeres are aging biomarkers, a study conducted on 217 elderly reported an association between leukocyte telomere length, telomerase activity as well as benefits on health status. In subjects with a strict adherence to the Mediterranean diet, the leucocyte chromosomes showed the greatest length of telomeres (*p* = 0.003) and the highest telomerase enzyme activity (*p* = 0.013) [12]. In addition, an Irish study was conducted on 2621 subjects of an average of 25 years old at the inclusion and who were followed for 30 years, monitoring their food habits. Subjects with good adherence to the Mediterranean diet were 46% less sensitive to declines in their cognitive capacities, particularly their thought capacities. For 868 participants with strong adherence, 9% showed low thought capacities while this frequency was 29% in 798 participants with low adherence to Mediterranean diet [13]. Recently, in relation to the current pandemic situation of COVID-19, flavan-3-ols and dimeric proanthocyanidins have been reported by Zhu et al., to prevent SARS-cov2 infection [14].

In this review, a focus on intestinal cancers has been chosen for two reasons. In France, in 2018, CRC (Figure 1) was the second cause of mortality (17,117) and morbidity (43,336) after breast cancer and lung cancer (men). The intestinal tract is directly exposed to and is the primary absorption/metabolism site for dietary compounds which can be either basic nutrients for energy supply and growth, micronutrients (polyphenols, etc.), and toxins.

## 3. Key Selected Polyphenols

From several thousand different polyphenols, we have focused on species present in significant quantities in the plants, fruits, and vegetables present in the Mediterranean diet i.e., apigenin, curcumin, EGCG (Epi Gallo Catechin Gallate), quercetin, resveratrol and rutin (Figure 2). Interesting antitumor properties of these compounds have been reported in the literature [15,16,17]. We first report the general properties of these polyphenols, their origin, abundance, physico-chemical, and biological properties (Table 3). We will also present the anticancer properties of these selected polyphenols.

## 4. Metabolism and Microbiota Metabolites

### 4.1. Metabolism

#### 4.1.1. Apigenin

Like many polyphenols, apigenin is also present in nature in a glycosylated form and this flavone is at the center of complex metabolic pathways. The apigenin glycoside, after deglycosidation in enterocytes, gives apigenin. In enterocytes, apigenin can then be sulfated, glucuronidated, dehydrogenated, hydrogenated, and hydroxylated; the hydroxylated product can then be methylated and hydroxylated. Apigenin is also a major metabolite of naringin, a flavone present in very large quantities in grapefruit and pomelo. Metabolites profiles of flavonoids are usually obtained with the use of UHPLC-QTOF-MS/MS in combination with in silico approach [36].

#### 4.1.2. Curcumin

As with all polyphenols, after oral administration, the bioavailability of curcumin is low and the blood levels are reduced relative to those ingested. However, curcumin’s resistance to highly acidic pH is high and it reaches the large intestine without chemical and structural changes [37]. Curcumin is poorly absorbed from the gastrointestinal tract and is extensively metabolized in the enteric and hepatic tract. Glucuronidation of curcumin by beta-glucuronidase is a major metabolic pathway and the glucuronide form of curcumin is found in biological fluids as well as organs and cells of different species (rat, mouse, man). Curcumin can also be sulfated by sulfotransferase to give curcumin sulfate [38]. In hepatocytes and enterocytes, curcumin can also be reduced by various enzymes, including cytochrome P450 enzymes [39]. The products formed, dihydrocurcumin and tetrahydrocurcumin, can in turn be metabolized to glucuronide and sulfate derivatives [37]. Commercial products often contain demethoxycurcumin and bisdemethoxycurcumin, which undergo metabolism similar to that of curcumin [37].

#### 4.1.3. EGCG

This polyphenol is essentially absorbed in the intestine and, like all polyphenols, the intestinal flora plays an important role in its metabolism before absorption, for additional biotransformation [40]. EGCG is metabolized by tissue extracts from the small and large intestine, whereas it is only slightly metabolized by tissue extracts from the liver. Since EGCG is poorly absorbed, blood concentrations of EGCG are low and its many derivatives are found in conjugated or unconjugated forms in plasma and urine [41].

#### 4.1.4. Quercetin

This polyphenol is one of the most studied and its complex metabolism gives rise to several derivatives. Quercetin, which is essentially poorly absorbed in the small intestine, like all polyphenols, supplies metabolites that are produced by biotransformation by the intestinal microbiota and these are further absorbed in the large intestine. The resulting compounds are bioavailable, circulate in the blood as conjugates with glucuronide, methyl, or sulfate groups attached, and are eventually excreted in the urine. At the enterocyte level, several enzymes are involved in providing additional metabolites of quercetin: UDP-glucuronosyltransferase (UGT); sulfotransferase (SULT); catechol-*O*-methyl transferase (COMT); lactase phloridzin hydrolase (LPH); cytosolic β-glucosidase (CBG) [42]. The resulting derivatives of quercetin are found in plasma and urine [42].

#### 4.1.5. Resveratrol

Resveratrol is efficiently absorbed on oral administration, then appears both in plasma, urine, and to a lesser degree, in feces (Figure 3). Concerning its transport, Lançon et al. showed that human hepatic cell uptake of resveratrol involves both carrier-mediated and diffusion-dependent process [43]. In plasma, resveratrol interacts with serum albumin [44]. Furthermore binding properties between resveratrol or other polyphenols and plasma albumin have been determined in the context of consequences for the health protecting effect of dietary plant microcomponents [45]. Recently, Tung et al., reported that polyphenols bind to LDL at biologically relevant concentrations that are protective against heart disease [46].

Resveratrol metabolism leads to the production of sulfates, glucuronides derivatives, as well as dihydroresveratrol (Figure 3). The nature and quantity of these metabolites may differ between subjects due to inter-individual variability. Once in the bloodstream, metabolites can be subjected to phase II metabolism, with further conversions occurring in the liver, where entero-hepatic transport in the bile may result in some recycling back to the small intestine. We have shown in hepatic cells, that resveratrol is highly conjugated after 4 h of incubation, into mono-(3-sulfate-resveratrol and 4′-sulfate-resveratrol) and disulfate (3,4′-disulfate-resveratrol and 3,5-disulfate-resveratrol) derivatives [47]. Resveratrol can induce its own metabolism by increasing the activity of phase II hepatic detoxifying enzymes [48]. Resveratrol metabolites have a plasma half-life similar to resveratrol aglycone. Despite its low bioavailability, resveratrol in vivo efficacy can be explained by: (i) The back conversion of resveratrol sulfates and glucuronides to resveratrol in target organs such as liver; (ii) the enterohepatic recirculation involving biliary secretion of resveratrol metabolites followed by deconjugation by gut microflora and then reabsorption; (iii) the activities of its metabolites. Two sulfate metabolites show biological activity: Resveratrol-4′-*O*-sulfate binds the cyclo-oxygenase (COX-1) and inhibits enzyme activity with nearly the same efficacy of resveratrol; resveratrol-3-*O*-sulfate shows a radical scavenger activity, inhibits COX-1, and is cytotoxic. In contrast to the resveratrol 3-*O*-sulfate, resveratrol-4′-*O*-sulfate metabolite is less active than resveratrol. Piceatannol (3,5,3′,4′-tetrahydroxystilbene), another analogue of the stilbene family, is a polyphenol found in grapes and other plants; it differs from resveratrol by an additional hydroxyl group at 3′ position of the benzene ring. In humans, piceatannol is produced as a major metabolite of resveratrol by cytochromes CYP1B1 and CYP1A. Piceatannol displays anticancer properties by inducing extrinsic and intrinsic pathways of apoptosis and by blocking cell cycle progression. Trans-resveratrol metabolism by human gut microbiota only generates one metabolite: dihydroresveratrol. The resveratrol derivatives, epsilon-viniferin and their acetates have been studied at the level of their cellular uptake and their antiproliferative effects [49].

### 4.2. Microbiota Metabolites

Microbiomics is a discipline that studies bacterial and viral ecosystems. Microbiomic investigations [50] are promising for addressing the interactions between microorganisms and human cells at the level of organs in which these cells and microorganisms cohabit. At the level of the digestive tract, and the intestine in particular, there are long-lasting interactions between the intestinal flora and the nutrients ingested. At the level of the digestive tract, the activities of the intestinal cells will depend on several epigenetic factors including the microbiota present, as well as nutrients and micronutrients of food origin (Figure 4).

The proper functioning of the cells of the intestinal wall will therefore depend on their interactions with the bacterial ecosystem present and the molecules provided by the food. Food also influences the bacterial ecosystem and the latter can transform certain food molecules by metabolizing them, which is particularly important in the case of polyphenols present in large quantities in the Mediterranean diet [51]. It is currently assumed that the intestinal microbiota of subjects consuming a Mediterranean diet, rich in polyphenols and ω3 fatty acids, is capable of preventing the appearance of chronic non-transmissible degenerative diseases, such as age-related diseases (cardiovascular diseases, neurodegeneration) and certain types of cancer [52]. For polyphenols, it is well established that understanding their activity is inseparable from understanding their interaction with the microbiota. This includes (i) the biotransformation of polyphenols by the microbiota, which depends on the type of bacteria present, and (ii) the ability of polyphenols to modulate the profile of the bacterial flora [53]. In humans, the ingestion of dietary polyphenols is associated with prebiotic activity (modification of the intestinal flora and its functions) which leads to the formation of several metabolites derived from polyphenols with a beneficial effect on lipid and carbohydrate metabolism (activation of lipogenesis and glycogenesis), which are deregulated in many diseases including cancers [54]. For phytoestrogens, which are a class of polyphenols with a diphenolic structure similar to 17β-estradiol, it has been shown that they can be transformed by the microbiota [55]. The same transformation process applies to certain isoflavones, prenylfavonoids, lignans, and stilbenes such as reveratrol [55]. The metabolism of intestinal bacteria can also convert food lignans (a non-flavonoid polyphenolic class found in plant foods) into therapeutically relevant polyphenols (i.e., enterolignans) such as enterolactone and enterodiol [56]. Ellagic tannins (substances from the polyphenol family) from the pomegranate are transformed into urolithin A, a mitophagy-activating polyphenol, which increases the lifespan of the worm *Caenorhabditis elegans*, running endurance in rodents (rats, mice), and transcription of mitochondrial genes in human muscle tissue [57]. Several studies have also shown that polyphenols, by affecting the composition of the intestinal microbiota, modulate oxidative stress and immune response [58]. In agreement with these observations, red wine extracts administered orally reduced the growth of colorectal tumors (hot tumors: associated with inflammatory infiltrates) in mice by reducing the activation of T lymphocytes to Th17-positive cells [59].

The mutual interaction of polyphenols with the microbiota is therefore a determining factor in understanding their biological activities. The notion of metabolites generated by the intestinal microbiota must necessarily be taken into account in the case of oral administration. Apigenin, curcumin, EGCG, and quercetin, which are polyphenols associated with the Mediterranean diet, are all subject to transformation by the microbiota.

Apigenin or its glycosides have been shown to be degraded by certain intestinal bacteria into smaller metabolites that can have biological activities [60]. Thus, the capacity of the human intestinal microbiota to convert apigenin-7-glucoside (A7G) has been proven in vitro by incubating A7G with fecal suspensions: apigenin, naringenin, and 3-(4-hydroxyphenyl) propionic acid are formed as the main metabolites [61]. After application of A7G to germ-free rats, apigenin, luteolin, and their conjugates were detected in urine and feces. In human microbiota-associated (HMA) rats (comparatively to germ-free animals), naringenin, eriodictyol, phloretin, 3-(3,4-dihydroxyphenyl)propionic acid, 3-(4-hydroxyphenyl)propionic acid, 3-(3-hydroxyphenyl)propionic acid, and 4-hydroxycinnamic acid were also formed in their free and conjugated forms. Furthermore, when apigenin is studied under culture conditions mimicking the ascending colon, it modifies the multiplication and gene expression of the *Enterococcus caccae* bacterium, whereas no effect has been observed on *Bifidobacterium catenulatum* and *Lactobacillus rhamnosus* G [62].

As for curcumin, which is a lipophilic polyphenol with low systemic bioavailability, it is largely biotransformed by human intestinal microflora to give different metabolites that are easily conjugated to glucuronides and *O*-conjugated sulfate derivatives [63]. It is also suspected that curcumin and its derivatives have direct regulatory effects on the intestinal microbiota. These findings may explain the paradox between the low systemic bioavailability of curcumin and its widely reported pharmacological activities [64].

Epigallocatechin-3-gallate (EGCG), one of the major phenolic compounds in green tea, has versatile bioactivity and exerts a strong anti-cancer effect. EGCG is mainly absorbed in the gut, and the intestinal microbiota plays an essential role in its metabolism prior to absorption [40]. The interaction of green tea phenolic compounds, particularly EGCG, with the intestinal microbiota may play a key role in the health benefits by generating beneficial bioactive compounds. When the metabolic fate of EGCG and its impact on the intestinal microbiota was studied through in vitro fermentation, it was shown by ultra-high performance liquid chromatography and Orbitrap quadrupole hybrid mass spectrometry (UHPLC-Q-Orbitrap-MS), that EGCG is rapidly degraded to a series of metabolites, including 4-phenylbutyric acid, 3-(3′,4′-dihydroxyphenyl)propionic acid and 3-(4′-hydroxyphenyl)propionic acid, by subsequent hydrolysis of the esters, opening of the C-ring, fission of the A-ring, dihydroxylation, and shortening of the aliphatic chain. Microbiome profiling indicated that, compared to negative controls, treatment with EGCG resulted in the stimulation of the beneficial bacteria *Bacteroides*, *Christensenellaceae*, and *Bifidobacterium*, while the growth of pathogenic bacteria *Fusobacterium varium*, *Bilophila,* and *Enterobacteriaceae* was inhibited [65].

Like other polyphenols, quercetin, which is a plant polyphenol widely present in the Mediterranean diet, is metabolized by the microbiota and acts on the composition and activity of the intestinal flora. As with other polyphenols, it is crucial to better understand the impact of the microbiota on quercetin in order to better understand its effects on health [66]. Thus, quercetin is bioaccessible to the microbiota in the colon, which metabolizes it by producing phenylpropanoic acid, phenylacetic acid, and benzoic acid derivatives [67]. In addition, the effects of quercetin have been documented on the intestinal commensal microbes *Ruminococcus gauvreauii*, *Bifidobacterium catenulatum*, and *Enterococcus caccae* [68]. Quercetin does not inhibit the growth of *Ruminococcus gauvreauii*, it slightly suppresses the growth of *Bifidobacterium catenulatum*, and moderately inhibits the growth of *Enterococcus caccae*. Transcriptomic analysis revealed that in response to quercetin, *Ruminococcus gauvreauii* negatively regulates the expression of genes responsible for protein folding, purine synthesis and metabolism; *Bifidobacterium catenulatum* increases the expression of the ABC transport pathway, decreases metabolic and cell wall synthesis pathways; *Enterococcus caccae* upregulates the genes responsible for energy production and metabolism and downregulates the stress response, translation, and sugar transport pathways [68].

Thus, a better understanding of the interactions of polyphenols with the intestinal microbiota will lead to a better knowledge of their biological activities and optimize their use in prevention or treatment of diseases such as cancer. Furthermore, the identification of metabolites resulting from their degradation by the intestinal microbiota may lead to the discovery of new molecules with greater pharmacological activities than the natural molecules from which they are derived.

## 5. Antitumor Efficiency

### 5.1. Apigenin

The Mediterranean diet contains high quantities of flavonoids, a group representing 60% of natural polyphenols. Apigenin is the most commonly occurring flavonoids found in several dietary constituents [69]. A diet rich in flavonoids may be considered useful to fight cancer development [70]. It has been reported that apigenin is able to suppress cell growth in many types of human cancer cells: leukemia [71], melanoma [72], breast cancer [73], bladder cancer [74], thyroid cancer [75], and colonic cancer [76]. Apigenin has multiple anti-cancer effects [69]: anti-proliferation, anti-inflammation, and anti-metastatic activities. It inhibits growth of different types of cancer. Many signaling pathways are involved on the anti-carcinogenic mechanism of apigenin; the best known of which are MAPK, NF-κB, P53, signal transducer, and activator of transcription (STAT3), PI3K/Akt, cyclooxygenase (COX), and Wnt/β-catenin pathway [77] (Table 4).

### 5.2. Curcumin

Curcumin acts as anti-inflammatory, anti-oxidant, and antiproliferative agent. It has been recommended for chemopreventive, anti-metastatic, and anti-angiogenic purposes [84]. Several studies have shown that curcumin as an anti-carcinogenetic agent, is used on many type of tumors such as prostate, pancreas, breast, stomach, liver carcinomas, and leukemia [85]. Despite its limited bioavailability observed by oral administration, curcumin modulates several illnesses especially intestinal inflammatory pathologies such as Crohn’s disease, necrotizing enterocolitis, and ulcerative colitis [86]. Epidemiological studies have shown that patients with long-term intestinal inflammation present a risk of developing colorectal cancer [87]. Curcumin is taken up by intestinal epithelial cells [88]. Curcumin inhibits anti-apoptotic Bcl-2 proteins family in intestinal epithelial cells and stimulates the expression of p53, Bax, procaspases 3, 8, and 9 and the pro-apoptotic Bax protein [89]. Curcumin binds vitamin D receptor present in intestinal epithelial cells involved in the promotion of colon cancer chemopreventing process [90]. Table 5 summarizes the antitumor effects of curcumin in colon cancer, from in vitro and in vivo models and human clinical trials (Table 5).

In vivo studies performed in rats show that curcumin elimination is slower by intravenous (i.v.) administration (44.5 min) for a dose of 1 g/kg [91], than when orally administered (28.1 min) at a dose of 300 mg/kg [92]. Results from human clinical trials [93], demonstrate that curcumin bioavailability was increased to about 20-fold when combined with piperine. Di Silvestro et al., [94] showed that lipidated curcumin increases its bioavailability.

### 5.3. EGCG

Epigallocatechin-3-gallate (EGCG) is a major biologically active component and the most abundant polyphenol from green tea [101]. Its antitumoral activities in colorectal cancer are well documented (Table 6).

Epidemiological investigations and experimental studies have shown that EGCG exerts an important role by its anticancer and cancer chemopreventive properties. It has been observed to suppress breast cancer [101], lung cancer [102], pancreatic cancer [103], and liver cancer [104]. In vivo studies have demonstrated an inhibitory effect of EGCG on inflammation and obesity-related colon carcinogenesis in rodent models [105]. Studies have demonstrated that EGCG exerts its inhibitory role in colorectal cancer and its metastasis by suppressing cell proliferation, inducing apoptosis in human CRC cells [106]. EGCG plays a crucial role in obstructing and inhibiting colorectal cancer sphere formation and proliferation, respectively [107]. The possible molecular mechanisms of EGCG in colorectal cancer involve various molecules and signaling pathways [108] (Table 5).

### 5.4. Quercetin

Quercetin has been shown to have anti-tumor efficacy against intestinal and colon cancer cells both in vitro and in vivo (Table 7).

In vitro exposure of colon cancer cells such as HT-29, Caco-2, SW480, Colo-320, and Colo-741 to different concentrations of quercetin (5–200 μM), induced apoptosis (indicated by chromatin condensation and internucleosomal DNA fragmentation) through procaspases activation coincident with an alteration in the expression of c-myc, p53, p16 and p21, or by inhibition of NF-κB pathway [129]. Another mechanism supporting the anticarcinogenic effect of quercetin is through downregulation of many cell cycle genes which cause cell cycle arrest, and suppression of tumor cell invasion and migration by regulating the expression of matrix metalloproteinases (MMPs), such as β-catenin [130], which can act as an invasion suppressor, and E-cadherin which is lost during tumor metastasis in association with the epithelial mesenchymal transformation [130]. Quercetin can also influence colon cancer cell proliferation and differentiation by downregulating the expression of cyclin D1 [127] and CDC6 [124]. The anti-inflammatory effect of quercetin at 5–20 μM has also been described in Caco-2 and DLD-1 cells as a result of inhibition of cyclooxygenase-2 (COX-2) [131] and cytokine expression [130].

Multiple studies confirmed quercetin’s effectiveness in reducing tumors in mice and rats, which is consistent with its in vitro effects. Quercetin at 50 mg/kg/day causes a reduction in tumor numbers in rats treated with 1,2-dimethylhydrazine (DMH) via activating the antioxidant defenses and limiting lipid peroxidation and protein peroxidation [131,132,133,134,135,136]. In CRC mice, quercetin decreased colon damage and mortality while reducing certain CRC-associated gut bacteria, such as *Allobaculum* and unclassified *Erysipelotrichaceae* [137]. Cruz-Correa M et al. [138], showed that after a mean of 6 months of treatment, a combination of curcumin and quercetin reduced the number and size of polyps in patients with Familial Adenomatous Polyposis without causing any significant toxicity. Although more research is required to fully understand the relationship between quercetin consumption and colon cancer in humans, regarding its bioavailability (that may be affected by the food source) and the presence of other bioactive compounds in foods that can interact with quercetin and modulate tumor risk.

### 5.5. Rutin

Many mechanisms have been suggested to explain the in vitro anti-carcinogenic effects of rutin (Table 8).

Rutin induces apoptosis (increase in the levels of proteins associated with apoptotic cell death: phospho-Bad, cleaved caspase 3, and cleaved PARP) and inhibits cell proliferation and adhesion (disrupting cell–extracellular matrix interactions) in HT-29, Caco-2, and HCT116 cells at concentrations ranging from 25 to 1000 μM [140]. When Caco-2 cells were exposed to rutin (over 1000 μM), the antiproliferative effects of 5-fluorouracil (5-FU) and oxaliplatin were enhanced and the potential side effects of these drugs were decreased, suggesting that rutin may have chemopreventive and chemotherapeutic effects on colon cancer. In dextran sulfate sodium (DSS) chemically induced mouse model of colitis, 6 mg/day of rutin suppressed the induction of pro-inflammatory cytokines, resulting in improved symptoms [146]. Another study showed that the administration of rutin (50, 100 mg/kg/day) offered an extensive protection against methotrexate (MTX)-induced intestinal lesions in rats [144]. This may be due to rutin’s anti-inflammatory activities (inhibition of COX-1, COX-2, and 15 LOX enzymatic activity) and free radical scavenging properties (restoration of MDA, protein carbonyl, SOD, GSH levels, and catalase activity).

### 5.6. Resveratrol

Cancer chemopreventive activity of resveratrol, a natural product derived from grapes was first reported by Jang et al. [147]. A review on cancer prevention potential of resveratrol has been recently published [148] and includes: case-control trials (resveratrol and breast cancer risk [149]); clinical studies (clinical pharmacology of resveratrol and its metabolites in colorectal cancer patients [150,151]. In vivo studies reported the antitumor effect of resveratrol on the development of spontaneous mammary tumors in HER-2/neu transgenic mice [152]), the impact of metastasis [153], the effect of resveratrol metabolites [154], and targets and mechanistic aspects [155].

Resveratrol shows all of the antiproliferative and pro-apoptotic properties necessary to be an antitumor agent [156]. Conversely, P-glycoprotein 1 affects chemoactivities of resveratrol against human colorectal cancer cell [157]. After cell exposure, resveratrol is accumulated in lipid rafts, which is the first step for endosome formation and triggers apoptosis [158]. Signaling molecules and transcription factors involved in the cell cancer promotion or prevention are sensitive to resveratrol: PPAR, PGC1α, NF-kB, NRF1,2, p53. Nuclear cell cycle components, phosphatases, and cyclins are targets of resveratrol [148].

Resveratrol modulates both oncogenic miRNAs and the expression of tumor-suppressor miRNAs. In SW480 colon cancer cell line, resveratrol treatment decreases the levels of several oncogenic miRNAs targeting genes encoding tumor suppressors and effectors of TGF**β** signaling pathway, while increasing the levels of miR-663. While upregulating several components of the TGF**β** signaling pathway, resveratrol decreases the transcriptional activity of SMADs, the main effectors of the canonical TGF**β** pathway [159].

Resveratrol inhibits human leiomyoma (a benign smooth muscle tumor) cell proliferation via crosstalk between integrin αvβ3 and IGF-1R [160].

Levi et al. [149] analyzed the relationship between dietary intake of resveratrol and breast cancer risk from a case-control study, conducted between 1993 and 2003 in the Swiss Canton of Vaud, on 369 cases and 602 controls. A significant inverse association was observed for resveratrol from grapes (OR = 0.64 and 0.55), but not from wine, suggesting that daily p.o. doses of resveratrol at 0.5 or 1.0 g produce levels in the human gastrointestinal tract in the order of magnitude sufficient to elicit anticarcinogenic effects.

### 5.7. Synergistic Effects of Polyphenol Mixtures

While these effects are mostly obtained in vitro or in small laboratory animals at concentrations that are too high to be compatible with its in vivo low availability, resveratrol can slow down tumor progression in numerous cancer types, for instance blood [161], breast [162], cardiac [163], colon [164], glioma [165], lung [166], melanoma [167], and prostate [168]. As resveratrol is not cytotoxic it can be used as food supplement at high concentrations compatible with significant plasma levels. Moreover, the synergistic effects of other food polyphenols [169], and the fact that resveratrol plasma metabolites also exhibit antiproliferative activities, is relevant for diet-dependent cancer prevention [154]. Interestingly, Mazue et al. reported protective effects of red wine extract in rat colon carcinogenesis [170].

## 6. Perspectives

### 6.1. Improvements

The health benefits of Mediterranean diet are now well established. To go further, the goal would be to expand this diet over the world in order to benefit more and more populations. This is a question of public health which needs to be relayed by the media. On the other hand, comparable polyphenols-rich diets are also observed in non-Mediterranean coastal regions at the same latitude, with analogous cooking and eating habits. Besides this, more experimental approaches are needed, especially at the preclinical and clinical studies (case-control studies) and information provided from cohorts. A better knowledge of the possible synergies between single polyphenols in mixtures with other nutrients should be accomplished. We may also pay attention to the influence of immune system in cancer prevention under a Mediterranean diet. Related to the role of microbiome, the following question should be addressed: is there any feedback between the dietary polyphenol-modification of the microbiome ecosystem to the cancer immune defense dependency, or from the gut (second brain) to the central nervous system? Indeed, polyphenol-microbiota interactions have led to reconsideration of the gut-brain axis, initially revealed by the capacity of certain enteric nerve cells to secrete serotonin (5-hydroxytryptamine) involved at the brain level in the control of behavior, mood, and anxiety [171]. Several studies highlight the intervention of the polyphenol-microbiota not only in the occurrence of neuropsychiatric diseases such as schizophrenia [172] but also in metabolic diseases [173]. The intestinal microbiota, by acting on immune, endocrine, and metabolic signaling, is thus able to exert beneficial clinical effects on depression and anxiety [174]. The term psychobiome takes into account the intestinal microbiota by considering its capacity for endocrine substances intervening at the level of several pathways involved in the modulation of neuro-inflammation and in the production of several neurotransmitter precursors [175]. In the context of polyphenol-microbiota interactions, beneficial effects of quercetin as well as curcumin [173], resveratrol [176], and EGCG [41] have been shown. The prevention of anxiety by polyphenols could be beneficial for cancer patients, particularly those with CRC, by providing better psychological support for restrictive therapies.

### 6.2. Nanoformulation Improvement

As described in Section 6.1, the selected polyphenols in this review provide numerous health benefits. However, the problem of their limited uptake from the diet is difficult to solve, because of their poor aqueous solubility and therefore their weak bioavailability. The nanoformulation of bioactive polyphenols may overcome these problems. The benefits of this concept is that it is able to transport and release the biological compound to target sites, without modifying its molecular structure. In addition, the transport of poorly aqueous soluble compounds by a molecular-carrier allows improvement of its bioavailability and therefore, its biological activities [177]. The polyphenols mentioned in this review have been subjected to various forms of nanoformulation, especially resveratrol [178], curcumin [179], quercetin [180], catechins [181], and to a lesser degree, rutin [32] and apigenin [182]. It appears from the literature that in vitro and in vivo bioactivities of nanoformulated polyphenols are generally much better than those obtained with free polyphenols, particularly in the cancer field. While quercetin is poorly soluble in water, the loading of this polyphenol in biodegradable monomethoxy poly(ethyleneglycol)-poly(ε-caprolactone micelle (MPEG-PCL) makes possible its complete dispersion in water [183]. Compared to free quercetin**,** quercetin-loaded MPEG-PCL micelles showed a better in vitro inhibition of growth of CT26 cells with a weaker IC50 and better induction of apoptosis. Similarly, quercetin-loaded MPEG-PCL micelles repressed colon tumor in mice subcutaneous CT26 colon cancer [184]. To further target the biomarkers overexpressed by colon cancer cells such as EpCAM (epithelial cell adhesion molecules), nanoparticles may be decorated with specific ligands. Thereby, apigenin-loaded nanoparticles were conjugated with aptamer. The resulting aptamer-conjugated apigenin-loaded nanoparticles were evaluated for their antiproliferative properties toward colon carcinoma cells in vitro, as well as an in vivo colorectal cancer model. Using pharmacokinetic and biodistribution studies, the authors highlighted a greater retention of aptamer-conjugated apigenin-loaded nanoparticle in the colon as compared to free apigenin and aptamer-conjugated empty nanoparticles [182]. Encapsulation of resveratrol in liposomes allows increased aqueous solubility, photostability, and bioavailability of this polyphenol [185]. In addition, resveratrol-loaded liposomes target cancer cells effectively and may show a greater apoptotic activity of HT29 colon cancer cells than free resveratrol [186]. Dual encapsulation involving liposomes and cyclodextrins improves the nanoformulation of resveratrol and its release to HT29 colon cancer cells. Indeed, cyclodextrin stabilizes the polyphenol in its hydrophobic cavity and liposome targets cancer cells [187]. Resveratrol may be encapsulated in PLGA-PEG (poly lactic glycol acid–poly ethylene glycol), a biodegradable and biocompatible polymeric nanoparticle coated with chitosan. The resulting resveratrol-loaded micelles were evaluated in vivo for their anti-proliferative and anti-angiogenic effects in human COLO205-luc colon cancer in xenograft and orthotopic implantation models in athymic mice [188]. In this study, the authors observed decreased tumor growth due to the action of resveratrol-loaded micelles. The anti-angiogenic effect of these resveratrol nanoparticles have been highlighted by the decrease of the Hb percentage of tumor mass. To explain these promising results, the authors put forward the idea of greater bioavailability of resveratrol-loaded micelles compared to those of free resveratrol.

Curcumin nanoformulation applied to colorectal cancer has been the object of a complete and recent review [189]. Various nanoformulations have been mentioned in particular liposomes, micelles, polymer nanoparticles, cyclodextrins. This review highlighted the advantages of most of the curcumin-loaded nanoparticles but also the limitations of others, such as some cyclodextrin and lipid nanoparticle formulations. The combination of successful results of in vitro and in vivo studies involving curcumin-loaded nanoparticles have allowed the commencement of clinical trials evaluation of the effects of these on intestinal cancer. Synergistic effects between two or more bioactive molecules may afford enhanced activities or complementary activities, more so when these molecules are nanoformulated in the same system. The molecular structures of the co-encapsulated bioactive compounds may be similar or very different from those of the selected polyphenols, as shown in Figure 5. For example, alantolactone (**7**) is a natural sesquiterpene lactone (Figure 5) isolated from *Inula racemose* Hook. f. showing antitumor activities through an apoptotic mechanism [190]. The combination of quercetin and alantolactone into micelles allowed induction of immunogenic cell death in microsatellite-stable CRC [191]. Chrysin (**8**), a natural flavone whose structure is similar to that of quercetin or apigenin (Figure 5) provides multiple properties including anti-tumor activities [192]. Thus, evaluation of activity of curcumin-chrysin loaded in PLGA-PEG nanoparticles toward Caco-2 cancer cells has shown a greater efficiency than either of the individual compounds: a quicker killing of cancer cells and a more efficient suppression of hTERT expression [193]. Piperine (**9**), an alkaloid isolated from black and long peppers (Figure 5), is known for its ability to inhibit proliferation of HT-29 colon carcinoma cells [194]. The synergistic effect of curcumin and piperine (non-nanoformulated) has been demonstrated in an in vivo experiment against hepatocellular carcinoma in rats [195]. The co-encapsulation in emulsomes of these both bioactive compounds confirmed their synergistic benefits toward HT116 colorectal cancer cells [97].

### 6.3. Future Prospects

Polyphenols may be absorbed by humans via the Mediterranean diet and in this condition, may play a preventative role against cancers and other diseases. To strengthen this prevention, the intake of polyphenols in the form of food supplements has been developed and some of these are widely marketed. In this field, the use of nanotechnology is an attractive concept. In a complete and interesting review, D.D. Milincic et al., reported various polyphenol nanoformulations applied in food industry [196]. The authors highlighted advantages of nanotechnology in preserving healthy benefits of the bioactive compounds, especially stability, anti-oxidant properties, and intestinal bioavailability. However, they warned about an uncontrolled consumption of polyphenols which can lead to harmful effects in humans, because these compounds at high doses may promote pro-oxidant effects [197]. In addition, polyphenols at high doses can chelate bio-elements and decrease their absorption by the gastrointestinal tract [198]. Indeed, nanoparticle toxicity must be not neglected and the use of nanoformulated polyphenols for therapeutic applications and food supplements requires further safety-risk assessments.

In conclusion, the future development of nanoformulations should provide new properties of dietary polyphenols, to benefit cancer prevention, especially due to increased polyphenols cell absorption, plasma level, and targeting.

## Figures and Tables

**Figure 1 molecules-26-03483-f001:**
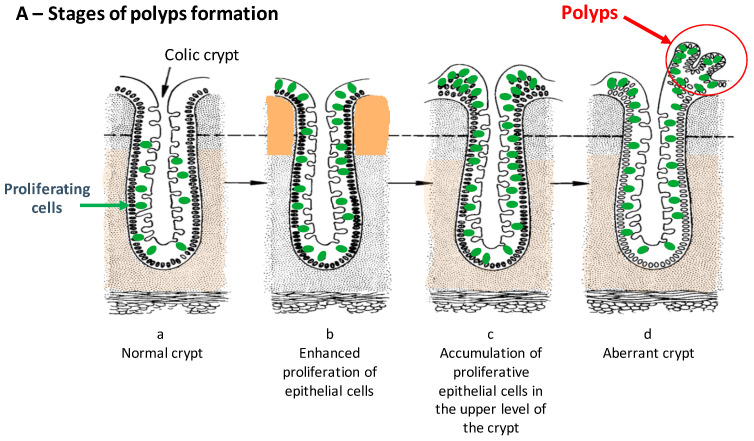
(**A**) Polyps as precancerous intestinal cancer, (**B**) colonic carcinogenesis stages. Adapted from the PhD Thesis of Dr F. Mazué (Director of Thesis; Prof. N. Latruffe; https://www.theses.fr/165862416, accessed on 7 June 2021).

**Figure 2 molecules-26-03483-f002:**
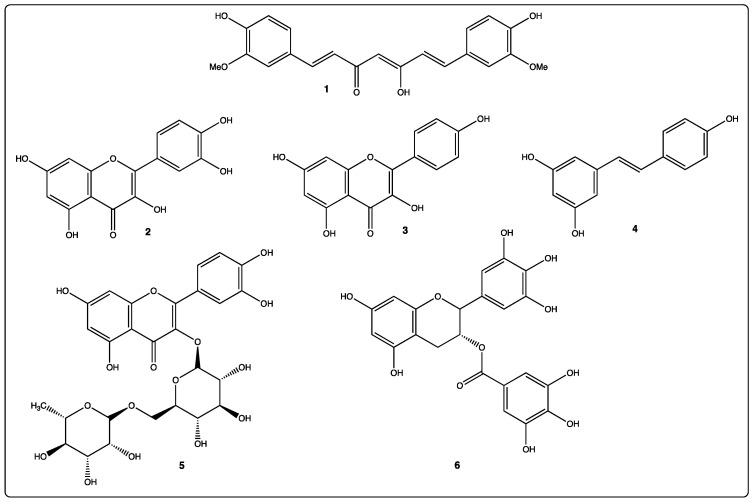
Chemical structures of selected polyphenols: 1: curcumin; 2: quercetin; 3: apigenin; 4: resveratrol; 5: Rutin; 6: EGCG.

**Figure 3 molecules-26-03483-f003:**
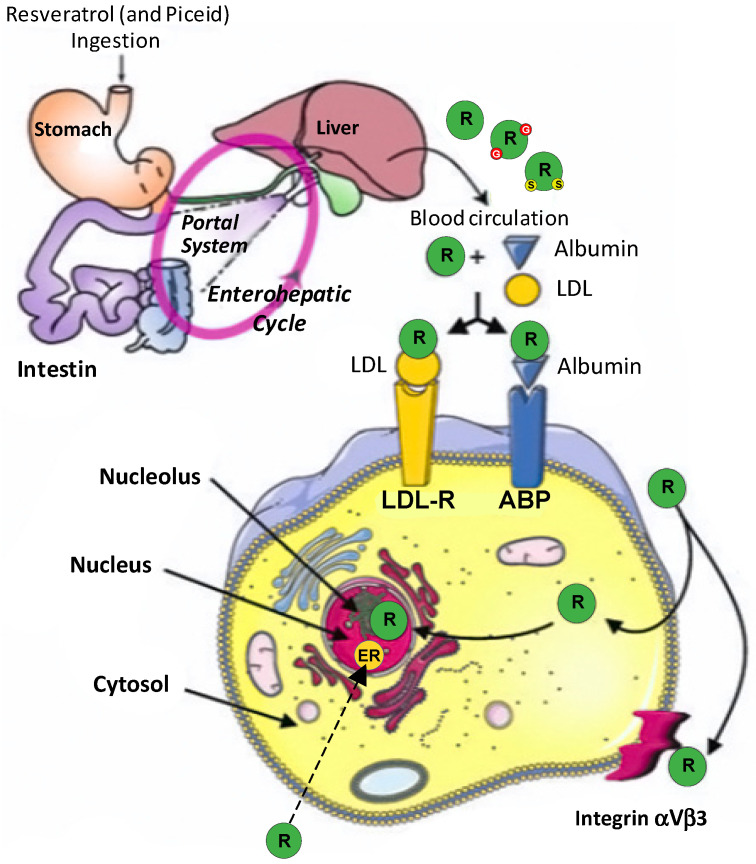
Metabolism of resveratrol. Adapted from Delmas et al., [47]. ER: estrogen-receptor; ABP: albumin-binding protein; LDL: low density lipoprotein; LDL-R: low density lipoprotein-receptor. 
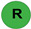
 resveratrol; 
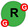
 glucurono resveratrol; 
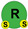
 sulfo resveratrol.

**Figure 4 molecules-26-03483-f004:**
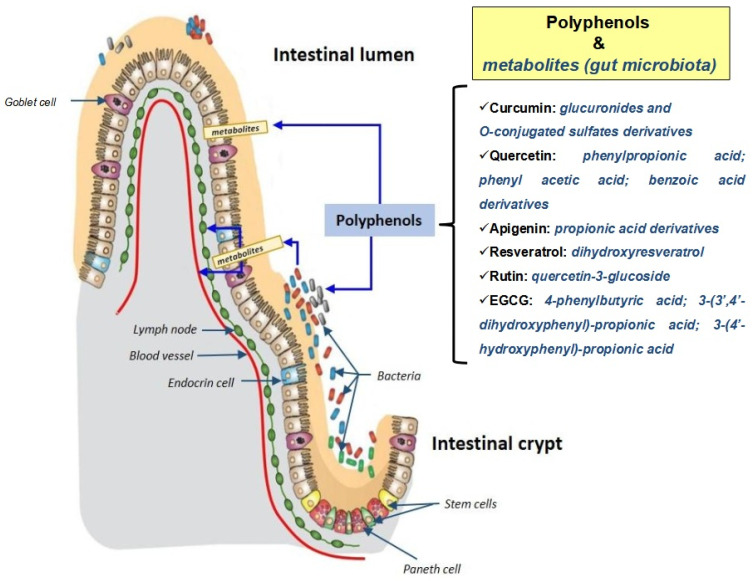
Contribution of the gut microbiota to polyphenol metabolism. The metabolites of polyphenols (curcumin, quercetin, apigenin, resveratrol, rutin and EGCG) produced by the gut microbiota are written in blue.

**Figure 5 molecules-26-03483-f005:**
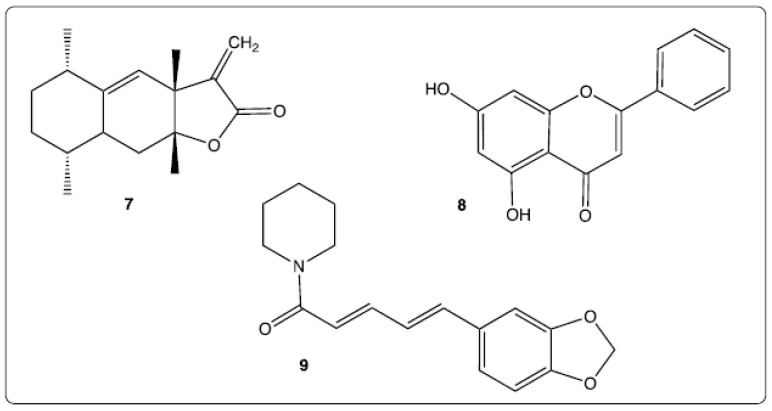
Chemical structures of co-encapsulated compounds: alantolactone (**7**), chrysin (**8**), piperine (**9**).

**Table 1 molecules-26-03483-t001:** Total polyphenols content of main edible plants from Mediterranean diet. Adapted from Debbabi et al. [2].

Fruits	Mean of Total Polyphenol Content (mg of GAE/100g Fresh Edible Portion)	Vegetables	Mean of Total Polyphenol Content (mg of GAE/100 g Fresh Edible Portion)
Strawberry	263.8	Artichoke heart	321.3
Lychee	222.3	Parsley	280.2
Grape	195.5	Brussels sprout	257.1
Apricot	179.8	Shallot	104.1
Apple	1179.1	Broccoli	98.9
Date	99.3	Celery	84.7
Cherry	94.3	Onion	79.1
Fig	92.5	Eggplant	65.6
Pear	69.2	Garlic	59.4
White nectarine	72.7	Turnip	54.7
Passion fruit	71.8	Celeriac	39.8
Mango	68.1	Radish	38.4
Yellow/white peach	59.3–44.2	Pea	36.7
Banana	51.5	Leek	32.7
Pineapple	47.2	Red bell pepper	26.8
Lemon	45	Cherry tomato	26.6
Grape fruit	43.5	Potato	23.1
Orange	31	Zucchini	18.8
Clementine	30.6	Green bell pepper	18.2
Lime	30.6	Chicory	14.7
Kiwi	28.1	Asparagus	14.5
Watermelon	11.6	Tomato	13.7
melon	7.8	Fennel	13
		Cauliflower	12.5
		Carrot	10.1
		French string bean	10
		Avocado	3.6

**Table 2 molecules-26-03483-t002:** Fruit and vegetables with high polyphenols content characterized by important anti-oxidant activity. Adapted from Debbabi et al. [2].

Extracts	Polyphenols Expressed in (mg/100 g of Fresh Matter)
Muscat grape	Gallic acid (1.7), epigallocatechin (10.8), catechin (21.8), epicatechin (5.3), quercetin-3-*O*-beta-*d*-glucuronide (10.8)
White grape	Gallic acid (1.1), caftaric acid (10.8), epigallocatechin (4.4), catechin (10.6), epicatechin (6.0), quercetin-3-*O*-beta-*d*-glucuronide (1.7)
Strawberry	Epigallocatechin (6.3), catechin (7.1), epicatechin (4.1), epicatechin gallate (1.3), quercetin-3-*O*-beta-*d*-glucuronide (15.4), kaempferol-3-*O*-glucoside (2.6)
Raspberry	Rutin (1.6), quercetin-3-*O*-beta-*d*-glucuronide (1.7)

Beans	Catechin (1.0), myricetin-3-*O*-glucoside (12.0), rutin (0.8), kaempferol-3-rutinoside (8.1)
Tomato	Chlorogenic acid (2.4), myricetin-3-*O*-glucoside (0.3), rutin (1.3), kaempferol-3-*O*-glucoside (0.1), quercetin (0.4)
Celery	Chlorogenic acid (75.3), quercetin-3,4′-diglucoside (2.3), rutin (368.9), quercetin-3-beta-glucoside (5.7), kaempferol-3-rutinoside (22.1)
Radish	Catechin (2.7)

**Table 3 molecules-26-03483-t003:** Main characteristics of selected polyphenols.

Polyphenol *Family*	Mol.Formula Mol. Weight	Water Solubility (mg/L)	Main Dietary and Geographic Sources	Ref.
1: Curcumin *Curcuminoid*	C_21_H_20_O_6_ 368.35 g/mol	0.125 mg/L	Rhizomes of C*urcuma longa* (turmeric) India	[18,19]
2: Quercetin *Flavonoid*	C_15_H_10_O_7_ 302.20 g/mol	0.48 mg/L (MilliQ water)	Red grape, onion, broccoli, tomato, lettuce	[20,21,22,23]
3: Apigenin *Flavonoid*	C_15_H_10_O_5_ 270.21 g/mol	1.43 mg/L (pH = 1)	Chamomile (Europe, Western Asia)	[24,25]
4: Resveratrol *Stilbenoid*	C_14_H_12_O_3_ 228.24 g/mol	30 mg/L	Grapes and red wine, Peanuts, blueberries, rhubarb	[26,27,28,29,30]
5: Rutin *Flavonoid*	C_27_H_30_O_16_ 610.44 g/mol	130 mg/L	Red grape, citrus, apple, fig, asparagus, onion, mulberry, tea	[21,31,32]
6: EGCG *Flavonoid*	C_22_H_18_O_11_ 358.32 g/mol	5733.12 mg/L	Green tea from leaves and buds of *Camellia sinensis* Japan, Chocolate, Red grape	[21,33,34,35]

**Table 4 molecules-26-03483-t004:** Antitumoral effects of apigenin in colorectal cancer.

		APIGENIN		
Experimental Models	Concentration Range	Biological Response	Pathway/Genes/Proteins Involved	Refs.
Cells and Cell Lines				
HCT-15	43.28 µM	↑ Cell cytotoxicity ↑ Apoptosis ↑ Cell cycle arrest G2/M	↑ p21, ↑ cyclin B1	[76]
SW480	40 µM	↓ Proliferation ↓ Invasion ↓ Migration	↓ Wnt/β-catenin	[78]
		↓ Cell migration, ↓ Invasion, ↓ Metastasis	↓ FAK, Src, crk-L, AKT	[79]
		↓ Proliferation	↓ NEDD9	
HCT-116	25 µM	↓ Proliferation, ↓ Apoptosis ↓ Autophagy	↓ Cyclin B1, ↓Cdc2, Cdc25c, ↑ PPAR cleavage, ↑ LC3-II	[80]
	20 μM and 40 μM	↑ Autophagy/Apoptosis ↓ Cell grouth, cell cycle arrest G2/M	↓ PI3K/AKT/Mtor	[81]
	10 µM	↑ Apoptosis, ↓ Trascriptional level	PKCδ/ATM kinase, ↓ NAG-1, ↓ p53, ↓ p21	[82]
LoVo	1–10 µM	↑ Apoptosis	↓ NAG-1, ↓ p53, ↓ p21	
DLD-1	40 µM	↓ Cell migration, ↓ Invasion, ↓ Migration	↓ NEDD9, ↓ FAK, Src, crk-L, AKT, ↑ TAGLN, ↓ MMP-9	[79]
		↓ Cell migration, ↓ Invasion,	↑ TAGLN, ↓ MMP-9	[83]
HT-29	45.96 µM	↑ cell cytotoxicity ↑ Apoptosis	↑ p21, ↑ cyclin B1	[76]
Animal models				
Athymic nude mice	20 mg/kg (I.P)	↓ Cell migration, ↓ Invasio ↓ proliferation n	↓ FAK, Src, crk-L, AKT	[79]
			↓ NEDD9	
	50 mg/kg (I.P)	↓ Angigenesis ↓ proliferation	↓ CD-31, ↓ Ki-67	[76]
BALB/c-nude mice	50 mg/kg (Per Os)	↓ Cell migration, ↓ Invasion, ↓ Proliferation	↑ TAGLN, ↓ MMP-9	[83]
APCMin/+ mice	50 mg/kg (I.P)	↓ tumor volume, ↑ Apoptosis	↑ p21, ↓ p53	[82]

**Table 5 molecules-26-03483-t005:** Antitumoral effects of curcumin in colorectal cancer.

Curcumin
Experimental Models	Concentration Range	Biological Response	Pathway/Genes/Proteins Involved	Refs.
Cells and Cell Lines
HCT-116	10–25 µM	↑ Apoptosis	↓ AP-1, ↓ NF-κB, ↓ MMP-9	[95]
	20 µM with 5-FU (5 µM)	↑ Cell cycle arrest (S) ↑ Apoptosis↓ Cell proliferation	↓ caspase-3, ↓ caspase-8, ↓ caspase-9, Bax, ↓ PARP, ↑ Bcl-2	[96]
	↓ cyclin D1	
	25 µM with Piperine (7 µM)	↓ Cell proliferation↑ Cell cycle arrest (G2/M↑ Apoptosis	↓ cyclin D1, ↑ caspase-3	[97]


HT29	41 µM	↓ Oxydative stress↓ Cell growth, ↓ Invasion, ↓ Metastasis	↓ NF-E2, ↓ Nrf2↓ Bcl-2, ↓ Cyclin D1, ↓ IL6, ↓ Cox2	[17]

HCT-8/5-Fu	10 µM with 5-FU (10 mM)	↑ Apoptosis,	↑ Nrf2, ↑ Bcl-2, ↓ Bax	[98]
Animal models
C57BL/6	300 mg/kg with DSS (5 mg/kg) I.P.	↓ Disease activity index, ↓ neoplasic lesions	↓ β-catenin, Cox2, iNOS	[99]
	↑ Apoptotosis	↓ cyclinD1, ↓ cyclinD3, ↑ caspase-3, ↑ caspase-7,↑ caspase-9, ↑ PARP	[92]
Oxaliplatin-resistant HCT116-xenograft	(1 g/kg) per os	↑ Radiosensitivity	↓ NF-κB, ↓ Ki-67, ↓ Notch-1	[100]
Orthopically implanted CRC tumors (HC116)	(1 g/kg) per os	↓ Cell growth, ↓ Metastasis	↓ NF-κB	[91]

**Table 6 molecules-26-03483-t006:** Antitumoral effects of EGCG in colorectal cancer.

		EGCG		
Experimental Models	Concentration Range	Biological Response	Pathway/Genes/Proteins Involved	Refs.
Cells and Cell Lines				
SW837	50–100 µM	↓ Cell growth	↑ IFN-γ, ↓ IDO, ↓ STAT1, ↓ JAK/STAT1, ↓ ISRE, ↓ GAS	[109]
	10 ng/mL	↓ Cell proliferation, ↑ Apoptosis	↑ CD133, CD44, ALDHA1, Oct-4, and Nanog, ↓ p-GSK3β, ↑ GSK3β, ↓ Wnt, ↓ β-catenin, ↓ Cyclin D1, ↓ PCNA, ↓ Bcl2, ↑ Bax, ↑ caspases 3, 8 and 9	[110]
	10–30 µM	↓ Cell growth, ↑ Cell cycle arrest (G2/M), ↓ Proliferation, ↓ Cell invasion, ↓ Cell adhesion, ↑ Apoptosis	↓ MMP2/9, ↑ caspases 3, 8 and 9, ↓ EGFR and IGF1R, ↓ MEK and ERK, ↓ PI3K and AKT, ↓ Bad	[111]
	35 µg/mL	↑ Cell cycle arrest (G0/G1) ↓ Cell proliferation, ↑ Apoptosis		[112]
LoVo	35 µg/mL	↓ Cell proliferation, ↑ Apoptosis, ↑ Cell cycle arrest (G0/G1)		[112]
	10–30 µM	↓ Cell growth, ↑ Cell cycle arrest (G2/M), ↓ Proliferation, ↓ Cell invasion, ↓ Cell adhesion, ↑ Apoptosis	↓ MMP2 and 9, ↑ caspases 3, 8 and 9, ↓ EGFR and IGF1R, ↓ MEK and ERK, ↓ PI3K and AKT, ↓ Bad	[111]
HT29	35 µg/mL	↑ Cell cycle arrest (S)		[112]
	88 μM 262 μM 190 μM/88 μM 262 μM/190 μM/88 μM 262 μM/190 μM/88 μM 88 μM	↑ ER stress, ↑ Apoptosis	↑ Bip ↑ p-eIF2α ↓ PERK, ATF4 ↑ IRE1α ↑ Caspases 3 and 7 ↑TfR	[113]
	100 µM (with 20 µM csplatin or 20 µM oxaliplatin)	↓ Cell viability, ↑ Autophagy	↑ LC3II, ↓ IP3K	[113]
HCT-8	35 µg/mL	↑ Cell cycle arrest G2/M		[112]
HCT116	12.5 µM	↑ Radiosensitivity, ↑ Autophagy and Apoptosis	↑ Nrf2, ↑ LC3, ↑ Caspase-9	[114]
	50–100 µM	↓ Cell proliferation		[115]
	50–100 µM	↑ Apoptosis	↓ VEGFR2, ↓ AKT, ↓ tumor growth, ↓ proliferation, ↓ migration and ↓ angiogenesis	[115]
	10–30 µM	↓ Cell growth, ↑ Cell cycle arrest (G2/M), ↓ Proliferation, ↓ Cell invasion, ↓ Cell adhesion, ↑ Apoptosis	↓ MMP2 and 9, ↑ caspases 3, 8 and 9, ↓ EGFR and IGF1R, ↓ MEK and ERK, ↓ PI3K and AKT, ↓ Bad	[111]
DLD-1,	100 µM with (20 µM cisplatin or 20 µM oxaliplatin)	↓ Cell viability, ↑ Autophagy	↑ LC3II, ↓ IP3K	[116]
	10 ng/mL	↓ Cell proliferation, ↑ Apoptosis	↑ CD133, CD44, ALDHA1, Oct-4, and Nanog, ↓ p-GSK3β, ↑GSK3β, ↓ Wnt, ↓ β-catenin, ↓ Cyclin D1, ↓ PCNA, ↓ Bcl2, ↑Bax, ↑ caspases 3, 8 and 9	[110]
RKO	50–100 µM	↑ Apoptosis	↑ p38	[115]
Caco-2	10–30 µM	↓ Cell growth, ↑ Cell cycle arrest G2/M, ↓ Proliferation, ↓ Cell invasion, ↓ Cell adhesion, ↑ Apoptosis	↓ MMP2 and 9, ↑ caspases 3, 8 and 9, ↓ EGFR and IGF1R, ↓ MEK and ERK, ↓ PI3K and AKT, ↓ Bad	[111]
Animal Models				
Male ICR mice	0.1% with (AOM 10 mg/kg body weight I.P followed by 2% (*w*/*v*) DSS)	↓ Weight, ↓ Inflammation	↓ COX2, ↓ mRNA (TNFα, IFN δ, IL6, IL12, IL18)	[105]
Eighty SPF Wistar rats	200 mg/kg with (DMH 40 mg/kg, s.c)	↓ Tumor volume, ↑ Apoptosis	↓ p53, PI3K-Akt, ↓ I-kappaB kinase/NF-kappaB, ↓ MAPK	[117]

**Table 7 molecules-26-03483-t007:** Antitumoral effects of quercetin in colorectal cancer.

		QUERCETIN		
Experimental Models	Concentration Range	Biological Response	Pathway/Genes/Proteins Involved	Refs.
Cells and cell lines				
HT-29	25, 50, 100 µM	Apoptosis	↓ Bcl-2, ↑ cleaved caspase-3, ↑ cleaved PARP, ↓ p-Akt, ↓ ErbB2/ErbB3 proteins	[117]
	5–30 µg/mL (with resveratrol, 1:1 ratio)	↓ Oncogenic µmicroRNA-27a	↓ Sp1, ↓ Sp3, ↓ Sp4, ↓ survivin mRNA and proteins	[118]
	50, 100, 200 µM	Apoptosis S-phase arrest	↓ p-Akt, ↓ CSN6, ↓ Myc, ↓ Bcl-2, ↑ p53, ↑ Bax proteins	[119]
	50 μM (with cisplatin: 10 mg/L)	↑ Cisplatin-induced Apoptosis	↓ Activation of NF-κB protein expression	[120]
	30 µM	↑ TRAIL-induced Apoptosis	Redistribution of death receptors DR4 and DR5 into lipid rafts ↑ cleaved caspase-3 and ↑ cleaved Bid proteins,↑ release Cyt-C	[121]
	50, 100 μM	Apoptosis G1-phase arrest	↑ AMPK, ↑ p53, ↑ p21 proteins	[122]
	50 µM (with dox: 250 nM)	↑ Doxorubicin-induced cytotoxicity	↓ Proliferation, ↑ apoptosis, and G2/M arrest for lower IC50 of Dox	[123]
Caco-2	5–50 µM	↓ Cell proliferation	↓ CDC6, ↓ CDK4, ↓ cyclin D1 mRNA	[124]
	5–20 μM	Anti-migration Anti-invasion	↓ MMP-2, ↓ MMP-9, ↓ TLR4, ↓ NF-κB, ↑ E-cadherin proteins ↓ TNF-α, ↓ COX-2, ↓ IL-6 production	[125]
Caco-2 and SW-620	25–100 µM	Apoptosis	↑ IκB-α, ↓ p-IκB-α, ↓ Bcl-2, ↑ Bax proteins	[126]
SW480	20–80 µM	Apoptosis	↓ Cyclin D1, ↓ survivin mRNA, and proteins	[127]
	10 µM	Apoptosis S-phase arrest	↓ EGF receptor phosphorylation	[128]
Colo-320 and Colo-741	25 μg/mL	Apoptosis Senescence	↑ p16, ↑ Lamin B1, ↑ cyclin B1, ↑ Bax, ↓ Bcl-2 proteins	[129]
CT26 and MC38	1–10 µM	Anti-metastasis	↑ E-cadherin, ↓ N-cadherin, ↓ β-catenin, ↓ snail proteins ↓ MMP-2, ↓ MMP-9 activities	[130]
DLD-1	10.5 µM	Anticarcinogenesis	↓ COX-2 transcription	[131]
CO115 and HCT15	12 µM (with 5-FU: 1 µM)	↑ Fluorouracil-induced apoptosis	↑ p53, ↑ cleaved caspase-9, ↑ cleaved caspase-3, ↑ cleaved PARP, ↓ Bcl-2 proteins	[132]
HCT8-β8	50 µM	↓ Cell proliferation	↑ ERβ mRNA, ↑ ER-responsive luciferase activity	[133]
Animal models				
HT-29 xenograft in Balb/C nude mice	10 mg/kg/day (SC; 4 weeks)	↑ Radiosensitivity	↓ Jagged-1, ↓ Notch-1, ↓ Hes-1, ↓ Presenilin 1, ↓ Nicastrin proteins	[134]

**Table 8 molecules-26-03483-t008:** Antitumoral effects of rutin in colorectal cancer.

RUTIN				
Experimental Models	Concentration Range	Biological Response	Pathway/Genes/Proteins Involved	Refs.
Cells and cell lines				
HT-29	100–200 µM	Apoptosis	↑ Bax, ↓ Bcl-2, ↑ cleaved caspases-3, 8, 9, ↑ cleaved PARP proteins	[139]
	39 mM (with Silibinin: 76 mM)	Apoptosis	↑ p53, ↓ Bcl-2, ↑ Bax, ↑ caspase 3, 8, 9 ↓ NFkB, ↓ IKK-α, ↓ IKK-β, ↑ p38MAPK, ↑ MK-2 proteins	[140]
HT-29 and Caco-2	25–200 µM	↓ Cell adhesion and Migration	↓ ROS level, impairing attachment to fibronectin, disrupting cell–ECM interactions	[141]
	136 µM	↓ Cell proliferation	↓ Growth potency	[142]
Animal models				
SW480 xenograft in nude mice	1–20 mg/kg/day (I.P; 32 days)	Anti-tumor Anti-angiogenesis	↑ Mean survival time, ↓ tumor volume, and weight, ↓ VEGF levels in serum	[143]
MTX-treated Wistar rats (Intestinal inflammation)	50, 100 mg/kg/day (I.P; 1 week)	↓ Oxidative stress ↓ Inflammation	↓ COX-1, ↓ COX-2, and ↓ 15 LOX enzymatic activities, restoration of MDA, protein carbonyl, SOD, GSH levels, and catalase activity, ↓ free acidity and total acidity	[144]
5-FU-treated Swiss mice (Intestinal Mucositis)	50–200 mg/kg/day (Per os; 3 days)	↓ Oxidative stress ↓ Inflammation	↓ MDA, ↑ GSH concentrations, ↓ MPO activity, ↓ intestinal mastocytosis, ↓ COX-2 proteins	[145]
DSS-treated ICR mice (Colitis)	0.6–6 mg/day (Per os; 2 weeks)	↓ Inflammation	↓ IL-1β, ↓ IL-6, ↓ GM-CSF, ↓ iNOS mRNA	[146]

## Data Availability

No new data were created or analyzed in this study. Data sharing is not applicable to this article.

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
