# Peer review of "Polyphenols of the Mediterranean Diet and Their Metabolites in the Prevention of Colorectal Cancer"

_molecules, 2021, doi:10.3390/molecules26123483_

Round 1

Reviewer 1 Report

Abstract:

  1. Line 18: ranking -Spell check
  2. Line 20: epigallocatechin gallate , -Remove space
  3. Line 20: quercetin-rutine, and resveratrol -Add comma
  4. Line 22: the plasma level, and the importance of other factors -Add comma
  5. Line 24: The emergence of -Add space
  6. Line 25: strenghen strengthen -Spell check
  7. Line 25: the Mediterranean -Add article

**Please check the manuscript throughout for these checks**.

Introduction:

  1. Line 68: the with consumption of -Add article
  2. Line 71: olive oil, and  -Add comma
  3. Line 77: for a daily consumption -Remove
  4. Line 81: diet is obviously associated -Remove
  5. Line 83: Mediterranean -CAPS
  6. Line 86: South Africa, and Australia (south and west) -Add comma
  7. Line 102: rutin, and glucoside -Add comma
  8. Line 103: rapsberries ---raspberries -Spell check
  9. Line 104: tomatoes, and celery -Add comma
  10. Line 111: Mediterranean -CAPS
  11. Line116/117: pro-differentiation effects -Change
  12. Line 128: a daily exercise -Change
  13. Line 134: adherence to the Mediterranean diet -Add article
  14. Line 136: the Mediterranean and north European -Add article
  15. Line 137: middle- aged populations -Remove space
  16. Line 140: Besides the anti-cancer activity ***of the Mediterranean diet -Change
  17. Line 148: on health status -Change
  18. Line 153: 46% less sensitive to declines -Change
  19. Line 164: (polyphenols, etc.), and possibly toxins -Add comma
  20. Line 171: CRC accounts -Change
  21. Line 187: epigenetics, and intestinal microbiota -Add comma
  22. Line 197: fruits, and vegetables -Add comma

Please go through the entire manuscript 3-4 times and use these changes as template to update the manuscript. Once you update please resubmit for second round

Also, change figures 3 and 7. The quality of structures is poor. Use the attached figures.

Check spell of Alantalactone (Alantolactone?).

Author Response

Covering letter 21 may 2021

Journal Molecules (ISSN 1420-3049)

Manuscript ID molecules-1210235

Polyphenols of the Mediterranean diet and their metabolites in the prevention of intestinal cancer

Aline Yammine , Amira Namsi , Dominique Vervandier-Fasseur , John J. Mackrill , Gérard Lizard , Norbert Latruffe *

Author's Reply to the Review Report (Reviewer 2)

Please provide a point-by-point response to the reviewer’s comments and either enter it in the box below or upload it as a Word/PDF file. Please write down "Please see the attachment." in the box if you only upload an attachment. An example can be found here.

Dear editor.

Thank you for your indications and for the comments sent  by the two revie-wers.

We submit a revised version which take into account the comments of the reviewers.
We hope our answers and modifications  shall allow the next axxeptance of the paper.

Sincerely yours

Norbert Latruffe, corresponding author

Answer to Report 1 comments

Dear reviewer.

Thank you for your carefull reading

English language and style. Extensive editing of English language and style required . The revised paper has been  checked again by John Mackrill Irish citizen 

Abstract: We have done the modifications according  all of your remarks

  1. Line 18: ranking -Spell check done
  2. Line 20: epigallocatechin gallate , -Remove space Done
  3. Line 20: quercetin-rutine, and resveratrol -Add comma OK
  4. Line 22: the plasma level, and the importance of other factors -Add OK comma
  5. Line 24: The emergence of -Add space OK
  6. Line 25: strenghen strengthen -Spell check done
  7. Line 25: the Mediterranean -Add article OK

**Please check the manuscript throughout for these checks**.

Introduction:

  1. Line 68: the with consumption of -Add article OK
  2. Line 71: olive oil, and  -Add comma OK
  3. Line 77: for adaily consumption –Remove OK
  4. Line 81: diet is obviouslyassociated –Remove OK
  5. Line 83: Mediterranean –CAPS ?? OK
  6. Line 86: South Africa, and Australia (south and west) -Add comma OK
  7. Line 102: rutin, and glucoside -Add comma OK
  8. Line 103: rapsberries ---raspberries -Spell check OK
  9. Line 104: tomatoes, and celery -Add comma OK
  • Line 111: Mediterranean –CAPS ?? OK
  • Line116/117: pro-differentiation effects –Change OK
  • Line 128: a daily exercise –Change OK
  • Line 134: adherence to the Mediterranean diet -Add article OK
  • Line 136: the Mediterranean and north European -Add article OK
  • Line 137: middle- aged populations -Remove space OK
  • Line 140: Besides the anti-cancer activity ***of the Mediterranean diet –Change OK
  • Line 148: on health status –Change OK
  • Line 153: 46% less sensitive to declines –Change OK
  • Line 164: (polyphenols, etc.), and possibly toxins -Add comma OK
  • Line 171: CRC accounts –Change OK
  • Line 187: epigenetics, and intestinal microbiota -Add comma OK
  • Line 197: fruits, and vegetables -Add comma OK

Please go through the entire manuscript 3-4 times and use these changes as template to update the manuscript. Once you update please resubmit for second round. Thank you for your recommendations

Also, change figures 3 and 7. The quality of structures is poor. Use the attached figures. In fact the pdf version is a reduced size version with a poor resolution. The docx version shows a much better resolution.

Check spell of Alantalactone (Alantolactone?). Thank you

Submission Date

20 April 2021

Date of this review

30 Apr 2021 21:12:17

Reviewer 2 Report

Manuscript by a. Yammine et al. Entitled „ Polyphenols of the Mediterranean diet and their metabolites in  the prevention of intestinal cancer”.

Major issues:

Manuscript has a lot of data and facts. Presently it is long collection of facts that are not fluently combined. Manuscript needs to be shortened and better combined to avoid unnecessary repeating of facts.  Use figures to present facts. A lot of text is used for presenting general facts and subject of manuscript is somehow lost and as it is it is more like textbook .  Manuscript  needs to be more focused on title “… polyphenol and their metabolites in ….intestinal cancer…”.  

 Minor issues.

1.-english language needs to be improved.:

2.-spelling mistakes line 153,145,260  etc.

3.-LINE 200-USE ACTIVE FORM OF WRITING- NO NEED TO ANONCE WHAT WILL BE DISCUSSED

4- lines 227,282.265 etc. :In vitro and in vivio-use IITALIC STYLE :

5-chapter 5- no need to separate 5.1 and 5.2. since the rei a lot of unnecessary repeating of fact.

6Figure4.-incorporate resveratrol derivatives as documented in text (line 405-433)

7-Figure 5. not informative – new figure showing the effect of microbiota on specific polyphenols : apigenin line 493; curcumin line 507, EGCG line 520

8.Page 17. Shorten the text 415-484

9.Table5.  and Table6.-low resolution

10.table 7: table not positioned well left side is partially missing

  1. Line 680-685- please clarify the statement- not understandable as it is written

12.Titel 6.2. – Title shod be more defined  eg. “Pro-differentiating effect of polyphenols

Author Response

Covering letter 21 may 2021

Journal Molecules (ISSN 1420-3049)

Manuscript ID molecules-1210235

Polyphenols of the Mediterranean diet and their metabolites in the prevention of intestinal cancer

Aline Yammine , Amira Namsi , Dominique Vervandier-Fasseur , John J. Mackrill , Gérard Lizard , Norbert Latruffe *

Author's Reply to the Review Report (Reviewer 2)

Please provide a point-by-point response to the reviewer’s comments and either enter it in the box below or upload it as a Word/PDF file. Please write down "Please see the attachment." in the box if you only upload an attachment. An example can be found here.

Dear editor.

Thank you for your indications and for the comments sent  by the two revie-wers.

We submit a revised version which take into account the comments of the reviewers.
We hope our answers and modifications  shall allow the next axxeptance of the paper.

Sincerely yours

Norbert Latruffe, corresponding author

Answer to Report 2 comments

Dear reviewer.

Thank you for your carefull reading and suggestions. Please find below our changes link to your comments

We hope our answers and modifications  shall agree with you

English language and style. Extensive editing of English language and style required . The revised paper has been  checked again by John Mackrill Irish citizen 
Is the English used correct and readable?

Major issues:

Manuscript has a lot of data and facts. Presently it is long collection of facts that are not fluently combined. Manuscript needs to be shortened and better combined to avoid unnecessary repeating of facts.  Use figures to present facts. A lot of text is used for presenting general facts and subject of manuscript is somehow lost and as it is it is more like textbook .  Manuscript  needs to be more focused on title “… polyphenol and their metabolites in ….intestinal cancer…”.  

Following your comments in order to short the lenght  and to be more clear, we changed as follow :

  • Figure ex #1 has been withdrawn and information are still in the text. This figure now is proposed as grapical abstract
  • Basis of intestinal cancer has been withdrawn because not directly link to the topic, but figure ex #2 (now figure 1) is maintained and mentioned in  the text.
  • Figure ex # 5 (now figure 4) on microbiota has been completd
  • The texts for presentation of the 6 selected polyphenols were withdrawn. Only table 3 is maintained since it contain enough information of the polyphenols. Conversely, since, the antitumor properties of each of the selected polyphenols are directly link to the topic are maintained.
  • The pro-differentiating effects of polyphenols  have been withdrawn. Indeed, we consider this chapter not enough related to intestine cancer.

 Minor issues.

1.-english language needs to be improved.: see above

2.-spelling mistakes line 153,145,260  etc. Done

3.-LINE 200-USE ACTIVE FORM OF WRITING- NO NEED TO ANONCE WHAT WILL BE DISCUSSED Done

4- lines 227,282.265 etc. :In vitro and in vivio-use IITALIC STYLE : Done

5-chapter 5- no need to separate 5.1 and 5.2. since the rei a lot of unnecessary repeating of fact. OK 5.1 & 5.2 have been fused

6Figure4.-incorporate resveratrol derivatives as documented in text (line 405-433). In fact  glucurono- and sulfo-derivatives were  represented in the figure but not mentioned in the legend. Now this is done

7-Figure 5. not informative – new figure showing the effect of microbiota on specific polyphenols : apigenin line 493; curcumin line 507, EGCG line 520. A new figure 4 is now provided

8.Page 17. Shorten the text 415-484 This section has been re-written

9.Table5.  and Table6.-low resolution. In fact the pdf version is a reduced size version with a poor resolution. The docx version shows a much better resolution

10.table 7: table not positioned well left side is partially missing. This is now repaired

  • Line 680-685- please clarify the statement- not understandable as it is written. This section has been re-written

12.Titel 6.2. – Title shod be more defined  eg. “Pro-differentiating effect of polyphenols. This section has been withdrawn to short the lenght

Submission Date

20 April 2021

Date of this review

05 May 2021 11:58:03

Round 2

Reviewer 1 Report

Well written manuscript.

Author Response

Assigned Editor Ryan Siu

Journal Molecules

Manuscript Status Pending minor revisions

Manuscript ID molecules-1210235

Academic Editor Notes

This manuscript analyze the impact of six of the most common polyphenols aboundant in fruits and vegetables consumed in the Mediterranian diet. A lot of data from literature is presented regarding the metabolism, microbiota metabolism, anticancer activity and some other biological activities, and nanoformulation of compounds. With respect to the title, a deeper focus on intestinal cancer should be maintained in the text. Some other data on different types of tumors are presented, while no data on other type of intestinal cancers (non-colorectal cancer, other type of intestinal tumors, like GIST even if less prevalent than the former). In this case I suggest the authors to change the title referring to cancer in general or mentioning "colorectal" cancer. Some minor specific comments are reported below.

Dijon june 3rd 2021

Dear Editor

Thank you for your carefull reading

Please see below the corrections we made according to the editor comments

Related to this we submit a new revised version. Hoping this suitable for publication

Sincerely yours

  1. Latruffe on behalf of the co-authors

line 21 separate epigallocatechin gallate; done
line 84 better to say islands like Sardinia and Crete ; done
References 1 and 2 are ok, but too much limited to France, I suggest to introduce other literature references if available;see web site
I suggest to report the Tables in the same style, if not retrieved from published articles, in this case a reference must be indicated in the caption; done
line 101 "focusing" while in the abstract "focussing" use the same spelling; corrected
lines 124-126 too much example are referred to France, but in this case maybe the data were available; indicate the source of this data as a reference (if retrivied from a website or document); this part has been modified
Figures 1 and 2 appears at low resolution; increase the resolution , perhaps fig.2 may be made again with chemdraw; re-worked
line 173 Figure 2A, remove the space between 2 and A; withdrawn see below
Fig. 2B is a table and should be presented as it is, apart from fig 2. Names of compounds could be indicated in the caption of this figure; new table
line 191 in combination "WITH" instead "OF" in silico... thanks
The resolution of all figures must be improved; we did as  much as we could
line 348 explain what kind of inhibition: growth? metabolites production? Sentence corrected

Language Editing

If the reviewers or editor recommended English language editing, this can be arranged by MDPI. Note that language editing by MDPI is not compulsory, nor does it guarantee that your manuscript will eventually be accepted for publication. Click on the link for more information and to request a quotation.

More information on English editing from MDPI.

English checked

Commentaire 1 change done. Thanks

Commentaire 2 accuracy confirmed

Commentaire 3 correct

Commentaire 4 done

Commentaire 5  corrected

Commentaire 6 web site is provided

Commentaire 7 N. Latruffe is the co-author and the editor of the book

Commentaire 8 see  answer to commentaire 7

Commentaire  9 we did as much as possible.

Commentaire 10  sentence changed. Thanks

commentaire 11 id°  answer to commentaire 7

Commentaire 12 yes

Commentaire 13 we reformulated the sentence

Commentaire 14 we reformulated the sentence

Commentaire 15 we did as much as we could

Commentaire 16 the figure 1B is a change of a figure  proposed in a  PhD student dissertation under the supervision of N.Latruffe, the corresponding author

Commentaire 17 figure 2A become figure 2 and former figure 2B becomes new table 3 with references in a new column right

Commentaire 18 figure has been reconsidered

Commentaire  19 figure has been reconsidered accordingly

Commentaire 20 done

commentaire 21 figure  has been re-drawn

Commentaire 22 done

Commentaire 23 done

Commentaire 24 it is a pure creation of the authors

Commentaire 25 change made

Commentaire 26 sentence corrected

Commentaire 27 we did as much as we could.

Commentaire 28 corrected

Commentaire  29 we did as much as we could

Commentaire 30 we dis as much as we could

Commentaire 31 we did as much as we could

Commentaire 32 done

Commentaire 33 we did as much as we could

Commentaire 34 ref 131 to 135 are now introduced

Commentaire 35 we did as much as we could

General English checked again

Review

Polyphenols of the Mediterranean Diet and Their Metabolites in the Prevention of Colorectal Cancer

Aline Yammine 1,†, Amira Namsi 1,†, Dominique Vervandier-Fasseur 2, John J. Mackrill 3, Gérard Lizard 1,‡ and Norbert Latruffe 1,*

Citation: Yammine, A.; Namsi, A.; Vervandier-Fasseur, D.; Mackrill, J.J.; Lizard, G.; Latruffe, N. Polyphenols of the Mediterranean Diet and Their Metabolites in the Prevention of Intestinal Cancer. Molecules 2021, 26, x. https://doi.org/10.3390/xxxxx

Academic Editor: Roberto Fabiani

Received: 20 April 2021

Accepted: date

Published: date

Publisher’s Note: MDPI stays neutral with regard to jurisdictional claims in published maps and institutional affiliations.

Copyright: © 2021 by the authors. Submitted for possible open access publication under the terms and conditions of the Creative Commons Attribution (CC BY) license (http://creativecommons.org/licenses/by/4.0/).

1    Team Bio-peroxIL, “Biochemistry of the Peroxisome, Inflammation and Lipid Metabolism” (EA7270), University of Bourgogne Franche-Comté, Inserm, 21000 Dijon, France; [email protected] (A.Y.); [email protected] (A.N.); [email protected] (G.L.)

2    Team OCS, Institute of Molecular Chemistry of University of Burgundy (ICMUB UMR CNRS 6302), University of Bourgogne Franche-Comté, 21000 Dijon, France; [email protected]

3    Department of Physiology, University College Cork, BioScience Institute, College Road, T12 YT20 Cork, Ireland; [email protected]

*    Correspondence: [email protected]; Tel.:+333-80-39-62-37; Fax:+333-80-39-62-50

†   These authors contributed equally to this work.

‡   Inserm Researcher; Researcher attached to the international network of the UNESCO Chair ‘Culture and tradition of wine’ (http://chaireunesco-vinetculture.u-bourgogne.fr/)

Abstract: The Mediterranean diet is a central element of a healthy lifestyle, where polyphenols play a key role due to their anti- oxidant properties, and for some of them, as nutripharmacological compounds capable of preventing a number of diseases, including cancer. Due to the high prevalence of intestinal cancer (ranking second in causing morbidity and mortality), this review is focussed on the beneficial effects of selected dietary phytophenols, largely present in Mediterranean cooking: apigenin, curcumin, epigallocatechin gallate, quercetin-rutine, and resveratrol. The role of the Mediterranean diet in the prevention of colorectal cancer and future perspectives are discussed in terms of food polyphenol content, the effectiveness, the plasma level, and the importance of other factors, such as the polyphenol metabolites and the influence of the microbiome. Perspectives are discussed in terms of microbiome-dependency of the brain-second brain axis. The emergence of polyphenol formulations may strengthen the efficiency of the Mediterranean diet in the prevention of cancer.

Keywords: intestinal cancer; dietary polyphenols; Mediterranean diet; resveratrol; curcumin; quercetin; rutin; apigenin; EGCG; microbiota; polyphenol nanoformulation

1. Introduction

The Mediterranean diet is associated with the Mediterranean climate, which is temperate and characterized by hot and dry summers, and mild and humid winters. However, interestingly, the Mediterranean climate and diet are not restricted to Mediterranean Sea area but are also found in similar regions of the world located between 30 and 40° of latitude. For instance, the western coasts of continents like America (California, mid-Chili), the Cape region in South Africa, and Australia (south and west). In all of these countries, food and beverages are quite comparable and provide good health to the populations as illustrated by the exceptional longevity of people living on Sardinia and Crete, and those of the Okinawa population. The Mediterranean diet has beneficial effects on health thanks to foods rich in polyphenols; the most abundant phytochemicals in the plant kingdom present in fruit, vegetables, wine, and honey (Table 1) [1].

Table 1. Total polyphenols content of main edible plants from Mediterranean diet. Adapted from Debbabi et al., [2]

Fruits

Mean of total polyphenol content (mg of GAE/100g fresh edible portion)

Vegetables

Mean of total polyphenol content
(mg of GAE/100g fresh edible portion)

Strawberry

263.8

Artichoke heart

321.3

Lychee

222.3

Parsley

280.2

Grape

195.5

Brussels sprout

257.1

Apricot

179.8

Shallot

104.1

Apple

1179.1

Broccoli

98.9

Date

99.3

Celery

84.7

Cherry

94.3

Onion

79.1

Fig

92.5

Eggplant

65.6

Pear

69.2

Garlic

59.4

White nectarine

72.7

Turnip

54.7

Passion fruit

71.8

Celeriac

39.8

Mango

68.1

Radish

38.4

Yellow / white peach

59.3 – 44.2

Pea

36.7

Banana

51.5

Leek

32.7

Pineapple

47.2

Red bell pepper

26.8

Lemon

45

Cherry tomato

26.6

Grape fruit

43.5

Potato

23.1

Orange

31

Zucchini

18.8

Clementine

30.6

Green bell pepper

18.2

Lime

30.6

Chicory

14.7

Kiwi

28.1

Asparagus

14.5

Watermelon

11.6

Tomato

13.7

melon

7.8

Fennel

13

Cauliflower

12.5

Carrot

10.1

French string bean

10

Avocado

3.6

The Mediterranean diet has age-cultural and nutritional aspects that contribute to wellbeing. In 2010, the UNESCO world organisation added the Mediterranean diet to the list of intangible cultural heritage, where cooking is associated with the consumption of beverages (wine and/or infusions/tea) with respect to the beliefs and traditions of each community. Mediterranean cooking is becoming more and more popular, providing food microcomponents including polyphenols, vitamins, fiber, vegetables, olive oil, fishes rich in polyunaturated fatty (ω3), and oligo-elements supplied by fruits and beverages. Besides fruits and vegetables, red wine provides additional unique polyphenols with strong anti-oxidant properties. These include resveratrol, procyanidines, and monophenols, including hydroxytyrosol and tyrosol. The Mediterranean diet is represented by a food pyramid proposed by Willett W.C. et al. [3] based on food groups, consumed weekly or less frequently.

There are about 8000 different dietary polyphenols, divided into two major classes: i) Flavonoids which include other subclasses based on their chemical structure such as (flavanols, isoflavones, flavanones, flavonones, and anthocyanidins) and ii) non- flavonoids divided on different groups (phenolic acids, stilbenes, and lignans; tannins are flavonoid polymers [4]. The Table 2 illustrates the polyphenol profiles of some fruits and vegetables of the Mediterranean diet that have important anti-oxidant potential and a high polyphenol content [2].

Table 2. Fruit and vegetables with high polyphenols content characterized by important anti-oxidant activity. Adapted from Debbabi et al. [2].

Extracts                                                           

Polyphenols expressed in (mg / 100 g of fresh matter)

Muscat  grape

Gallic acid (1.7), epigallocatechin (10.8), catechin (21.8), epicatechin (5.3), quercetin-3-O-beta-D-glucuronide (10.8)

White grape

Gallic acid (1.1), caftaric acid (10.8), epigallocatechin (4.4), catechin (10.6), epicatechin (6.0), quercetin-3-O-beta-D-glucuronide (1.7)

Strawberry

Epigallocatechin (6.3), catechin (7.1), epicatechin (4.1), epicatechin gallate (1.3), quercetin-3-O-beta-D-glucuronide (15.4), kaempferol-3-O-glucoside (2.6)

Raspberry

Rutin (1.6), quercetin-3-O-beta-D-glucuronide (1.7)

Beans

Catechin (1.0), myricetin-3-O-glucoside (12.0), rutin (0.8), kaempferol-3-rutinoside (8.1)

Tomato

Chlorogenic acid (2.4), myricetin-3-O-glucoside (0.3), rutin (1.3), kaempferol-3-O-glucoside (0.1), quercetin (0.4)

Celery

 Chlorogenic acid (75.3), quercetin-3,4’-diglucoside (2.3), rutin (368.9), quercetin-3-beta-glucoside (5.7), kaempferol-3-rutinoside (22.1)

Radish

Catechin (2.7)

We can observe that quercetin, rutin, and glucoside derivatives are from the major polyphenols present in grapes, strawberries, raspberries, beans, tomatoes, and celery. There is considerable evidence from in vitro studies, animal models, and clinical studies supporting the concept that several polyphenols have powerful anti-oxidant [5] and anti-tumor properties (cell cycle delay, apoptosis induction, metastasis prevention) [6].

This review will concern: a) the benefits of the Mediterannean diet based on cohorts and case-control studies for prevention of cancer, cardiovascular risk and cognition, b) polyphenols as major disease-preventing components and analyses of the roles of these molecules, c) the metabolism of polyphenols and the importance of the microbiome in generating active compounds, d) the intestine antitumor efficiency and the mechanisms of these polyphenols and the use of nanoformulations, e) the future developments of the Mediterranean diet, which is growing over the world, and the interest in nanoformulation supplements based on this diet.

2. Benefits of the Mediterannean Diet in Preventing or Treating Cancer

Taking France as an example, since 2004 cancer (152,708 death) is ahead of cardiovascular disease (147,323 death) as the leading cause of mortality. Interestingly, numerous works illustrate the benefits of the Mediterranean diet in preventing cancer and other important illnesses.

In cancer, the analysis of an Italian cohort of breast cancer suggests that the adoption of a strict Mediterranean diet is associated to a decrease of 30–50% of the incidence of its development. In addition, relapse and mortality in breast cancer is decreased in patients that have a daily exercise of at least 30 min/day [7]. A case-control study performed in Catania, Italy, on 338 patients diagnosed with colorectal cancer (CRC) compared to 676 subjects with no apparent clinical syndrome, showed a significant association between the adoption of a strict Mediterranean diet and a decrease of the relative risk of developing CRC, with an odds ratio of 0.46 [8]. In head and neck cancer, analysis of a sample of 168 subjects (100 controls and 68 cases) showed that medium-high adherence to the Mediterranean diet was associated with a lower risk of developing head and neck cancer (OR, 0.48 [95% CI: 0.20–1.07], P = 0.052) [9]. Very recently, Melander et al., [10] studied the prevalence of cardiovascular disease and cancer in Mediterranean and the north European middle-aged populations. They showed the odds ratio for cancer in Malmö (Sweden) versus Cilento (Italy) was 2.78 (1.81–4.27) (P < 0.001) in inhabitants adjusted for age and sex.

Besides the anti-cancer activity of the Mediterannean diet, other benefits have been reported. A meta-analysis on 18 studies (cohort of 4 172 412 subjects), demonstrated that consumption of the Mediterranean diet led to an 8% reduction in general mortality and a 10% decrease in the risk of development of cardiovascular illness [11]. Knowing that decreased telomere length at the chromosome ends and increased shortening rate of telomeres are aging biomarkers, a study conducted on 217 elderly reported an association between leukocyte telomere length, telomerase activity as well as benefits on health status. In subjects with a strict adherence to the Mediterranean diet, the leucocyte chromosomes showed the greatest length of telomeres (P = 0.003) and the highest telomerase enzyme activity (P = 0.013) [12]. In addition, an Irish study was conducted on 2,621 subjects of an average of 25 years old at the inclusion and who were followed for 30 years, monitoring their food habits. Subjects with good adherence to the Mediterranean diet were 46% less sensitive to declines in their cognitive capacities, particularly their thought capacities. For 868 participants with strong adherence, 9% showed low thought capacities while this frequency was 29% in 798 participants with low adherence to Mediterranean diet [13]. Recently, in relation to the current pandemic situation of COVID-19, flavan-3-ols and dimeric proanthocyanidins have been reported by Zhu et al., to prevent SARS-cov2 infection [14].

In this review, a focus on intestinal cancers has been chosen for two reasons. In France, in 2018, CRC (Figure 1) was the second cause of mortality (17,117) and morbidity (43,336) after breast cancer and lung cancer (men). The intestinal tract is directly exposed to and is the primary absorption/metabolism site for dietary compounds which can be either basic nutrients for energy supply and growth, micronutrients (polyphenols, etc.), and toxins.

Figure 1. (A) Polyps as precancerous intestinal cancer, (B) colonic carcinogenesis stages. Adapted from the PhD Thesis of Dr F. Mazué (Director of Thesis; Prof. N. Latruffe; https://www.theses.fr/165862416)

3. Key Selected Polyphenols

From several thousand different polyphenols, we have focussed on species present in significant quantities in the plants, fruits and vegetables present in the Mediterranean diet i.e., apigenin, curcumin, EGCG (Epi Gallo Catechin Gallate), quercetin, resveratrol and rutin (Figure 2). Interesting antitumor properties of these compounds have been reported in the literature [15–17]. We firstly report the general properties of these polyphenols, their origin, abundance, physico-chemical and biological properties (Table 3). We will also present the anticancer properties of these selected polyphenols.

Figure 2. chemical structures of selected polyphenols: 1: Curcumin; 2: Quercetin;

3: Apigenin; 4: Resveratrol; 5: Rutin6: EGCG

Table 3: main characteristics of selected polyphenols

Polyphenol

Family

Mol.Formula

Mol. Weight

Water solubility (mg/L)

Main dietary and geographic sources

Ref.

1: Curcumin

Curcuminoid

C21H20O6

368.35 g/mol

0.125 mg/L

Rhizomes of Curcuma longa(turmeric) India

[18,19]

2: Quercetin

Flavonoid

C15H10O7

302.20 g/mol

0.48 mg/L

(MilliQ water)

Red grape, onion, broccoli, tomato, lettuce

[20-23]

3: Apigenin

Flavonoid

C15H10O5

270.21 g/mol

1.43 mg/L

(pH = 1)

Chamomile (Europe, Western Asia)

[24,25]

4: Resveratrol

Stilbenoid

C14H12O3

228.24 g/mol

30 mg/L

Grapes and red wine,

Peanuts, blueberries, rhubarb

[26-30]

5: Rutin

Flavonoid

C27H30O16

610.44 g/mol

130 mg/L

Red grape, citrus, apple, fig,

asparagus, onion, mulberry, tea

[21,31 32]

6: EGCG

Flavonoid

C22H18O11

358.32 g/mol

5733.12 mg/L

Green tea from leaves and buds of Camellia sinensisJapan,

Chocolate, Red grape

[21,33-35]

4. Metabolism and Microbiota Metabolites

4.1. Metabolism

4.1.1. Apigenin

Like many polyphenols, apigenin is also present in nature in a glycosylated form and this flavone is at the centre of complex metabolic pathways. The apigenin glycoside, after deglycosidation in enterocytes, gives apigenin. In enterocytes, apigenin can then be sulfated, glucuronidated, dehydrogenated, hydrogenated and hydroxylated; the hydroxylated product can then be methylated and hydroxylated. Apigenin is also a major metabolite of naringin, a flavone present in very large quantities in grapefruit and pomelo. Metabolites profiles of flavonoids are usualy obtained with the use of UHPLC-QTOF-MS/MS in combination with in silico approach [36].

4.1.2. Curcumin

As with all polyphenols, after oral administration, the bioavailability of curcumin is low and the blood levels are reduced relative to those ingested. However, curcumin’s resistance to highly acidic pH is high and it reaches the large intestine without chemical and structural changes [37]. Curcumin is poorly absorbed from the gastrointestinal tract and is extensively metabolized in the enteric and hepatic tract. Glucuronidation of curcumin by beta-glucuronidase is a major metabolic pathway and the glucuronide form of curcumin is found in biological fluids as well as organs and cells of different species (rat, mouse, man). Curcumin can also be sulfated by sulfotransferase to give curcumin sulfate [38]. In hepatocytes and enterocytes, curcumin can also be reduced by various enzymes, including cytochrome P450 enzymes [39]. The products formed, dihydrocurcumin and tetrahydrocurcumin, can in turn be metabolised to glucuronide and sulfate derivatives [37]. Commercial products often contain demethoxycurcumin and bisdemethoxycurcumin, which undergo metabolism similar to that of curcumin [37].

4.1.3. EGCG

This polyphenol is essentially absorbed in the intestine and, like all polyphenols, the intestinal flora plays an important role in its metabolism before absorption, for additional biotransformation [40]. EGCG is metabolized by tissue extracts from the small and large intestine, whereas it is only slightly metabolized by tissue extracts from the liver. Since EGCG is poorly absorbed, blood concentrations of EGCG are low and its many derivatives are found in conjugated or unconjugated forms in plasma and urine [41].

4.1.4. Quercetin

This polyphenol is one of the most studied and its complex metabolism gives rise to several derivatives. Quercetin, which is essentially poorly absorbed in the small intestine, like all polyphenols, supplies metabolites that are produced by biotransformation by the intestinal microbiota and these are further absorbed in the large intestine. The resulting compounds are bioavailable, circulate in the blood as conjugates with glucuronide, methyl, or sulfate groups attached, and are eventually excreted in the urine. At the enterocyte level, several enzymes are involved in providing additional metabolites of quercetin: UDP- glucuronosyltransferase (UGT); sulfotransferase (SULT); catechol-O-methyl transferase (COMT); lactase phloridzin hydrolase (LPH); cytosolic β-glucosidase (CBG) [42]. The resulting derivatives of quercetin are found in plasma and urine [42].

4.1.5. Resveratrol

Resveratrol is efficiently absorbed on oral administration, then appears both in plasma, urine and to a lesser degree, in feces (Figure 3). Concerning its transport, Lançon et al., showed that human hepatic cell uptake of resveratrol involves both carrier mediated and diffusion-dependent process [43]. In plasma, resveratrol interacts with serum albumin [44]. Furthermore binding properties between resveratrol or other polyphenols and plasma albumin have been determined in the context of consequences for the health protecting effect of dietary plant microcomponents [45]. Recently, Tung et al., reported that polyphenols bind to LDL at biologically relevant concentrations that are protective against heart disease [46].

Figure 3. Metabolism of resveratrol. Adapted from Delmas et al., [47]. ER: estrogen-receptor; ABP: Albumin Binding Protein; LDL: Low Density Lipoprotein; LDL-R: Low Density Lipoprotein-receptor.  resveratrol;    glucurono resveratrol;    sulfo resveratrol.

Resveratrol metabolism leads to the production of sulfates, glucuronides derivatives as well as dihydroresveratrol (Figure 3). The nature and quantity of these metabolites may differ between subjects due to inter-individual variability. Once in the bloodstream, metabolites can be subjected to phase II metabolism, with further conversions occurring in the liver, where entero-hepatic transport in the bile may result in some recycling back to the small intestine. We have shown in hepatic cells, that resveratrol is highly conjugated after 4h of incubation, into mono- (3-sulfate-resveratrol and 4’-sulfate-resveratrol) and disulfate (3,4’-disulfate-resveratrol and 3,5-disulfate-resveratrol) derivatives [47]. Resveratrol can induce its own metabolism by increasing the activity of phase II hepatic detoxifying enzymes [48]. Resveratrol metabolites have a plasma half-life similar to resveratrol aglycone. Despite its low bioavailability, resveratrol in vivo efficacy can be explained by i) the back conversion of resveratrol sulfates and glucuronides to resveratrol in target organs such as liver; ii) the enterohepatic recirculation involving biliary secretion of resveratrol metabolites followed by deconjugation by gut microflora and then reabsorption; iii) the activities of its metabolites. Two sulfate metabolites show biological activity: resveratrol-4’-O-sulfate binds the cyclo-oxygenase (COX-1) and inhibits enzyme activity with nearly the same efficacy of resveratrol; resveratrol-3-O-sulfate shows a radical scavenger activity, inhibits COX-1, and is cytotoxic. In contrast to the resveratrol 3-O-sulfate, resveratrol-4’-O-sulfate metabolite is less active than resveratrol. Piceatannol (3,5,3’,4’-tetrahydroxystilbene), another analogue of the stilbene family, is a polyphenol found in grapes and other plants; it differs from resveratrol by an additional hydroxyl group at 3’ position of the benzene ring. In humans, piceatannol is produced as a major metabolite of resveratrol by cytochromes CYP1B1 and CYP1A. Piceatannol displays anticancer properties by inducing extrinsic and intrinsic pathways of apoptosis and by blocking cell cycle progression. Trans-resveratrol metabolism by human gut microbiota only generates one metabolite: dihydroresveratrol. The resveratrol derivatives, epsilon-viniferin and their acetates have been studied at the level of their cellular uptake and their antiproliferative effects [49].

4.2. Microbiota Metabolites.

Microbiomics is a discipline that studies bacterial and viral ecosystems. Microbiomic investigations [50] are promising for addressing the interactions between microorganisms and human cells at the level of organs in which these cells and microorganisms cohabit. At the level of the digestive tract, and the intestine in particular, there are long-lasting interactions between the intestinal flora and the nutrients ingested. At the level of the digestive tract, the activities of the intestinal cells will depend on several epigenetic factors including the microbiota present, as well as nutrients and micronutrients of food origin (Figure 4).

Figure 4. Contribution of the gut microbiota to polyphenol metabolism. The metabolites of polyphenols (curcumin, quercetin, apigenin, resveratrol, rutin and EGCG) produced by the gut microbiota are written in blue.

The proper functioning of the cells of the intestinal wall will therefore depend on their interactions with the bacterial ecosystem present and the molecules provided by the food. Food also influences the bacterial ecosystem and the latter can transform certain food molecules by metabolising them, which is particularly important in the case of polyphenols present in large quantities in the Mediterranean diet [51]. It is currently assumed that the intestinal microbiota of subjects consuming a Mediterranean diet, rich in polyphenols and ω3 fatty acids, is capable of preventing the appearance of chronic non-transmissible degenerative diseases, such as age-related diseases (cardiovascular diseases, neurodegeneration) and certain types of cancer [52]. For polyphenols, it is well established that understanding their activity is inseparable from understanding their interaction with the microbiota. This includes i) the biotransformation of polyphenols by the microbiota, which depends on the type of bacteria present, and ii) the ability of polyphenols to modulate the profile of the bacterial flora [53]. In humans, the ingestion of dietary polyphenols is associated with prebiotic activity (modification of the intestinal flora and its functions) which leads to the formation of several metabolites derived from polyphenols with a beneficial effect on lipid and carbohydrate metabolism (activation of lipogenesis and glycogenesis), which are deregulated in many diseases including cancers [54]. For phytoestrogens, which are a class of polyphenols with a diphenolic structure similar to 17β-estradiol, it has been shown that they can be transformed by the microbiota [55]. The same transformation process applies to certain isoflavones, prenylfavonoids, lignans and stilbenes such as reveratrol [55]. The metabolism of intestinal bacteria can also convert food lignans (a non-flavonoid polyphenolic class found in plant foods) into therapeutically relevant polyphenols (i.e., enterolignans) such as enterolactone and enterodiol [56]. Ellagic tannins (substances from the polyphenol family) from the pomegranate are transformed into urolithin A, a mitophagy-activating polyphenol, which increases the lifespan of the worm Caenorhabditis elegans, running endurance in rodents (rats, mice) and transcription of mitochondrial genes in human muscle tissue [57]. Several studies have also shown that polyphenols, by affecting the composition of the intestinal microbiota, modulate oxidative stress and immune response [58]. In agreement with these observations, red wine extracts administered orally reduced the growth of colorectal tumours (hot tumours: associated with inflammatory infiltrates) in mice by reducing the activation of T lymphocytes to Th17-positive cells [59].

The mutual interaction of polyphenols with the microbiota is therefore a determining factor in understanding their biological activities. The notion of metabolites generated by the intestinal microbiota must necessarily be taken into account in the case of oral administration. Apigenin, curcumin, EGCG and quercetin, which are polyphenols associated with the Mediterranean diet, are all subject to transformation by the microbiota.

Apigenin or its glycosides have been shown to be degraded by certain intestinal bacteria into smaller metabolites that can have biological activities [60]. Thus, the capacity of the human intestinal microbiota to convert apigenin-7-glucoside (A7G) has been proven in vitro by incubating A7G with faecal suspensions: apigenin, naringenin and 3-(4-hydroxyphenyl)propionic acid are formed as the main metabolites [61]. After application of A7G to germ-free rats, apigenin, luteolin and their conjugates were detected in urine and faeces. In human microbiota-associated (HMA) rats (comparatively to germ-free animals), naringenin, eriodictyol, phloretin, 3-(3,4-dihydroxyphenyl)propionic acid, 3-(4-hydroxyphenyl)propionic acid, 3-(3-hydroxyphenyl)propionic acid and 4-hydroxycinnamic acid were also formed in their free and conjugated forms. Furthermore, when apigenin is studied under culture conditions mimicking the ascending colon, it modifies the multiplication and gene expression of the Enterococcus caccae bacterium, whereas no effect has been observed on Bifidobacterium catenulatumand Lactobacillus rhamnosus G [62].

As for curcumin, which is a lipophilic polyphenol with low systemic bioavailability, it is largely biotransformed by human intestinal microflora to give different metabolites that are easily conjugated to glucuronides and O-conjugated sulfate derivatives [63]. It is also suspected that curcumin and its derivatives have direct regulatory effects on the intestinal microbiota. These findings may explain the paradox between the low systemic bioavailability of curcumin and its widely reported pharmacological activities [64].

Epigallocatechin-3-gallate (EGCG), one of the major phenolic compounds in green tea, has versatile bioactivity and exerts a strong anti-cancer effect. EGCG is mainly absorbed in the gut, and the intestinal microbiota plays an essential role in its metabolism prior to absorption [40]. The interaction of green tea phenolic compounds, particularly EGCG, with the intestinal microbiota may play a key role in the health benefits by generating beneficial bioactive compounds. When the metabolic fate of EGCG and its impact on the intestinal microbiota was studied through in vitro fermentation, it was shown by ultra-high performance liquid chromatography and Orbitrap quadrupole hybrid mass spectrometry (UHPLC- Q-Orbitrap-MS), that EGCG is rapidly degraded to a series of metabolites, including 4-phenylbutyric acid, 3-(3’,4’-dihydroxyphenyl)propionic acid and 3-(4’-hydroxyphenyl)propionic acid, by subsequent hydrolysis of the esters, opening of the C-ring, fission of the A-ring, dehydroxylation and shortening of the aliphatic chain. Microbiome profiling indicated that, compared to negative controls, treatment with EGCG resulted in the stimulation of the beneficial bacteria Bacteroides, Christensenellaceaeand Bifidobacterium, while the growth of pathogenic bacteria Fusobacterium varium, Bilophila and Enterobacteriaceaewas inhibited [65].

Like other polyphenols, quercetin, which is a plant polyphenol widely present in the Mediterranean diet, is metabolized by the microbiota and acts on the composition and activity of the intestinal flora. As with other polyphenols, it is crucial to better understand the impact of the microbiota on quercetin in order to better understand its effects on health [66]. Thus, quercetin is bioaccessible to the microbiota in the colon, which metabolizes it by producing phenylpropanoic acid, phenylacetic acid and benzoic acid derivatives [67]. In addition, the effects of quercetin have been documented on the intestinal commensal microbes Ruminococcus gauvreauii, Bifidobacterium catenulatum and Enterococcus caccae [68]. Quercetin does not inhibit the growth of Ruminococcus gauvreauii, it slightly suppresses the growth of Bifidobacterium catenulatum, and moderately inhibits the growth of Enterococcus caccae. Transcriptomic analysis revealed that in response to quercetin, Ruminococcus gauvreauii negatively regulates the expression of genes responsible for protein folding, purine synthesis and metabolism; Bifidobacterium catenulatum increases the expression of the ABC transport pathway, decreases metabolic and cell wall synthesis pathways; Enterococcus caccae upregulates the genes responsible for energy production and metabolism and downregulates the stress response, translation and sugar transport pathways [68].

Thus, a better understanding of the interactions of polyphenols with the intestinal microbiota will lead to a better knowledge of their biological activities and optimise their use in prevention or treatment of diseases such as cancer. Furthermore, the identification of metabolites resulting from their degradation by the intestinal microbiota may lead to the discovery of new molecules with greater pharmacological activities than the natural molecules from which they are derived.

5. Antitumor Efficiency

5.1. Apigenin

The Mediterranean diet contains high quantities of flavonoids, a group representing 60% of natural polyphenols. Apigenin is the most commonly occurring flavonoids found in several dietary constituents [69]. A diet rich on flavonoids may be considered useful to fight cancer development [70]. It has been reported that apigenin is able to suppress cell growth in many types of human cancer cells: leukemia [71], melanoma [72], breast cancer [73], bladder cancer [74], thyroid cancer [75] and colonic cancer [76]. Apigenin has multiple anti-cancer effects [69]: anti-proliferation, anti-inflammation, and anti-metastatic activities. It inhibits growth of different types of cancer. Many signaling pathways are involved on the anti-carcinogenic mechanism of apigenin; the best known of which are MAPK, NF-κB, P53, signal transducer and activator of transcription (STAT3), PI3K/Akt, cyclooxygenase (COX) and Wnt/β-catenin pathway [77] (Table 4).

Table 4. Antitumoral effects of apigenin in colorectal cancer.

APIGENIN

Experimental
Models

Concentration range

Biological response

Pathway/genes/proteins
involved

Refs

Cells & cell lines

HCT-15

43.28 µM

cell cytotoxicity

Apoptosis

Cell cycle arrest G2/M

↑p21, ↑cyclin B1

[76]

SW480

40 µM

Proliferation

Invasion

Migration

↓Wnt/β-catenin

[78]

Cell migration, Invasion, Metastasis

↓FAK, Src, crk-L, AKT

[79]

Proliferation

↓NEDD9

HCT-116

25 µM

Proliferation, Apoptosis  Autophagy

↓Cyclin B1,  ↓Cdc2, Cdc25c,
↑PPAR cleavage, ↑LC3-II

[80]

20 μM and
40 μM

Autophagy/ ApoptosisCell grouth, cell cycle arrest G2/M

↓PI3K/AKT/Mtor

[81]

10 µM

Apoptosis, Trascriptional level

PKCδ /ATM kinase, ↓NAG-1,
↓p53, ↓p21

[82]

LoVo

1-10 µM

Apoptosis

↓NAG-1, ↓p53, ↓p21

DLD-1

40 µM

Cell migration, Invasion, Migration

↓NEDD9, ↓FAK, Src, crk-L, AKT, ↑TAGLN, ↓MMP-9

[79]

Cell migration, Invasion,

↑TAGLN, ↓MMP-9

[88]

HT-29

45.96 µM

cell cytotoxicity

Apoptosis

↑p21, ↑cyclin B1

[76]

Animal models

Athymic nude mice

20 mg/kg

(I.P)

Cell migration, Invasio

proliferation n

↓FAK, Src, crk-L, AKT

[79]

↓NEDD9

50 mg/kg (I.P)

Angigenesis

proliferation

 ↓CD-31, ↓Ki-67

[76]

BALB/c-nude mice

50 mg/kg

(Per Os)

Cell migration, Invasion, Proliferation

↑TAGLN, ↓MMP-9

[88]

APCMin/+ mice

50 mg/kg (I.P)

tumor volume, Apoptosis

↑p21,  ↓p53

[82]

5.2. Curcumin

Curcumin acts as anti-inflammatory, anti-oxidant and antiproliferative agent. It has been recommended for chemopreventive, anti-metastatic, and anti-angiogenic purposes [84]. Several studies have shown that curcumin as an anti-carcinogenetic agent, used on many type of tumors such as prostate, pancreas, breast, stomach, liver carcinomas, and leukemia [85]. Despite its limited bioavailability observed by oral administration, curcumin modulates several illnesses especially intestinal inflammatory pathologies such as Crohn’s disease, necrotizing enterocolitis and ulcerative colitis [86]. Epidemiological studies have shown that patients with long-term intestinal inflammation present a risk of developing colorectal cancer [87]. Curcumin is taken up by intestinal epithelial cells [88]. Curcumin inhibits anti-apoptotic Bcl-2 proteins family in intestinal epithelial cells and stimulates the expression of p53, Bax, procaspases 3, 8, and 9 and the pro-apoptotic Bax protein [89]. Curcumin binds vitamin D receptor present in intestinal epithelial cells involved in the promotion of colon cancer chemopreventing process [90]. Table 5 summarizes the antitumor effects of curcumin in colon cancer, from in vitro and in vivo models and human clinical trials (Table 5).

In vivo studies performed in rats show that curcumin elimination is slower by intraveinous (i.v.)administration (44.5 min) for a dose of 1 g/kg [91], than when orally administered (28.1 min) at a dose of 300 mg/kg [92]. Results from human clinical trials [93], demonstrate that curcumin bioavailability was increased to about 20-fold when combined with piperine. Di Silvestro et al., [94] showed that lipidated curcumin increases its bioavailability.

Table 5. Antitumoral effects of curcumin in colorectal cancer.

CURCUMIN

Experimental

Models

Concentration range

Biological response

Pathway/genes/proteins

involved

Refs

Cells & cell lines

HCT-116

10-25 µM

Apoptosis

↓AP-1, ↓ NF-κB ,

↓MMP-9 

[95]

20 µM with

5-FU (5 µM)

Cell cycle arrest (S)

Apoptosis

Cell proliferation

↓caspase-3, ↓caspase-8, ↓caspase-9, Bax, ↓PARP, ↑Bcl-2

[96]

↓cyclin D1

25 µM with Piperine (7 µM)

Cell proliferation

Cell cycle arrest (G2/M

Apoptosis

↓cyclin D1, ↑caspase-3

[97]

HT29

41 µM

Oxydative stress

Cell growth,

Invasion,

Metastasis

↓NF-E2, ↓Nrf2

↓Bcl-2, ↓Cyclin D1,

↓IL6, ↓Cox2

[17]

HCT-8/5-Fu

10 µM with

5-FU (10 mM)

Apoptosis,

↑Nrf2, ↑Bcl-2, ↓Bax

 [98]

Animal models

C57BL/6

300 mg/kg with  DSS (5 mg/kg) I.P.

Disease activity index,

neoplasic lesions

↓β-catenin, Cox2, iNOS

[99]

Apoptotosis

↓cyclinD1, ↓cyclinD3, ↑caspase-3, ↑caspase-7,

↑caspase-9, ↑PARP

[92]

Oxaliplatin-resistant

HCT116-xenograft

(1 g/kg) per os

Radiosensitivity

↓NF-κB, ↓Ki-67,

↓Notch-1

[100]

Orthopically implanted CRC tumors (HC116)

(1 g/kg) per os

cell growth,

metastasis

↓NF-κB

[91]

5..3. EGCG

Epigallocatechin-3-gallate (EGCG) is the major biologically active component and the most abundant polyphenol from green tea [101]. Its antitumoral activities in colorectal cancer, are well documented (Table 6).

Epidemiological investigations and experimental studies have shown that EGCG exerts an important role by its anticancer and cancer chemopreventive properties. It has been observed to suppress breast cancer [101], lung cancer [102], pancreatic cancer [103] and liver cancer [104]. In vivo studies have demonstrated an inhibitory effect of EGCG on inflammation and obesity related colon carcinogenesis in rodent models [105]. Studies have demonstrated that EGCG exerts its inhibitory role in colorectal cancer and its metastasis by suppressing cell proliferation, inducing apoptosis in human CRC cells [106]. EGCG plays a crucial role on obstructing and inhibiting colorectal cancer sphere formation and proliferation, respectively [107]. The possible molecular mechanisms of EGCG in colorectal cancer involve various molecules and signaling pathways [108] (Table 5).

Table 6. Antitumoral effects of EGCG in colorectal cancer.

EGCG

Experimental
Models

Concentration range

Biological response

Pathway/genes/proteins
involved

Refs

Cells & cell lines

SW837

50-100 µM

Cell growth

↑IFN‑γ, ↓IDO,

↓STAT1,

↓JAK/STAT1,

↓ISRE, ↓GAS

[109]

10 ng/mL

Cell proliferation,

Apoptosis

↑ CD133, CD44, ALDHA1,

Oct-4, and Nanog,

↓p-GSK3β, ↑GSK3β,

↓Wnt, ↓β-catenin, ↓Cyclin D1, ↓PCNA, ↓Bcl2,

↑Bax, ↑caspases 3, 8 and 9

[110]

10–30 µM

Cell growth,

Cell cycle arrest (G2/M),

Proliferation,

Cell invasion,
Cell adhesion,

Apoptosis

↓MMP2/9, 

↑caspases 3, 8 and 9, 

↓EGFR and IGF1R, 

↓MEK and ERK,

↓PI3K and AKT, ↓Bad

[111]

35 µg/mL

Cell cycle arrest (G0/G1)

Cell proliferation, Apoptosis

[112]

 LoVo

35 µg/mL

Cell proliferation,

Apoptosis,
Cell cycle arrest (G0/G1)

[112]

10–30 µM

Cell growth,

Cell cycle arrest (G2/M),

Proliferation,

Cell invasion,
Cell adhesion, 

Apoptosis

↓MMP2 and 9,

↑caspases 3, 8 and 9,

↓EGFR and IGF1R,

↓MEK and ERK,

↓PI3K and AKT, ↓Bad

[111]

HT29

35 µg/mL

Cell cycle arrest (S)

[112]

88 μM

262 μM

190 μM/ 88 μM

262 μM/190 μM/ 88 μM

262 μM/190 μM/ 88 μM

88 μM

ER stress,

Apoptosis

↑Bip

↑p-eIF2α

↓PERK, ATF4

↑IRE1α

↑Caspases 3 and 7

↑TfR

[113]

100 µM (with 20 µM csplatin

or 20 µM oxaliplatin )

Cell viability,

Autophagy

↑LC3II, ↓ IP3K

[113]

HCT-8

35 µg/mL

Cell cycle arrest G2/M

[112]

HCT116

12.5 µM

Radiosensitivity,

Autophagy & Apoptosis

↑Nrf2 , ↑LC3,

↑Caspase-9

[115]

50-100 µM

Cell proliferation

[116]

50-100 µM

Apoptosis

↓VEGFR2,  ↓AKT, 

↓tumor growth, ↓proliferation,

↓migration and ↓angiogenesis

[116]

10–30 µM

Cell growth,

Cell cycle arrest (G2/M),

Proliferation,

Cell invasion,

Cell adhesion, Apoptosis

↓MMP2 and 9, 

↑caspases 3, 8 and 9,

↓EGFR and IGF1R,

↓MEK and ERK,

↓PI3K and AKT, ↓Bad

[111]

DLD-1,

100 µM with (20 µM csplatin
or 20 µM oxaliplatin)

Cell viability,

Autophagy

↑LC3II, ↓ IP3K

[114]

10 ng/mL

Cell proliferation,

Apoptosis

↑ CD133, CD44, ALDHA1,

Oct-4, and Nanog, 

↓p-GSK3β, ↑GSK3β, ↓Wnt,

↓β-catenin, ↓Cyclin D1, ↓PCNA, ↓Bcl2, ↑Bax, ↑caspases 3, 8 and 9

[110]

RKO

50-100 µM

Apoptosis

 ↑ p38

[116]

Caco-2

10–30 µM

Cell growth,

Cell cycle arrest G2/M,

Proliferation,

Cell invasion,

Cell adhesion,

Apoptosis

↓MMP2 and 9, ↑caspases 3, 8 and 9, ↓EGFR and IGF1R,  ↓MEK and ERK,

↓PI3K and AKT, ↓Bad

[111]

Animal models

 Male ICR mice

0.1 % with (AOM 10 mg/kg body weight I.P followed by 2 % (w/v) DSS)

Weight,

Inflammation

↓COX2, ↓mRNA

 (TNFα, IFN δ, IL6, IL12, IL18)

[105]

Eighty SPF Wistar rats

200 mg/kg with ( DMH 40 mg/kg, s.c)

Tumor volume,

Apoptosis

↓p53, PI3K-Akt ,

↓I-kappaB kinase/NF-kappaB , ↓MAPK

[117]

5.4. Quercetin

Quercetin has been shown to have anti-tumor efficacy against intestinal and colon cancer cells both in vitro and in vivo (Table 7).

Table 7. Antitumoral effects of quercetin in colorectal cancer.

QUERCETIN

Experimental
Models

Concentration range

Biological response

Pzathway/genes/proteins
involved

Refs

Cells & cell lines

HT-29

25, 50, 100 µM

Apoptosis

Bcl-2, cleaved caspase-3, cleaved PARP, p-Akt, ErbB2/ErbB3 proteins

[117]

5-30 µg/mL (with resveratrol,

1:1 ratio)

 Oncogenic
µmicroRNA-27a

Sp1, Sp3, Sp4,

survivin mRNA & proteins

[118]

50, 100, 200 µM

Apoptosis
 S-phase arrest

p-Akt, CSN6, Myc, Bcl-2, p53, Bax proteins

[119]

50 μM (with cisplatin: 10 mg/L)

Cisplatin-induced Apoptosis

Activation of NF-κB protein expression

[120]

30 µM

 TRAIL-induced Apoptosis

 Redistribution of death receptors DR4 & DR5 into lipid rafts

cleaved caspase-3 & cleaved Bid proteins,release Cyt-C

[121]

50, 100 μM

Apoptosis
 G1-phase arrest

AMPK, p53, p21 proteins

[122]

50 µM

(with dox: 250 nM)

Doxorubicin-induced cytotoxicity

Proliferation, apoptosis & G2/M arrest  for lower IC50 of Dox

[123]

Caco-2

5-50 µM

Cell proliferation

CDC6, CDK4, cyclin D1 mRNA

[124]

5-20 μM

Anti-migration
Anti-invasion

MMP-2, MMP-9, TLR4, NF-κB, E-cadherin proteins
TNF-α, COX-2, IL-6 production

[125]

Caco‑2
   & SW‑620

25-100 µM

Apoptosis

IκB-α, p-IκB-α, Bcl-2,
Bax proteins

[126]

SW480

20-80 µM

Apoptosis

Cyclin D1, survivin mRNA & proteins

[127]

10 µM

Apoptosis
S-phase arrest

EGF receptor phosphorylation

[128]

Colo-320
& Colo-741

25 μg/mL

 Apoptosis Senescence

p16, Lamin B1, cyclin B1, Bax, Bcl-2 proteins

[129]

CT26
& MC38

1-10 µM

Anti-metastasis

E- cadherin, N-cadherin,
β-catenin, snail proteins
MMP-2, MMP-9 activities

[130]

DLD-1

10.5 µM

Anticarcinogenesis

COX-2 transcription

[131]

CO115
& HCT15

 12 µM
(with 5-FU: 1 µM)

Fluorouracil-induced apoptosis

p53, cleaved caspase-9,
cleaved caspase-3,

cleaved PARP, Bcl-2 proteins

[132]

HCT8-β8

50 µM

Cell proliferation

ERβ mRNA, ER-responsive luciferase activity

[133]

Animal models

HT-29 xenograft in Balb/C nude mice

10 mg/kg/day
(SC; 4 weeks)

 Radiosensitivity

Jagged-1, Notch-1, Hes-1,
Presenilin 1, Nicastrin proteins

[134]

In vitro exposure of colon cancer cells such as HT-29, Caco-2, SW480, Colo-320 and Colo-741 to different concentrations of quercetin (5–200 μM), induced apoptosis (indicated by chromatin condensation and internucleosomal DNA fragmentation) through procaspases activation coincident with an alteration in the expression of c-myc, p53, p16 and p21, or by inhibition of NF-κB pathway [129]. Another mechanism supporting the anticarcinogenic effect of quercetin is through downregulation many cell cycle genes which cause cell cycle arrest, and suppression of tumor cell invasion and migration by regulating the expression of matrix metalloproteinases (MMPs), such as β-catenin [130], which can act as an invasion suppressor, and E-cadherin which is lost during tumor metastasis in association with the epithelial mesenchymal transformation [130]. Quercetin can also influence colon cancer cell proliferation and differentiation by downregulating the expression of cyclin D1 [127] and CDC6 [124]. The anti-inflammatory effect of quercetin at 5–20 μM have also been described in Caco-2 and DLD-1 cells as a result of inhibition of cyclooxygenase-2 (COX-2) [131] and cytokine expression [130].

Multiple studies confirmed quercetin’s effectiveness in reducing tumors in mice and rats, which is consistent with its in vitro effects. Quercetin at 50 mg/kg/day causes a reduction in tumor numbers in rats treated with 1,2-dimethylhydrazine (DMH) via activating the antioxidant defenses and limiting lipid peroxidation and protein peroxidation [131-136]. In CRC mice, quercetin decreased colon damage and mortality while reducing certain CRC-associated gut bacteria, such as Allobaculum and unclassified Erysipelotrichaceae [137]. Cruz-Correa M et al.,[138], showed that after a mean of 6 months of treatment, a combination of curcumin and quercetin reduced the number and size of polyps in patients with Familial Adenomatous Polyposis without causing a significant toxicity. Although more research is required to fully understand the relationship between quercetin consumption and colon cancer in humans, regarding its bioavailability (that may be affected by the food source) and the presence of other bioactive compounds in foods that can interact with quercetin and modulate tumor risk.

5.5. Rutin

Many mechanisms have been suggested to explain the in vitro anti-carcinogenic effects of rutin (Table 8).

Table 8. Antitumoral effects of rutin in colorectal cancer.

RUTIN

Experimental
Models

Concentration range

Biological response

Pathway/genes/proteins
involved

Refs

Cells & cell lines

HT-29

100-200 µM

Apoptosis

Bax, Bcl-2, cleaved caspases‑3, 8, 9, cleaved PARP proteins

[139]

39 mM (with Silibinin: 76 mM)

Apoptosis

p53, Bcl-2, Bax ,caspase 3, 8, 9
NFkB, IKK-α, IKK-β,
p38MAPK, MK-2 proteins

[140]

HT-29
& Caco-2

25-200 µM

Cell adhesion & Migration

ROS level, impairing attachment to fibronectin, disrupting cell–ECM interactions

[141]

136 µM

Cell proliferation

Growth potency

[142]

Animal models

SW480 xenograft in nude mice

1-20 mg/kg/day (I.P; 32 days)

Anti-tumor
Anti-angiogenesis

 Mean survival time, tumor volume & weight, VEGF levels in serum

[143]

MTX-treated Wistar rats (Intestinal inflammation)

50, 100 mg/kg/day (I.P; 1 week)

Oxidative stress
Inflammation

COX-1, COX-2 & 15 LOX enzymatic activities, restoration of MDA, protein carbonyl, SOD, GSH levels & catalase activity, free acidity & total acidity

[144]

5-FU-treated Swiss mice (Intestinal Mucositis)

50-200 mg/kg/day (Per os; 3 days)

Oxidative stress
Inflammation

MDA, GSH concentrations,
MPO activity, intestinal mastocytosis, COX-2 proteins

[145]

 DSS-treated ICR mice (Colitis)

 0.6-6 mg/day (Per os; 2 weeks)

Inflammation

IL-1β, IL-6, GM-CSF, iNOS mRNA

[146]

Rutin induces apoptosis (increase in the levels of proteins associated with apoptotic cell death: phospho-Bad, cleaved caspase 3 and cleaved PARP) and inhibits cell proliferation and adhesion (disrupting cell–extracellular matrix interactions) in HT-29, Caco-2 and HCT116 cells at concentrations ranging from 25 to 1000 μM [140]. When Caco-2 cells were exposed to rutin (over 1,000 μM), the antiproliferative effects of 5-fluorouracil (5-FU) and oxaliplatin were enhanced and the potential side effects of these drugs were decreased, suggesting that rutin may have chemopreventive and chemotherapeutic effects on colon cancer. In dextran sulfate sodium (DSS) chemically-induced mouse model of colitis, 6 mg/day of rutin suppressed the induction of pro-inflammatory cytokines, resulting in improved symptoms [146]. Another study showed that the administration of rutin (50, 100 mg/kg/day) offered an extensive protection against methotrexate (MTX)-induced intestinal lesions in rats [144]. This may be due to rutin’s anti-inflammatory activities (inhibition of COX-1, COX-2, and 15 LOX enzymatic activity) and free radical scavenging properties (restoration of MDA, protein carbonyl, SOD, GSH levels, and catalase activity).

5.6. Resveratrol

Cancer chemopreventive activity of resveratrol, a natural product derived from grapes was firstly reported by Jang et al., [147]. A review on cancer prevention potential of resveratrol has been recently published [148] and includes: case-control trials (Resveratrol and breast cancer risk [149]); clinical studies (clinical pharmacology of resveratrol and its metabolites in colorectal cancer patients [150,151]. In vivo studies reported the antitumor effect of resveratrol on the development of spontaneous mammary tumors in HER-2/neu transgenic mice [152]); the impact of metastasis [153]; the effect of resveratrol metabolites [154]; and targets and mechanistic aspects [155].

Resveratrol shows all of the antiproliferative and pro-apoptotic properties necessary to be an antitumor agent [156]. Conversely, P-glycoprotein 1 affects chemoactivities of resveratrol against human colorectal cancer cell [157]. After cell exposure, resveratrol is accumulated in lipid rafts, which is the first step for endosome formation and triggers apoptosis [158]. Signaling molecules and transcription factors involved in the cell cancer promotion or prevention are sensitive to resveratrol: PPAR, PGC1α, NF-kB, NRF1,2, p53. Nuclear cell cycle components, phosphatases, and cyclins are targets of resveratrol [148].

Resveratrol modulates both oncogenic miRNAs and the expression of tumor-suppressor miRNAs. In SW480 colon cancer cell line, resveratrol treatment decreases the levels of several oncogenic miRNAs targeting genes encoding tumor suppressors and effectors of TGFβ signaling pathway, while increasing the levels of miR-663. While upregulating several components of the TGFβ signaling pathway, resveratrol decreases the transcriptional activity of SMADs, the main effectors of the canonical TGFβ pathway [159].

Resveratrol inhibits human leiomyoma (a begnin smooth muscle tumor) cell proliferation via crosstalk between integrin αvβ3 and IGF-1R [160].

Levi et al., [149] analysed the relationship between dietary intake of resveratrol and breast cancer risk from a case-control study, conducted between 1993 and 2003 in the Swiss Canton of Vaud, on 369 cases and 602 controls. A significant inverse association was observed for resveratrol from grapes (OR = 0.64 and 0.55), but not from wine, suggesting that daily p.o. doses of resveratrol at 0.5 or 1.0 g produce levels in the human gastrointestinal tract in the order of magnitude sufficient to elicit anticarcinogenic effects.

5.7. Synergistic effects of polyphenol mixtures

While these effects are mostly obtained in vitro or in small laboratory animals at concentrations that are too high to be compatible with its in vivo low availability, resveratrol can slow down tumor progression in numerous cancer types, for instance blood [161], breast [162], cardiac [163], colon [164], glioma [165], lung [166], melanoma [167] and prostate [168]. As resveratrol is not cytotoxic it can be used as food supplement at high concentrations compatible with a significant plasma levels. Moreover, the synergistic effects of other food polyphenols [169], and the fact that resveratrol plasma metabolites also exhibit antiproliferative activities, is relevant for diet-dependent cancer prevention [154]. Interestingly, Mazue et al., reported protective effects of red wine extract in rat colon carcinogenesis [170].

6. Perspectives

6.1. Improvements

The health benefits of Mediterranean diet are now well established. To go further, the goal would be to expand this diet over the world in order to benefit more and more populations. This is a question of public health which needs to be relayed by the media. On the other hand, comparable polyphenols-rich diets are also observed in non-Mediterranean coastal regions at the same latitude, with analagous cooking and eating habits. Besides this, more experimental approaches are needed, especially at the preclinical and clinical studies (case-control studies) and information provided from cohorts. A better knowledge of the possible synergies between single polyphenols in mixtures with other nutrients should be accomplished. We may also pay attention to the influence of immune system in the cancer prevention under a Mediterranean diet. Related to the role of microbiome, the following question should be addressed: is there any feedback between the dietary polyphenol-modification of the microbiome ecosystem to the cancer immune defense dependency, or from the gut (second brain) to the central nervous system? Indeed, polyphenol-microbiota interactions have led to reconsideration of the gut-brain axis, initially revealed by the capacity of certain enteric nerve cells to secrete serotonin (5-hydroxytryptamine) involved at the brain level in the control of behaviour, mood and anxiety [171]. Several studies highlight the intervention of the polyphenol-microbiota duo not only in the occurrence of neurospsychiatric diseases such as shizophrenia [172] but also in metabolic diseases [173]. The intestinal microbiota, by acting on immune, endocrine and metabolic signaling, is thus able to exert beneficial clinical effects on depression and anxiety [174]. The term psychobiome takes into account the intestinal microbiota by considering its capacity for endocrine substances intervening at the level of several pathways involved in the modulation of neuro-inflammation and in the production of several neurotransmitter precursors [175]. In the context of polyphenol-microbiota interactions, beneficial effects of quercetin as well as curcumin [173], resveratrol [176] and EGCG [41] have been shown. The prevention of anxiety by polyphenols could be beneficial for cancer patients, particularly those with CRC, by providing better psychological support for restrictive therapies.

6.2. Nanoformulation Improvement

As described in Section 6.1, the selected polyphenols in this review provide numerous healthy benefits. However, the problem of their limited uptake from the diet is difficult to solve, because their poor aqueous solubility (Figure 2B) and therefore their weak bioavailability. The nanoformulation of bioactive polyphenols may overcome these problems. The benefits of this concept is that it is able to transport and to release the biological compound to target sites, without modifying its molecular structure. In addition, the transport of a poorly aqueous soluble compounds by a molecular-carrier allows improvement of its bioavailability and therefore, its biological activities [177]. The polyphenols mentioned in this review have been subjected to various forms of nanoformulation, especially resveratrol [178], curcumin [179], quercetin [180], catechins [181] and to a lesser degree, rutin [32] and apigenin [182]. It appears from the literature that in vitro and in vivo bioactivities of nanoformulated polyphenols are generally much better than those obtained with free polyphenols, particularly in the cancer field. While quercetin is poorly soluble in water, the loading of this polyphenol in biodegradable monomethoxy poly(ethyleneglycol)-poly(ε-caprolactone micelle (MPEG-PCL) makes possible its complete dispersion in water [183]. Compared to free quercetin, quercetin-loaded MPEG-PCL micelles showed a better in vitro inhibition of growth of CT26 cells with a weaker IC50 and better induction of apoptosis. Similarly, quercetin-loaded MPEG-PCL micelles repressed colon tumor in mice subcutaneous CT26 colon cancer [184]. To further target the biomarkers overexpressed by colon cancer cells such as EpCAM (Epithelial Cell Adhesion Molecules), nanoparticles may be decorated with specific ligands. Thereby, apigenin-loaded nanoparticles were conjugated with aptamer. The resulting aptamer-conjugated apigenin-loaded nanoparticles were evaluated for their antiproliferative properties toward colon carcinoma cells in vitro, as well as an in vivo colorectal cancer model. Using pharmacokinetic and biodistribution studies, the authors highlighted a greater retention of aptamer- conjugated apigenin-loaded nanoparticle in the colon as compared to free apigenin and aptamer-conjugated empty nanoparticles [182]. Encapsulation of resveratrol in liposomes allows increased aqueous solubility, photostability and bioavailability of this polyphenol [185]. In addition, resveratrol-loaded liposomes target cancer cells effectively and may show a greater apoptotic activity of HT29 colon cancer cells than free resveratrol [186]. Dual encapsulation involving liposomes and cyclodextrins improves the nanoformulation of resveratrol and its release to HT29 colon cancer cells. Indeed, cyclodextrin stabilizes the polyphenol in its hydrophobic cavity and liposome targets cancer cells [187]. Resveratrol may be encapsulated in PLGA-PEG (poly lactic glycol acid–poly ethylene glycol), a biodegradable and biocompatible polymeric nanoparticle coated with chitosan. The resulting resveratrol-loaded micelles were evaluated in vivo for their anti-proliferative and anti-angiogenic effects in human COLO205-luc colon cancer in xenograft and orthotopic implantation models in athymic mice [188]. In this study, the authors observed decreased tumor growth due to the action of resveratrol-loaded micelles. The anti-angiogenic effect of these resveratrol nanoparticles have been highlighted by the decrease of the Hb percentage of tumor mass. To explain these promising results, the authors put forward the idea of greater bioavailability of resveratrol-loaded micelles compared to those of free resveratrol.

Curcumin nanoformulation applied to colorectal cancer has been the object of a complete and recent review [189]. Various nanoformulations have been mentioned in particular liposomes, micelles, polymer nanoparticles, cyclodextrins. This review highlighted the advantages of most of the curcumin- loaded nanoparticles but also the limitations of others, such as some cyclodextrin and lipid nanoparticle formulations. The combination of successful results of in vitro and in vivo studies involving curcumin-loaded nanoparticles have allowed the commencement of clinical trials evaluation of the effects of these in intestinal cancer. Synergistic effects between two or more bioactive molecules may afford enhanced activities or complementary activities, more so when these molecules are nanoformulated in a same system. The molecular structures of the co-encapsulated bioactive compounds may be similar or very different from those of the selected polyphenols, as shown in Figure 5. For example, alantolactone (7) is a natural sesquiterpene lactone (Figure 5) isolated from Inula racemose Hook. f. providing antitumor activities through an apoptotic mechanism [190]. The combination of quercetin and alantolactone into micelles allowed induction of immunogenic cell death in microsatellite-stable CRC [191]. Chrysin (8), a natural flavone whose structure is similar to that of quercetin or apigenin (Figure 5) provides multiple properties including anti-tumor activities [192]. Thus, evaluation of activity of curcumin-chrysin loaded in PLGA-PEG nanoparticles towards Caco-2 cancer cells has shown a greater efficiency than either individual compound: a quicker killing of cancer cells and a more efficient suppression of hTERT expression [193]. Piperine (9), an alkaloid isolated from black and long peppers (Figure 5), is known for its ability to inhibit proliferation of HT-29 colon carcinoma cells [194]. The synergistic effect of curcumin and piperine (non-nanoformulated) has been demonstrated in an in vivo experiment against hepatocellular carcinoma in rats [195]. The co-encapsuation in emulsomes of these both bioactive compounds confirmed their synergistic benefits towards HT116 colorectal cancer cells [97].

Figure 5. Chemical structures of co-encapsulated compounds: Alantolactone (7), Chrysin (8) Piperine (9).

6.3. Future Prospects

Polyphenols may be absorbed by humans via the Mediterranean diet and in this condition, may play a preventative role against cancers and other diseases. To strengthen this prevention, the intake of polyphenols in the form of food supplements has been developed and some of these are widely marketed. In this field, the use of nanotechnology is an attractive concept. In a complete and interesting review, D.D. Milincic et al., reported various polyphenol nanoformulations applied in food industry [196]. The authors highlighted advantages of nanotechnology in preserving healthy benefits of the bioactive compounds, especially stability, anti-oxidant properties and intestinal bioavailability. However, they warned about an uncontrolled consumption of polyphenols which can lead to harmful effects in humans, because these compounds at high doses may promote pro-oxidant effects [197]. In addition, polyphenols at high doses can chelate bio-elements and decrease their absorption by the gastrointestinal tract [198]. Indeed, nanoparticle toxicity must be not neglected and the use of nanoformulated polyphenols for therapeutic applications and food supplements requires further safety-risk assessments.

In conclusion, the future development of nanoformulations should provide new properties of dietary polyphenols, to benefit cancer prevention, especially due to increased polyphenols cell absorption, plasma level and targeting.

Author Contributions: Conceptualization, N.L. Investigation: A.Y., A.N., G.L. and D.V-F; Writing original draft, A.N., G.L., D.V.-F; A.Y. and N.L. Formal analysis, GL, A.Y., A.N., D.V.-F., and N.L. Scientific additive of dietary polyphenols, and English correction, J.J.M. All authors have read and agreed to the submitted version of the manuscript.

Funding: This work was funded by Université de Bourgogne and by CNRS (Dijon, France). This work was also supported by the Council of Bourgogne Franche-Comté (MEDICTA project).

Acknowledgments: The authors would like to thank the association “Mediterranean Nutrition and Health (NMS: Nutrition Méditerranéenne & Santé)”. A.Y. received financial support from NMS and awarded the NMS prize in 2019. A.N. is currently post-doc fellow in the laboratory after granting at in cotutelle with the University of Burgundy and the University of Tunis El Manar (Tunis, Tunisia). We also thank the UNESCO Chair, wine culture and tradition, Dijon, for its continuous interest.

Conflicts of Interest: The authors declare no conflict of interest.

Abbreviations: 5-FU: 5-FluoroUracil; ACC: Acetyl-CoA Carboxylase; Akt: Protein kinase B; ALDH1A1: ALdehyde DeHydrogenase 1 family member A1; AMPK: 5’ AMP-activated Protein Kinase; AOM: AzOxyMethane; AP-1: Activator Protein 1; ATF4: Activating Transcription Factor 4; Bad: Bcl-2 associated agonist of cell death; Bax: Bcl-2- associated X protein; Bcl-2: B-cell lymphoma-2; Bid: BH3 interacting domain death agonist; BIP: Binding Immunoglobulin Protein; BrdU: BromodeoxyUridine; CDC6: Cell Division Cycle 6; CDK4: Cyclin-Dependent Kinase 4; COX: CycloOXygenase; COX-2: CycloOXygenase-2; CSN: Constitutive photomorphogenesis 9 Signalosome; Cyt-C: Cytochrome c; DMH: 1,2-DiMethylHydrazine; DSS: Dextran Sulfate Sodium; ECM: ExtraCellular Matrix ; EGF: Epidermal Growth Factor; EGFR: Epidermal Growth Factor Receptor; EpCAM: Epithelial Cell Adhesion Molecules; ErbB2/ErbB3: Epidermal growth factor receptors; ERK: Extracellular Signal Regulated Kinases; ERβ: Estrogen Receptor beta; Fabp2: Fatty acid-binding protein 2; FAK: Focal Adhesion Kinase; FAP: Familial Adenomatous Polyposis; FtH: Ferritin receptor; GAS: Growth Arrest-Specific protein; GM-CSF: Granulocyte-Macrophage Colony- Stimulating Factor ; Gpt: Glutamic-pyruvic transaminase; GSH: Glutathione ; Hes-1: Hairy and enhancer of split-1; Hmgcs2: 3-Hydroxy-3-methylglutaryl-CoA synthase 2; hTERT: human TElomerase Reverse Transcriptase ; IDO: Indoleamine 2,3-DiOxygenase;IFN-γ: InterFeroN- gamma; IGF1R: Insulin like Growth Factor 1 Receptor; IKK: Inhibitory-κB Kinase ; IL-1β: InterLeukin 1 beta ; IL-6: InterLeukin 6; iNOS: inducible Nitric Oxide Synthase ; IRE1α: Inositol-Requiring Enzyme 1; ISRE: Interferon-Stimulated Response Element; IκB-α: Inhibitor of nuclear factor kappa B; JAK: JAnus Kinase; LC3-II: Light Chain 3-II; LOX: LipOXygenase ; MAPK: Mitogen-Activated Protein Kinase ; MDA: MalonDiAldehyde; MEK: MAP/ERK kinase-1; MK-2: MAPK-activated protein Kinase-2 ; MMP: Matrix MetalloProteinase; MP 2/9: Matrix MetalloProteinases 2/9; MPEG-PCL: Monomethoxy Poly(EthyleneGlycol)- Poly(e-CaproLactone) micelle; MPO: MyeloPerOxidase ; MTX: MethoTreXate ; NAG-1: NSAID Activated Gene; NANOG: NANOG homeobox; NEDD9: Neural precursor cell Expressed Developmentally Down-regulated protein 9; NF-E2/ Nuclear Factor, Erythroid 2; NF-κB: Nuclear Factor kappa B; Nrf2: Nuclear-related factor 2; Oct-4: Octamer-binding transcription factor 4; p53: protein 53; PARP: Poly (ADP- Ribose) Polymerase; PCNA: Proliferating Cell Nuclear Antigen; p-eIF2α: phosphorylation of eukaryotic Initiation Factor-2α; PERK: PKR-like ER protein Kinase; p-GSK3β: Glycogen Synthase Kinase 3 beta; PI3K: PhosphoInositide 3-Kinase; PLGA-PEG: Poly Lactic GlycolAcid-Poly Ethylene Glycol; PPAR-γ: Peroxisome Proliferator-Activated Receptors-γ; ROS: Reactive Oxygen Species; SOD: SuperOxide Dismutase; Sp: Specificity protein transcriptions factors; STAT1: Signal Transducer and Activator of Transcription 1; TAGLN: TrAnsGeLiN; TfR: Transferrin Receptor; TLR4: Toll-Like Receptor 4; TNF alpha: Tumour Necrosis Factor alpha; TRAIL: Tumor-necrosis-factor Related Apoptosis Inducing Ligand; VEGF: Vascular Endothelial Growth Factor ; VEGFR2: Vascular Endothelial Growth Factor Receptor 2.

References

  1. Brat, P.; Georgé, S.; Bellamy, A.; Du Chaffaut, L.; Scalbert, A.; Mennen, L.; Arnault, N.; Amiot, M.J. Daily polyphenol intake in France from fruit and vegetables. Nutr. 2006, 136, 2368-2373. doi: 10.1093/jn/136.9.2368.
  2. Debbabi, M.; Karym, E.M; Camus, E.; Durand, P.; Cherkaoui-Malki, M.; Prost, M.; Jabrane, A.; Nasser, B.; Hammami, M.; Lizard, G. Etude comparative des profils anti-oxydants et de la composition en polyphénols de fruits et de légumes associés au régime méditerranéen; in Latruffe, N. Wine, Mediterranean Diet and Health (in French). 2017, pp. 63-70, EUD ed. Dijon.
  3. Willett, W.C.; Sacks, F.; Trichopoulou, A.; Drescher, G.; Ferro-Luzzi, A.; Helsing E.; Trichopoulos D. Mediterranean diet pyramid: a cultural model for healthy eating,. J. Clin. Nutr. 1995, 61, 14025-14065
  4. Del Rio, D.; Rodriguez-Mateos, A.; Spencer, J.P.; Tognolini, M.; Borges, G.; Crozier, A. Dietary (poly)phenolics in human health: structures, bioavailability, and evidence of protective effects against chronic diseases. Redox Signal. 2013, 18, 1818-1892. doi: 10.1089/ars.2012.4581.
  5. Schaffer, S.; Eckert, G.P.; Schmitt-Schillig, S.; Müller, W.E. Plant foods and brain aging: a critical appraisal. Forum Nutr. 2006, 59, 86-115. doi: 10.1159/000095209.
  6. Gorzynik-Debicka, M.; Przychodzen, P.; Cappello, F.; Alicja Kuban Jankowska, A.; Gammazza, A.M.; Knap, N.; Wozniak, M.; Gorska-Ponikowska, M. Potential Health Benefits of Olive Oil and Plant Polyphenols. J. Mol. Sci. 2018, 19, 686. doi: 10.3390/ijms19030686.
  7. Villarini, M.; Lanari, C.; Nucci, D.; Gianfredi, V.; Marzulli, T.; Berrino, F.; Borgo, A.; Bruno, E.; Gargano, G.; Moretti, M.; Villarini, A.Community-based participatory research to improve life quality and clinical outcomes of patients with breast cancer (DianaWeb in Umbria pilot study). BMJ Open 2016, 6, e009707. doi: 10.1136/bmjopen-2015-009707.
  8. Grosso, G.; Biondi, A.; Galvano, F.; Mistretta, A.; Marventano, S.; Buscemi, S.; Drago, F.; Basile, F. Factors associated with colorectal cancer in the context of the Mediterranean diet: a case-control study. Cancer. 2014, 66, 558-565. doi: 10.1080/01635581.2014.902975.
  9. Salvatore Benito, A.; Valero Zanuy, M.Á.; Alarza Cano, M.; Ruiz Alonso, A.; Alda Bravo, I.; Rogero Blanco, E.; Maíz Jiménez, M.; León Sanz, M. Adherence to Mediterranean diet: A comparison of patients with head and neck cancer and healthy population. Diabetes Nutr. 2019, 66, 417-424.
  10. Melander, O.; Antonini, P.; Ottosson, F.; Brunkwall, L.; Gallo, W.; Nilsson, P.M.; Orho-Melander, M.; Pacente, G.; D’Arena, G.; Di Somma, S. Comparison of cardiovascular disease and cancer prevalence between Mediterranean and north European middle-aged populations (The Cilento on Ageing Outcomes Study and The Malmö Offspring Study). Emerg. Med. 2021, Jan 30. doi: 10.1007/s11739-020-02625-4.
  11. Sofi, F.; Macchi, C.; Abbate, R.; Gensini, GF.; Casini, A. Mediterranean diet and health status: an updated meta-analysis and a proposal for a literature-based adherence score. Public Health Nutr. 2014, 17, 2769-2782. doi: 10.1017/S1368980013003169.
  12. Boccardi, V.; Esposito, A.; Rizzo, M.R.; Marfella, R.; Barbieri, M.; Paolisso, G. Mediterranean diet, telomere maintenance and health status among elderly.PLoS One 2013, 8, e62781. doi: 10.1371/journal.pone.0062781.
  13. McEvoy, C.T.; Hoang, T.; Sidney, S.; Steffen, L.M.; Jacobs, DR Jr.; Shikany, J.M.; Wilkins, J.T.; Yaffe, K. Dietary patterns during adulthood and cognitive performance in midlife: The CARDIA study. Neurology 2019, 92, e1589-e1599. doi: 10.1212/WNL.0000000000007243.
  14. Zhu, Y.; Xie, D.Y. Docking Characterization and in vitro Inhibitory Activity of Flavan-3-ols and Dimeric Proanthocyanidins Against the Main Protease Activity of SARS-Cov-2. Plant Sci. 2020, 11, 601316.
  15. Nadtochiy, S.M.; Redman, E.K. Mediterranean diet and cardioprotection: the role of nitrite, polyunsaturated fatty acids, and polyphenols. Nutrition 2011, 27, 733-744. doi:10.1016/j.nut.2010.12.006.
  16. Mojzer, E.B.; Hrncˇicˇ, M.K.; Škerget, M.; Knez, Z.; Bren, U. Polyphenols: Extraction Methods, Antioxidative Action, Bioavailability and Anticarcinogenic Effects. Molecules 2016, 21, 901. doi:10.3390/molecules21070901
  17. Estrela, J.M.; Mena, S.; Obrador, E.; Benlloch, M.; Castellano, ; Salvador, R.; Dellinger, R.W. Polyphenolic Phytochemicals in Cancer Prevention and Therapy: Bioavailability versus Bioefficacy. J. Med. Chem. 2017, 60, 9413−9436. doi: 10.1021/acs.jmedchem.6b01026
  18. Hu, Y.; He,Y.; Ji, J.; Zheng, S.; Cheng, Y. Tumor Targeted Curcumin Delivery by Folate-Modified MPEG-PCL Self-Assembly Micelles for Colorectal Cancer Therapy. J. Nanomed. 2020, 15, 1239-1252.
  19. Esatbeyoglu, T.; Huebbe, P.; Ernst, I. M. A.; Chin, D.; Wagner, A. E.; Rimbach, G. Curcumin - From molecule to biological function. Chem. Int. Ed.2012, 51, 5308-5332.
  20. Hatahet, T.; Morille, M.; Hommoss, A.; Dorandeu, C.; Muller, RH.; Begu, S. Dermal quercetin smartCrystals: Formulation development, antioxidant activity and cellular safety. J. Pharm. Biopharm. 2016, 102, 51-63.
  21. Iacopini, P.; Baldi, M.; Storchi, P.; Sebastiani, L. Catechin, epicatechin, quercetin, rutin and resveratrol in red grape: Content, in vitro antioxidant activity and interactions. Food Compos. Anal. 2008, 21, 589-598.
  22. Wein, S.; Beyer, B.; Zimmermann, B.F.; Blank, R. H.; Wolffram, S. Bioavailability of Quercetin from Onion Extracts after Intraruminal Application in Cows. Agric. Food Chem. 2018, 66, 10188-10192.
  23. Hertog, M.G.L.; Hollman, P.C.H.; Katan, M.B. Content of potentially anticarcinogenic flavonoids of 28 vegetables and 9 fruits commonly consumed in the Netherlands. Agric. Food Chem. 1992, 40, 2379-2383.
  24. Zhang, J.; Liu, D.; Huang, Y.; Gao, Y.; Qian, S. Biopharmaceutics classification and intestinal absorption study of apigenin. J. Pharm. 2012, 436, 311-317.
  25. Shukla, S.; Gupta, S. Apigenin: A Promising Molecule for Cancer Prevention. Res. 2010, 27, 962-978.
  26. Balata, G.F.; Essa, E.A.; Shamardl, H.A.; Zaidan, S.H.; Abourehab, M.A.S. Self-​emulsifying drug delivery systems as a tool to improve solubility and bioavailability of resveratrol. Drug Des. Dev.Ther. 2016, 10, 117-128.
  27. Hasan, M.M.; Bae, H. An overview of stress-​induced resveratrol synthesis in grapes: perspectives for resveratrol-​enriched grape products. Molecules 2017, 22, 294.
  28. Sales, J.M.; Resurreccion, A.V.A. Resveratrol in peanuts. Rev. Food Sci. Nutr. 2014, 54, 734-770.
  29. Lyons, M.M.; Yu, C.; Toma, R. B.; Cho, S. Y.; Reiboldt, W.; Lee, J.; Van Breemen, R. B. Resveratrol in raw and baked blueberries and bilberries. Agric. Food Chem. 2003, 51, 5867-5870.
  30. Raal, A.; Pokk, P.; Arend, A.; Aunapuu, M.; Jogi, J.; Okva, K.; Pussa, T. Trans-​resveratrol alone and hydroxystilbenes of rhubarb (Rheum rhaponticum L.) root reduce liver damage induced by chronic ethanol administration: a comparative study in mice. Res. 2009, 23, 525-532.
  31. Gullon, B.; Lu-Chau, T.A.; Moreira, M.T.; Lema, J.M.; Eibes, G. Rutin: A review on extraction, identification and purification methods, biological activities and approaches to enhance its bioavailability. Trends Food Sci.Technol. 2017, 67, 220-235.
  32. Fahra, A.K.; Gian, R.Y.; Li, H.B.; Wu, D.T.; Atanasov, A.T.; Gul. K.; Zhang, J.R.; Yang, Q.Q.; Corke, H. The anticancer potential of the dietary polyphenol rutin: current status, challenges, and perspectives. Rev. Food Sci. Nutr. 2020, doi: 10.1080/10408398.2020.1829541.
  33. Zhang, X.; Wang, J.; Hu, J.M.; Huang, Y.W.; Wu, X.Y.; Zi, C.T.; Wang, X.J.; Sheng, J. Synthesis and Biological Testing of Novel Glucosylated Epigallocatechin Gallate (EGCG) Derivatives. Molecules 2016, 21, 620. doi: 10.3390/molecules21050620.
  34. Fujiki, H. Green tea: health benefits as cancer preventive for humans. Record 2005, 5,119-132.
  35. Arts, I.C.; Hollman, P.C.; Kromhout, D. Chocolate as a source of tea flavonoids. Lancet. 1999, 354, 488.
  36. Zhang, Y.; Ouyang, L.; Mai, X.; Wang, H.; Liu, S.; Zeng, H.; Chen, T.; Li, J. Use of UHPLC-QTOF-MS/MS with combination of in silico approach for distributions and metabolites profile of flavonoids after oral administration of Niuhuang Shangqing tablets in rats. Chromatogr. B Analyt. Technol. Biomed. Life Sci. 2019, 1114-1115, 55-70. doi: 10.1016/j.jchromb.2019.03.021.
  37. Di Meo, F.; Margarucci, S.; Galderisi, U.; Crispi, S.; Peluso, G. Curcumin, Gut Microbiota, and Neuroprotection. Nutrients. 2019, 11, doi: 10.3390/nu11102426.
  38. Sharma, R.A.; Steward, W.P.; Gescher, A.J. Pharmacokinetics and pharmacodynamics of curcumin. Exp. Med. Biol. 2007, 595, 453-70. doi: 10.1007/978-0-387-46401-5_20.
  39. Hoehle, S.I.;Pfeiffer, E.; Sólyom, A.M.; Metzler, M. Metabolism of curcuminoids in tissue slices and subcellular fractions from rat liver. Agric. Food Chem.2006, 54, 756-64. doi: 10.1021/jf058146a.
  40. Gan, R.Y.; Li, H.B.; Sui, Z.Q, Corke, H. Absorption, metabolism, anti-cancer effect and molecular targets of epigallocatechin gallate (EGCG): An updated review. Rev. Food Sci. Nutr. 2018, 58, 924-941. doi: 10.1080/10408398.2016.1231168.
  41. Pervin, M.; Unno, K.; Takagaki, A.; Isemura, M.; Nakamura, Y. Function of Green Tea Catechins in the Brain: Epigallocatechin Gallate and its Metabolites. J. Mol. Sci. 2019, 20, 3630. doi: 10.3390/ijms20153630.
  42. Almeida, A.F.; Borge, G.I.A.; Piskula, M.; Tudose, A.; Tudoreanu, L.; Valentová, K.; Williamson, G.; Santos, CN. Bioavailability of Quercetin in Humans with a Focus on Interindividual Variation. Rev. Food Sci. Food Saf. 2018, 17, 714-731. doi: 10.1111/1541-4337.12342.
  43. Lançon, A.; Delmas, D.; Osman, H.; Thenot, J.P.; Jannin, B.; Latruffe, N. Human hepatic cell uptake of resveratrol: involvement of both carrier mediaten and diffusion process. Biophys. Res. Commun 2004. 316, 1132-1137.
  44. Jannin, B.; Menzel, M.; Berlot, J.P.; Delmas, D.; Lançon, A.; Latruffe, N. Interactions of resveratrol, a chemopreventive agent, with serum albumin. Pharmacol. 2004, 68, 1113-1118.
  45. Latruffe, N.; Menzel, M.; Delmas, D.; Buchet, R.; Lancçon, A. Compared binding properties between resveratrol and of other polyphenols to plasmatic albumin. Consequences for the health protecting effect of dietary plant microcomponents. Molecules 2014, 19, 17066-17077.
  46. Tung, W.C.; Rizzo, B.; Dabbagh, Y.; Saraswat, S.; Romanczyk, M.; Codorniu-Hernández, E.; Rebollido-Rios, R.; Needs, P.W.; Kroon, PA.; Rakotomanomana, N.; Dangles, O.; Weikel, K.; Vinson, J. Polyphenols bind to low density lipoprotein at biologically relevant concentrations that are protective for heart disease. Biochem. Biophys. 2020, 694, 108589. doi: 10.1016/j.abb.2020.108589.
  47. Delmas, D.; Aires,V.; Limagne, E.; Dutartre, P.; Mazue,F.; Ghiringhelli, F.; Latruffe, N. Transport and stability of Resveratrol condition its biological activity. N.Y. Acad. Sci. 2011, 1215, 48-59.
  48. Lançon, A.; Hanet, N.; Jannin, B.; Delmas, D.; Heydel, J.M.; Chagnon, M.C.; Lizard, G.; Artur, Y.; Latruffe, N. Resveratrol in human hepatoma HepG2 cells: metabolism and inducibility of detoxifying enzymes. Metab. Dispo. 2007, 35, 699-703.
  49. Colin, D.; Lançon, A.; Delmas, D.; Abrossimow, J.; Kahn, E.; Lizard, G.; Jannin, B.; Latruffe, N. Comparative study of the cell uptake and the antiproliferative effects of resveratrol, epsilon-viniferin and their acetates. Biochimie 2008, 90,1674-1684.
  50. Kumar, P.S. Microbiomics: Were we all wrong before? 2000, 85, 8-11. doi: 10.1111/prd.12373.
  51. Hardman, W.E. Diet components can suppress inflammation and reduce cancer risk. Res. Pract. 2014, 8, 233-240. doi: 10.4162/nrp.2014.8.3.233.
  52. Merra, G.; Noce, A.; Marrone, G.; Cintoni, M.; Tarsitano, M.G.; Capacci, A.; De Lorenzo, A. Influence of Mediterranean Diet on Human Gut Microbiota. Nutrients 2020, 13, 7. doi: 10.3390/nu13010007.
  53. Ozdal, T.; Sela, D.A.; Xiao, J.; Boyacioglu, D.; Chen, F.; Capanoglu, E. The reciprocal interactions between polyphenols and gut microbiota and effects on bioaccessibility. Nutrients 2016, 8, 78. doi: 10.3390/nu8020078.
  54. Koudoufio, M.; Desjardins, Y.; Feldman, F.; Spahis, S.; Delvin, E.; Levy, E. Insight into polyphenol and gut microbiota crosstalk: are their metabolites the key to understand protective effects against metabolic disorders ? Antioxidants 2020, 9, 982.
  55. Seyed Hameed, A.S.; Rawat, P.S.; Meng, X.; Liu, W. Biotransformation of dietary phytoestrogens by gut microbes: A review on bidirectional interaction between phytoestrogen metabolism and gut microbiota. Adv. 2020, 43, 107576. doi: 10.1016/j.biotechadv.2020.107576.
  56. Senizza, A.; Rocchetti, G.; Mosele, J.I.; Patrone, V.; Callegari, M.L.; Morelli, L.; Lucini, L. Lignans and gut microbiota: an interplay revealing potential health implications. Molecules 2020, 25, 5709. doi: 10.3390/molecules25235709.
  57. Andreux, P.A.; Blanco-Bose, W.; Ryu, D.; Burdet, F.; Ibberson, M.; Aebischer, P.; Auwerx, J.; Singh, A.; Rinsch, C. The mitophagy activator urolithin A is safe and induces a molecular signature of improved mitochondrial and cellular health in humans. Metab. 2019, 1, 595-603. doi: 10.1038/s42255-019-0073-4.
  58. Man, A.W.C.; Zhou, Y.; Xia, N.; Li, H. Involvement of gut microbiota, microbial metabolites and interaction with polyphenol in host immunometabolism. Nutrients 2020, 12, doi: 10.3390/nu12103054.
  59. Chalons, P.; Courtaut, F.; Limagne, E.; Chalmin, F.; Cantos-Villar, E.; Richard, T.; Auger, C.; Chabert, P.; Schini-Kerth, V.; Ghiringhelli, F.; Aires, V.; Delmas, D. Red Wine Extract Disrupts Th17 Lymphocyte Differentiation in a Colorectal Cancer Context. Nutr. Food. Res. 2020, 64, e1901286. doi: 10.1002/mnfr.201901286.
  60. Wang, M.; Firrman, J.; Liu, L.; Yam, K. A review on flavonoid apigenin: dietary intake, adme, antimicrobial effects, and interactions with human gut microbiota. Res. Int. 2019, 2019, 7010467. doi: 10.1155/2019/7010467.
  61. Hanske, L.; Loh, G.; Sczesny, S.; Blaut, M.; Braune, A. The bioavailability of apigenin-7-glucoside is influenced by human intestinal microbiota in rats. Nutr. 2009, 139, 1095-1102. doi: 10.3945/jn.108.102814.
  62. Wang, M.; Firrman, J.; Zhang, L.; Arango-Argoty, G.; Tomasula, P.; Liu, L.; Xiao, W.; Yam, K. Apigenin impacts the growth of the gut microbiota and alters the gene expression of enterococcus. Molecules 2017, 22, 1292. doi: 10.3390/molecules22081292.
  63. Zam; W. Gut microbiota as a prospective therapeutic target for curcumin: a review of mutual influence. J Nutr. Metab. 2018, 2018, 1367984. doi: 10.1155/2018/1367984.
  64. Pluta, R.; Januszewski, S.; Ułamek-Kozioł, M. Mutual two-way interactions of curcumin and gut microbiota. J. Mol. Sci. 2020, 21, 1055. doi: 10.3390/ijms21031055.
  65. Liu, Z.; de Bruijn, W.J.C.; Bruins, M.E.; Vincken, J.P. Reciprocal Interactions between Epigallocatechin-3-gallate (EGCG) and Human Gut Microbiota In vitro. Agric. Food. Chem. 2020, 68, 9804-9815. doi: 10.1021/acs.jafc.0c03587.
  66. Murota, K.; Nakamura, Y.; Uehara, M. Flavonoid metabolism: the interaction of metabolites and gut microbiota. Biotechnol. Biochem. 2018, 82, 600-610. doi: 10.1080/09168451.2018.1444467.
  67. Di Pede, G.; Bresciani, L.; Calani, L.; Petrangolini, G.; Riva, A; Allegrini, P.; Del Rio, D.; Mena, P. The human microbial metabolism of quercetin in different formulations: an in vitro evaluation. Foods 2020, 9, 1121. doi: 10.3390/foods9081121.
  68. Firrman, J.; Liu, L.; Zhang, L.; Arango Argoty, G.; Wang, M.; Tomasula, P.; Kobori, M.; Pontious, S.; Xiao, W. The effect of quercetin on genetic expression of the commensal gut microbes Bifidobacterium catenulatum, Enterococcus caccae and Ruminococcus gauvreauii. Anaerobe 2016, 42, 130-141. doi: 10.1016/ j.anaerobe.2016.10.004.
  69. Birt, D.F.; Mitchell, D.; Gold, B.; Pour, P.; Pinch, H.C. Inhibition of ultraviolet light induced skin carcinogenesis in SKH-1 mice by apigenin, a plant flavonoid. Anticancer Res. 1997, 17, 85-91.
  70. Ren, W.; Qiao, Z.; Wang, H.; Zhu, L.; Zhang, L. Flavonoids: promising anticancer agents. Res. Rev. 2003, 23, 519-534. doi: 10.1002/med.10033.
  71. Wang, I.K.; Lin-Shiau, S.Y.; Lin, J.K. Induction of apoptosis by apigenin and related flavonoids through cytochrome c release and activation of caspase-9 and caspase-3 in leukaemia HL-60 cells. J. Cancer. 1999, 35, 1517-1525.
  72. Caltagirone, S.; Rossi, C.; Poggi, A.; Ranelletti, FO.; Natali, PG.; Brunetti, M.; Aiello, FB.; Piantelli, M. Flavonoids apigenin and quercetin inhibit melanoma growth and metastatic potential. J. Cancer. 2000, 87, 595-600.
  73. Yin, F.; Wakino, S.; Liu, Z.; Kim, S.; Hsueh, W.A.; Collins, A.R.; Van Herle, A.J.; Law, R.E. Troglitazone inhibits growth of MCF-7 breast carcinoma cells by targeting G1 cell cycle regulators. Biophys. Res. Commun. 2001, 286, 916-922. doi: 10.1006/bbrc.2001.5491.
  74. Zhu, Y.; Mao, Y.; Chen, H.; Lin, Y.; Hu, Z.; Wu, J.; Xu, X.; Xu, X.; Qin, J.; Xie, L. Apigenin promotes apoptosis, inhibits invasion and induces cell cycle arrest of T24 human bladder cancer cells. Cancer Cell. Int. 2013, 13, doi: 10.1186/1475-2867-13-54.
  75. Zhang, L.; Cheng, X.; Gao, Y.; Zheng, J.; Xu, Q.; Sun, Y.; Guan, H.; Yu, H.; Sun, Z. Apigenin induces autophagic cell death in human papillary thyroid carcinoma BCPAP cells. Food Funct. 2015, 6, 3464-3472. doi: 10.1039/c5fo00671f.
  76. Banerjee, K.; Banerjee, S.; Mandal, M. Enhanced chemotherapeutic efficacy of apigenin liposomes in colorectal cancer based on flavone-membrane interactions, Colloid Interface Sci. 2017, 491, 98-110, https://doi.org/10.1016/j.jcis.2016.12.025
  77. Li, J.; Yu, B.; Deng, P.; Cheng, Y.; Yu, Y.; Kevork, K.; Ramadoss, S.; Ding, X.; Li, X.; Wang, CY. KDM3 epigenetically controls tumorigenic potentials of human colorectal cancer stem cells through Wnt/β-catenin signalling. Commun. 2017, 8, 15146. doi: 10.1038/ncomms15146. Erratum in: Nat Commun. 2019, 10, 5020.
  78. Xu, M.; Wang, S.; Song, YU.; Yao, J.; Huang, K.; Zhu, X. Apigenin suppresses colorectal cancer cell proliferation, migration and invasion via inhibition of the Wnt/β-catenin signaling pathway. Lett. 2016, 11, 3075-3080. doi: 10.3892/ol.2016.4331.
  79. Dai, J.; Van Wie, PG.; Fai, LY.; Kim, D.; Wang, L.; Poyil, P.; Luo, J.; Zhang, Z. Downregulation of NEDD9 by apigenin suppresses migration, invasion, and metastasis of colorectal cancer cells. Appl. Pharmacol. 2016, 311, 106-112. doi: 10.1016/j.taap.2016.09.016.
  80. Lee, Y.; Sung, B.; Kang, Y.J.; Kim, DH.; Jang, J.Y.; Hwang, S.Y.; Kim, M.; Lim, H.S.; Yoon, J.H.; Chung, H.Y.; Kim, N.D. Apigenin-induced apoptosis is enhanced by inhibition of autophagy formation in HCT116 human colon cancer cells. J. Oncol. 2014, 44, 1599-606. doi: 10.3892/ijo.2014.2339.
  81. Pandey, M.; Kaur, P.; Shukla, S.; Abbas, A.; Fu, P.; Gupta, S. Plant flavone apigenin inhibits HDAC and remodels chromatin to induce growth arrest and apoptosis in human prostate cancer cells: in vitro and in vivo study. Carcinogenesis. 2012, 51, 952–962. https://doi.org/10.1002/mc.20866.
  82. Zhong, Y.; Krisanapun, C.; Lee, S.H.; Nualsanit, T.; Sams, C.; Peungvicha, P.; Baek, S.J. Molecular targets of apigenin in colorectal cancer cells: involvement of p21, NAG-1 and p53. J. Cancer 2010, 46, 3365–3374. https://doi.org/10.1016/j.ejca.2010.07.007
  83. Chunhua, L.; Donglan, L.; Xiuqiong, F.; Lihua, Z.; Qin, F.; Yawei, L.; Liang, Z.; Ge, W.; Linlin, J.; Ping, Z.; Kun, L.; Xuegang, S. Apigenin up-regulates transgelin and inhibits invasion and migration of colorectal cancer through decreased phosphorylation of AKT. Nutr. Biochem. 2013, 24, 1766-1775. doi: 10.1016/j.jnutbio.2013.03.006.
  84. Aggarwal, B.B.; Sung, B. Pharmacological basis for the role of curcumin in chronic diseases: an age-old spice with modern targets. Trends Pharmacol. Sci. 2009, 30, 85-94. doi: 10.1016/j.tips.2008.11.002.
  85. López-Lázaro, M. Anticancer and carcinogenic properties of curcumin: considerations for its clinical development as a cancer chemopreventive and chemotherapeutic agent. Nutr. Food Res. 2008, 52,S103-S127. doi: 10.1002/mnfr.200700238.
  86. Aggarwal, B.B.; Harikumar, K.B. Potential therapeutic effects of curcumin, the anti-inflammatory agent, against neurodegenerative, cardiovascular, pulmonary, metabolic, autoimmune and neoplastic diseases. J. Biochem. Cell. Biol. 2009, 41, 40-59.
  87. Söderlund, S.; Brandt, L.; Lapidus, A.; Karlén, P.; Broström, O.; Löfberg, R.; Ekbom, A.; Askling, J. Decreasing time-trends of colorectal cancer in a large cohort of patients with inflammatory bowel disease. 2009, 136, 1561-1567. doi: 10.1053/j.gastro.2009.01.064.
  88. Dempe, J.S.; Scheerle, R.K.; Pfeiffer, E.; Metzler, M. Metabolism and permeability of curcumin in cultured Caco-2 cells. Nutr. Food Res. 2013, 57, 1543-1549. doi: 10.1002/mnfr.201200113.
  89. Kuttan, G.; Kumar, K.B.; Guruvayoorappan, C.; Kuttan, R. Antitumor, anti-invasion, and antimetastatic effects of curcumin. Exp. Med. Biol. 2007, 595, 173-184. doi: 10.1007/978-0-387-46401-5_6.
  90. Bartik, L.; Whitfield, GK.; Kaczmarska, M.; Lowmiller, C.L.; Moffet, E.W.; Furmick, J.K.; Hernandez, Z.; Haussler, C.A.; Haussler, M.R.; Jurutka, P.W. Curcumin: a novel nutritionally derived ligand of the vitamin D receptor with implications for colon cancer chemoprevention. Nutr. Biochem. 2010, 21, 1153-1161. doi: 10.1016/j.jnutbio.2009.09.012.
  91. Kunnumakkara, A.B.; Diagaradjane, P.; Anand, P.; Harikumar, K.B.; Deorukhkar, A.; Gelovani, J.; Guha, S.; Krishnan, S.; Aggarwal, B.B. Curcumin sensitizes human colorectal cancer to capecitabine by modulation of cyclin D1, COX-2, MMP-9, VEGF and CXCR4 expression in an orthotopic mouse model. J. Cancer 2009, 125, 2187-97. doi: 10.1002/ijc.24593. Erratum in: Int. J. Cancer. 2010, 126, 799.
  92. Pricci, M.; Girardi, B.; Giorgio, F.; Losurdo, G.; Ierardi, E.; Di Leo, A. Curcumin and Colorectal Cancer: From Basic to Clinical Evidences. J. Mol. Sci. 2020, 21, 2364. doi: 10.3390/ijms21072364.
  93. Shoba, G.; Joy, D.; Joseph, T.; Majeed, M.; Rajendran, R.; Srinivas, PS. Influence of piperine on the pharmacokinetics of curcumin in animals and human volunteers. Med. 1998, 64, 353-356. doi: 10.1055/s-2006-957450.
  94. Di Silvestro, RA.; Joseph, E.; Zhao, S.; Bomser, J. Diverse effects of a low dose supplement of lipidated curcumin in healthy middle aged people. J. 2012, 11, 79. doi: 10.1186/1475-2891-11-79.
  95. Wang, X.; Wang, Q.; Ives, K.L.; Evers, B.M. Curcumin inhibits neurotensin-mediated interleukin-8 production and migration of HCT116 human colon cancer cells. Cancer Res. 2006, 12, 5346-5355. doi: 10.1158/1078-0432.CCR-06-0968.
  96. Shakibaei, M.; Mobasheri, A.; Lueders, C.; Busch, F.; Shayan, P.; Goel, A. Curcumin enhances the effect of chemotherapy against colorectal cancer cells by inhibition of NF-κB and Src protein kinase signaling pathways. PLoS One 2013, 8, e57218. doi: 10.1371/journal.pone.0057218.
  97. Bolat, ZB.; Islek, Z.; Demir, B.N.; Yilmaz, E.N.; Sahin, F.; Ucisik, M.H. Curcumin- and Piperine-Loaded Emulsomes as Combinational Treatment Approach Enhance the Anticancer Activity of Curcumin on HCT116 Colorectal Cancer Model. Bioeng Biotechnol. 2020, 8, 50. doi: 10.3389/fbioe.2020.00050.
  98. Zhang, C.; He, L.J.; Ye, H.Z.; Liu, D.F.; Zhu, Y.B.; Miao, D.D.; Zhang, S.P.; Chen, Y.Y., Jia, Y.W.; Shen, J.; Liu, X.P. Nrf2 is a key factor in the reversal effect of curcumin on multidrug resistance in the HCT‑8/5‑Fu human colorectal cancer cell line. Med. Rep. 2018, 18, 5409–5416. https://doi.org/10.3892/mmr.2018.9589.
  99. Villegas, I.; Sánchez-Fidalgo, S.; de la Lastra, C.A. Chemopreventive effect of dietary curcumin on inflammation-induced colorectal carcinogenesis in mice. Nutr. Food. Res. 2011, 55, 259-267. doi: 10.1002/mnfr.201000225.
  100. Kunnumakkara, A.B.; Diagaradjane, P.; Guha, S.; Deorukhkar, A.; Shentu, S.; Aggarwal, B.B.; Krishnan, S. Curcumin sensitizes human colorectal cancer xenografts in nude mice to gamma-radiation by targeting nuclear factor-kappaB-regulated gene products. Cancer Res. 2008, 14, 2128-2136. doi: 10.1158/1078-0432.CCR-07-4722.
  101. Xiang, L.P.; Wang, A.; Ye, J.H.; Zheng, X.Q.; Polito, C.A.; Lu, J.L.; Li, Q.S.; Liang, Y.R. Suppressive Effects of Tea Catechins on Breast Cancer. Nutrients 2016, 8, doi: 10.3390/nu8080458.
  102. Cromie, M.M.; Gao, W. (2015). Epigallocatechin-3-gallate enhances the therapeutic effects of leptomycin B on human lung cancer a549 cells. Med. Cell. Longev. 2015, 2015, 217304. doi: 10.1155/2015/217304.
  103. Kumazoe, M.; Takai, M.; Hiroi, S.; Takeuchi, C.; Yamanouchi, M.; Nojiri, T.; Onda, H.; Bae, J.; Huang, Y.; Takamatsu, K.; Yamashita, S.; Yamada, S.; Kangawa, K.; Takahashi, T.; Tanaka, H.; Tachibana, H. PDE3 inhibitor and EGCG combination treatment suppress cancer stem cell properties in pancreatic ductal adenocarcinoma. Rep. 2017, 7, 1917. doi: 10.1038/s41598-017-02162-9.
  104. Bravi, F.; La Vecchia, C.; Turati, F. Green tea and liver cancer. Hepatobiliary Surg Nutr. 2017, 6, 127-129. doi: 10.21037/hbsn.2017.03.07.
  105. Shirakami, Y.; Shimizu, M.; Tsurumi, H.; Hara, Y.; Tanaka, T.; Moriwaki, H. EGCG and Polyphenon E attenuate inflammation-related mouse colon carcinogenesis induced by AOM plus DDS. Med. Rep. 2008, 1, 355-361.
  106. Shirakami, G.; Magaribuchi, T.; Shingu, K.; Suga, S.; Tamai, S.; Nakao, K.; Mori, K. Positive end-expiratory pressure ventilation decreases plasma atrial and brain natriuretic peptide levels in humans. Analg. 1993, 77, 1116-1121. doi: 10.1213/00000539-199312000-00006.
  107. Wubetu, G.Y.; Shimada, M.; Morine, Y.; Ikemoto, T.; Ishikawa, D.; Iwahashi, S.; Yamada, S.; Saito, Y.; Arakawa, Y.; Imura, S. Epigallocatechin gallate hinders human hepatoma and colon cancer sphere formation. Gastroenterol. Hepatol. 2016, 31, 256-264. doi: 10.1111/jgh.13069.
  108. Alam, M.N.; Almoyad, M.; Huq, F. Polyphenols in Colorectal Cancer: Current State of Knowledge including Clinical Trials and Molecular Mechanism of Action. Res. Int. 2018, 2018, 4154185. doi: 10.1155/2018/4154185.
  109. Ogawa, K.; Hara, T.; Shimizu, M.; Nagano, J.; Ohno, T.; Hoshi, M.; Ito, H.; Tsurumi, H.; Saito, K.; Seishima, M.; Moriwaki, H. (-)-Epigallocatechin gallate inhibits the expression of indoleamine 2,3-dioxygenase in human colorectal cancer cells. Lett. 2012, 4, 546-550. doi: 10.3892/ol.2012.761
  110. Chen, Y.; Wang, X.Q.; Zhang, Q.; Zhu, J.Y.; Li, Y.; Xie, C.F.; Li, X.T.; Wu, J.S.; Geng, S.S.; Zhong, C.Y.; Han, H.Y. (-)-Epigallocatechin-3-Gallate Inhibits Colorectal Cancer Stem Cells by Suppressing Wnt/β-Catenin Pathway. Nutrients 2017, 9, doi: 10.3390/nu9060572.
  111. Zhu, W.; Li, M.C.; Wang, F.R.; Mackenzie, G.G.; Oteiza, P.I. The inhibitory effect of ECG and EGCG dimeric procyanidins on colorectal cancer cells growth is associated with their actions at lipid rafts and the inhibition of the epidermal growth factor receptor signaling. Pharmacol. 2020, 175, 113923. doi: 10.1016/j.bcp.2020.113923.
  112. Jin, H.; Gong, W.; Zhang, C.; Wang, S. Epigallocatechin gallate inhibits the proliferation of colorectal cancer cells by regulating Notch signaling. Targets Ther. 2013, 6, 145-153. doi: 10.2147/OTT.S40914
  113. Md Nesran, Z.N.; Shafie, N.H.; Ishak, A.H.; Mohd Esa, N.; Ismail, A.; Md Tohid, SF. Induction of Endoplasmic Reticulum Stress Pathway by Green Tea Epigallocatechin-3-Gallate (EGCG) in Colorectal Cancer Cells: Activation of PERK/p-eIF2a/ATF4 and IRE1a. Res. Int. 2019, 2019, 3480569. doi: 10.1155/2019/3480569.
  114. Hu, F.; Wei, F.; Wang, Y.; Wu, B.; Fang, Y.; Xiong, B. EGCG synergizes the therapeutic effect of cisplatin and oxaliplatin through autophagic pathway in human colorectal cancer cells. Pharmacol. Sci. 2015, 128, 27-34. doi: 10.1016/j.jphs.2015.04.003.
  115. Enkhbat, T.; Nishi, M.; Yoshikawa, K.; Jun, H.; Tokunaga, T.; Takasu, C.; Kashihara, H.; Ishikawa, D.; Tominaga, M.; Shimada, M. Epigallocatechin-3-gallate Enhances Radiation Sensitivity in Colorectal Cancer Cells Through Nrf2 Activation and Autophagy. Anticancer Res. 2018, 38, 6247-6252. doi: 10.21873/anticanres.12980
  116. Maruyama, T.; Murata, S.; Nakayama, K.; Sano, N.; Ogawa, K.; Nowatari, T.; Tamura, T.; Nozaki, R.; Fukunaga, K.; Ohkohchi, N. (-)-Epigallocatechin-3-gallate suppresses liver metastasis of human colorectal cancer. Rep. 2014, 31, 625-633. doi: 10.3892/or.2013.2925
  117. Kim, W.K.; Bang, M.H.; Kim, E.S.; Kang, N.E.; Jung, K.C.; Cho, H.J.; Park, J.H. Quercetin decreases the expression of ErbB2 and ErbB3 proteins in HT-29 human colon cancer cells. Nutr. Biochem. 2005, 16, 155-162.
  118. Del Follo-Martinez, A.; Banerjee, N.; Li, X.; Safe, S.; Mertens-Talcott, S. Resveratrol and quercetin in combination have anticancer activity in colon cancer cells and repress oncogenic microRNA-27a. Cancer. 2013, 65, 494-504.
  119. Yang, L.; Liu, Y.; Wang, M.; Qian, Y.; Dong, X.; Gu, H.; Wang, H.; Guo, S.; Hisamitsu, T. Quercetin-induced apoptosis of HT-29 colon cancer cells via inhibition of the Akt-CSN6-Myc signaling axis. Med. Rep. 2016, 14, 4559-4566.
  120. Jin, W.; Han, H.; Ma, L.; Zhou, H.; Zhao, C. The chemosensitization effect of quercetin on cisplatin induces the apoptosis of human colon cancer HT-29 cell line. J. Clin. Med. 2016, 9, 2285-2292.
  121. Psahoulia, F.H.; Drosopoulos, K.G.; Doubravska, L.; Andera, L.; Pintzas, A. Quercetin enhances TRAIL-mediated apoptosis in colon cancer cells by inducing the accumulation of death receptors in lipid rafts. Cancer Ther. 2007, 6, 2591-2599.
  122. Kim, H.J.; Kim, S.K.; Kim, B.S.; Lee, S.H.; Park, Y.S.; Park, B.K.; Kim, S.J.; Kim, J.; Choi, C.; Kim, J.S.; Cho, S.D.; Jung, J.W.; Roh, K.H.; Kang, K.S.; Jung, J.Y. Apoptotic effect of quercetin on HT-29 colon cancer cells via the AMPK signaling pathway. Agric. Food. Chem. 2010, 58, 8643-8650.
  123. Atashpour, S.; Fouladdel, S.; Movahhed, T.K.; Barzegar, E.; Ghahremani, M.H.; Ostad, S.N.; Azizi, E. Quercetin induces cell cycle arrest and apoptosis in CD133(+) cancer stem cells of human colorectal HT29 cancer cell line and enhances anticancer effects of doxorubicin. Iran J. Basic. Med. Sci. 2015, 18, 635-643.
  124. van Erk, M.J.; Roepman, P.; van der Lende, T.R.; Stierum, R.H.; Aarts, J.M.; van Bladeren, P.J.; van Ommen, B. Integrated assessment by multiple gene expression analysis of quercetin bioactivity on anticancer-related mechanisms in colon cancer cells in vitro. J. Nutr. 2005, 44, 143-156.
  125. Han, M.; Song, Y.; Zhang, X. Quercetin Suppresses the Migration and Invasion in Human Colon Cancer Caco-2 Cells Through Regulating Toll-like Receptor 4/Nuclear Factor-kappa B Pathway. Pharmacogn Mag. 2016, 12, S237-44.
  126. Zhang, X.A.; Zhang, S.; Yin, Q.; Zhang, J. Quercetin induces human colon cancer cells apoptosis by inhibiting the nuclear factor-kappa B Pathway. Mag. 2015, 11, 404-409.
  127. Shan, B.E.; Wang, M.X.; Li, R.Q. Quercetin inhibit human SW480 colon cancer growth in association with inhibition of cyclin D1 and survivin expression through Wnt/beta-catenin signaling pathway. Cancer Invest. 2009, 27, 604-612.
  128. Richter, M.; Ebermann, R.; Marian, B. Quercetin-induced apoptosis in colorectal tumor cells: possible role of EGF receptor signaling. Cancer. 1999, 34, 88-99.
  129. Özsoy, S.; Becer, E.; Kabaday, H.; Vatansever, H.S.; Yücecan, S. Quercetin-Mediated Apoptosis and Cellular Senescence in Human Colon Cancer. Anticancer Agents Med. Chem. 2020, 20, 1387-1396.
  130. Kee, J.Y.; Han, Y.H.; Kim, D.S.; Mun, J.G.; Park, J.; Jeong, M.Y.; Um, J.Y.; Hong, S.H. Inhibitory effect of quercetin on colorectal lung metastasis through inducing apoptosis, and suppression of metastatic ability. Phytomedicine 2016, 23, 1680-1690.
  131. Mutoh, M.; Takahashi, M.; Fukuda, K.; Komatsu, H.; Enya, T.; Matsushima-Hibiya, Y.; Mutoh, H.; Sugimura, T.; Wakabayashi, K. Suppression by flavonoids of cyclooxygenase-2 promoter-dependent transcriptional activity in colon cancer cells: structure-activity relationship. J. Cancer Res. 2000, 91, 686-691.
  132. Xavier, C.P.; Lima, C.F.; Rohde, M.; Pereira-Wilson, C. Quercetin enhances 5-fluorouracil-induced apoptosis in MSI colorectal cancer cells through p53 modulation. Cancer Chemother. Pharmacol. 2011, 68, 1449-1457.
  133. Pampaloni, B.; Palmini, G.; Mavilia, C.; Zonefrati, R.; Tanini, A.; Brandi, M.L. In vitro effects of polyphenols on colorectal cancer cells. World J. Gastrointest. Oncol. 2014, 6, 289-300.
  134. Li, Y.; Wang, Z.; Jin, J.; Zhu, S.X.; He, G.Q.; Li, S.H.; Wang, J.; Cai, Y. Quercetin pretreatment enhances the radiosensitivity of colon cancer cells by targeting Notch-1 pathway. Biophys Res. Commun. 2020, 523, 947-953.
  135. Wang, Y.; Jin, H.Y.; Fang, M.Z.; Wang, X.F.; Chen, H.; Huang, S.L.; Kong, D.S.; Li, M.; Zhang, X.; Sun, Y.; Wang, S.M. Epigallocatechin gallate inhibits dimethylhydrazine-induced colorectal cancer in rats. World J. Gastroenterol. 2020, 26, 2064-2081. doi: 10.3748/wjg.v26.i17.2064
  136. Darband, S.G.; Sadighparvar, S.; Yousefi, B.; Kaviani, M.; Ghaderi-Pakdel, F.; Mihanfar, A.; Rahimi, Y.; Mobaraki, K.; Majidinia, M. Quercetin attenuated oxidative DNA damage through NRF2 signaling pathway in rats with DMH induced colon carcinogenesis. Life Sci. 2020, 253,
  137. Qi, J.; Yu, J.; Li, Y.; Luo, J.; Zhang, C.; Ou, S.; Zhang, G.; Yang, X.; Peng, X. Alternating consumption of β-glucan and quercetin reduces mortality in mice with colorectal cancer. Food Sci. Nutr. 2019, 7, 3273-3285.
  138. Cruz-Correa, M.; Shoskes, DA.; Sanchez, P.; Zhao, R.; Hylind, L.M.; Wexner, S.D.; Giardiello, F.M. Combination treatment with curcumin and quercetin of adenomas in familial adenomatous polyposis. Gastroenterol. Hepatol. 2006, 4, 1035-1038
  139. Guon, T.E.; Chung, H.S. Hyperoside and rutin of Nelumbo nucifera induce mitochondrial apoptosis through a caspase-dependent mechanism in HT-29 human colon cancer cells. Lett. 2016, 11, 2463-2470.
  140. Nafees, S.; Mehdi, S.H.; Zafaryab, M.; Zeya, B.; Sarwar, T.; Rizvi, M.A. Synergistic Interaction of Rutin and Silibinin on Human Colon Cancer Cell Line. Med. Res. 2018, 49, 226-234.
  141. Ben Sghaier, M.; Pagano, A.; Mousslim, M.; Ammari, Y.; Kovacic, H.; Luis, J. Rutin inhibits proliferation, attenuates superoxide production and decreases adhesion and migration of human cancerous cells. Pharmacother. 2016, 84,1972-1978.
  142. Kuntz, S.; Wenzel, U.; Daniel, H. Comparative analysis of the effects of flavonoids on proliferation, cytotoxicity, and apoptosis in human colon cancer cell lines. J. Nutr. 1999, 38, 133-142.
  143. Alonso-Castro, A.J.; Domínguez, F.; García-Carrancá, A. Rutin exerts antitumor effects on nude mice bearing SW480 tumor. Med. Res. 2013, 44, 346-351.
  144. Gautam, R.; Singh, M.; Gautam, S.; Rawat, J.K.; Saraf, S.A.; Kaithwas, G. Rutin attenuates intestinal toxicity induced by Methotrexate linked with anti-oxidative and anti-inflammatory effects. BMC Complement Altern. Med. 2016, 16,
  145. Fideles, L.S.; de Miranda, J.A.L.; Martins, C.D.S.; Barbosa, M.L.L.; Pimenta, H.B.; Pimentel, P.V.S.; Teixeira, C.S.; Scafuri, M.A.S.; Façanha, S.O.; Barreto, J.E.F.; Carvalho, P.M.M.; Scafuri, A.G.. Araújo, J.L.; Rocha, J.A.; Vieira, I.G.P.; Ricardo N.M.P.S.; da Silva Campelo, M.; Ribeiro, M.E.N.P.; de Castro Brito, G.A.; Cerqueira, G.S. Role of Rutin in 5-Fluorouracil-Induced Intestinal Mucositis: Prevention of Histological Damage and Reduction of Inflammation and Oxidative Stress. Molecules 2020, 25,
  146. Kwon, K.H.; Murakami, A.; Tanaka, T.; Ohigashi, H. Dietary rutin, but not its aglycone quercetin, ameliorates dextran sulfate sodium-induced experimental colitis in mice: attenuation of pro-inflammatory gene expression. Pharmacol. 2005, 69, 395-406.
  147. Jang, M.; Cai, L.; Udeani, G.O.; Slowing, K.V.; Thomas, C.F.; Beecher, C.W.; Fong, H.H.; Farnsworth, N.R.; Kinghorn, A.D.; Mehta, R.G.; Moon, R.C.; Pezzuto, J.M. Cancer chemopreventive activity of resveratrol, a natural product derived from grapes. Science 1997, 275, 218-220.
  148. Vervandier-Fasseur, D.; Latruffe, N. Cancer prevention potential of resveratrol. Molecules 2019, 24, 4506. doi: 10.3390/molecules24244506.
  149. Levi, F.; Pasche, C.; Lucchini, F.; Ghidoni, R.; Ferraroni, M.; La Vecchia, C. Resveratrol and breast cancer risk. J. Cancer Prev. 2005, 14, 139-142.
  150. Patel, K.R.; Brown, V.A.; Jones, D.J.; Britton, R.G.; Hemingway, D.; Miller, A.S.; West, K.P.; Booth, T.D.; Perloff, M.; Crowell, J.A.; Brenner, D.E.; Steward, W.P.; Gescher, A.J.; Brown, K. Clinical pharmacology of resveratrol and its metabolites in colorectal cancer patients. Cancer Res. 2010, 70, 7392-7399.
  151. Semba, R.D.; Ferrucci, L.; Bartali, B.; Urpí-Sarda, M.; Zamora-Ros, R.; Sun, K.; Cherubini, A.; Bandinelli, S.; Andres-Lacueva, C. Resveratrol levels and all-cause mortality in older community-dwelling adults. JAMA Intern. Med. 2014, 174, 1077-1084.
  152. Provinciali, M.; Re, F.; Donnini, A.; Orlando, F.; Bartozzi, B.; Di Stasio, G.; Smorlesi, A. Effect of resveratrol on the development of spontaneous mammary tumors in HER-2/neu transgenic mice. J. Cancer. 2005, 115, 36-45.
  153. Busquets, S.; Ametller, E.; Fuster, G.; Olivan, M.; Raab, V.; Argilés, J.M.; López-Soriano, F.J. Resveratrol, a natural diphenol, reduces metastatic growth in an experimental cancer Cancer Lett. 2007, 245, 144-148.
  154. Aires, V.; Limagne, ; Cotte, A.; Latruffe, N.; Ghiringhelli, F.; Delmas, D. Resveratrol metabolites inhibit human metastatic colon cancer cells progression and synergizes with chemotherapeutic drugs to induce cell death. Mol. Nutr. Food Res. 2013, 57, 1170-1181.
  155. Abraham, S.K.; Khandelwal, N.; Hintzsche, H.; Stopper, H. Antigenotoxic effects of resveratrol: assessment of in vitro and in vivo response. Mutagenesis 2016, 31, 27-33.
  156. Delmas, D.; Rébé, C.; Lacour, S.; Filomenko, R.; Athias, A.; Gambert, P.; Cherkaoui-Malki, M.; Jannin, B.; Dubrez-Daloz, L.; Latruffe,; Solary, E. Resveratrol-induced apoptosis is associated with Fas redistribution in the rafts and the formation of a death-inducing signaling complex in colon cancer cells. J. Biol. Chem. 2003, 278, 41482-41490.
  157. Aires, V.; Colin, D.J.; Agnès, A.; Di Pietro, A.; Heydel, J.M.; Artur, Y.; Latruffe; N.; Delmas, D. P-glycoprotein 1 affects chemoactivities of resveratrol against human colorectal cancer cells. Nutrients 2019, 11, 2098.
  158. Colin, D.; Limagne, E.; Jeanningros, S.; Jacquel, A.; Lizard, G.; Athias, A.; Gambert, P.; Hichami, A.; Latruffe,; Solary, E.; Delmas, D. Endocytosis of resveratrol via lipid rafts and activation of downstream signaling pathways in cancer cells. Cancer Prev. Res. 2011, 4, 1095-1106.
  159. Tili, E.; Michaille, J.-J.; Alder, H.; Volinia, S.; Delmas, D.; Latruffe,; Croce, C.M. Resveratrol modulates the levels of microRNAs targeting genes encoding tumor-suppressors and effectors of TGFβ signaling pathway in SW480 cells. Biochem. Pharmacol. 2010, 80, 2057-2065.
  160. Ho, Y.; S.H. Yang, Y.C.; Chin, Y.T.; Chou, S.Y.; Chen, Y.R.; Shih, Y.J; Whang-Peng, J.; Changou, C.A.; Liu, H.L.; Lin, S.J.; Tang, H.Y.; Lin, H.Y.; Davis, P.J. Resveratrol inhibits human leiomyoma cell proliferation via crosstalk between integrin αvβ3 and IGF-1R. Food Chem. Toxicol. 2018, 120, 346-355.
  161. Frazzi, R.; Valli, R.; Tamagnini, I.; Casali, B.; Latruffe,; Merli, F. Resveratrol-mediated apoptosis of hodgkin lymphoma cells involves SIRT1 inhibition and FOXO3a hyperacetylation. Int. J. Cancer, 2013, 132, 1013-1021.
  162. Poschner, S.; Maier-Salamon, A.; Thalhammer, T.; Jäger, W. Resveratroland other dietary polyphenols are inhibitors of estrogen metabolism in humanbreastcancerJ. Steroid Biochem. Mol. Biol. 2019, 190, 11-18. doi: 10.1016/j.jsbmb.2019.03.001.
  163. Baarine, M.; Thandapilly, S.J.; Louis, X.L.; Mazué, F.; Yu, L.; Delmas, D.; Netticadan, T.; Lizard, G.; Latruffe, Pro-apoptotic versus anti-apoptotic properties of dietary resveratrol on tumoral and normal cardiac cells. Genes Nutr. 2011, 6, 161-169.
  164. Marel, A.K.; Lizard, G.; Izard, J.-C.; Latruffe,; Delmas, D. Inhibitory effects of trans-resveratrol analogs molecules on the proliferation and the cell cycle progression of human colon tumoral cells. Mol. Nutr. Food Res. 2008, 52, 538-548.
  165. Cilibrasi, C.; Riva, G.; Romano, G.; Cadamuro, M.; Bazzoni, R.; Butta, V.; Paoletta, L.; Dalprà, L.; Strazzabosco, M.; Lavitrano, M.; Giovannoni, R.; Bentivegna, A. Resveratrol Impairs Glioma Stem Cells Proliferation and Motility by Modulating the Wnt Signaling Pathway. PLoS One. 2017, 12, e0169854.
  166. Yousef, M.; Vlachogiannis, IA.; Tsiani, E. Effects ofResveratrolagainstLungCancer: In vitro and In vivo Studies. Nutrients 2017, 9, doi: 10.3390/nu9111231.
  167. Ferrer, P.; Asensi, M.; Segarra, R.; Ortega.; Benlloch, M.; Obrador, E.; Varea, M.T.; Asensio, G.; Jordá, L.; Estrela, J.M. Association between pterostilbene and quercetin inhibits metastatic activity of B16 melanoma. Neoplasia 2005, 7, 37-47.
  168. Singh, S.K.; Banerjee, S.; Acosta, E.P.; Lillard, J.W.; Singh, R. Resveratrol induces cell cycle arrest and apoptosis with docetaxel in prostate cancer cells via a p53/ p21WAF1/CIP1 and p27KIP1 pathway. Oncotarget 2017, 8, 17216-1722.
  169. Mertens-Talcott, S.U.; Percival, S.S. Ellagic acid and quercetin interact synergistically with resveratrol in the induction of apoptosis and cause transient cell cycle arrest in human leukemia cells. Cancer Lett. 2005, 218, 141-151.
  170. Mazué, F.; Delmas, D.; Murillo, G.; Saleiro, D.; Limagne, E.; Latruffe, Differential protective effects of red wine polyphenol extracts (RWEs) on colon carcinogenesis. Food Funct. 2014, 5, 663-670.
  171. Dabrowski, W.; Siwicka-Gieroba, D.; Kotfis, K.; Zaid, S.; Terpilowska, S.; Robba, C.; Siwicki, A.K. The brain-gut axis - where are we now and how can we modulate these connections? Neuropharmacol. 2020. doi: 10.2174/1570159X18666201119155535.
  172. Westfall, S.; Pasinetti, G.M. The Gut Microbiota Links Dietary Polyphenols With Management of Psychiatric Mood Disorders. Front Neurosci. 2019, 13, 1196. doi: 10.3389/fnins.2019.01196.
  173. Shabbir, U.; Rubab, M.; Daliri, E.B.; Chelliah, R.; Javed, A.; Oh, DH. Curcumin, Quercetin, Catechins and Metabolic Diseases: The Role of Gut Microbiota. Nutrients 2021, 13, 206. doi: 10.3390/nu13010206.
  174. Sun, Q.; Cheng, L.; Zhang, X.; Wu, Z.; Weng, P. The interaction between tea polyphenols and host intestinal microorganisms: an effective way to prevent psychiatric disorders. Food Funct. 2021, 12, 952-962. doi: 10.1039/d0fo02791j.
  175. Matarazzo, I.; Toniato, E.; Robuffo, I. Psychobiome Feeding Mind: Polyphenolics in Depression and Anxiety. Top. Med. Chem. 2018, 18, 2108-2115. doi: 10.2174/1568026619666181210151348.
  176. Chung, J.Y.; Jeong, J.H.; Song, J. Resveratrol Modulates the Gut-Brain Axis: Focus on Glucagon-Like Peptide-1, 5-HT, and Gut Microbiota. Aging Neurosci. 2020, 12, 588044. doi: 10.3389/fnagi.2020.588044.
  177. Bonferoni, M.C.; Rossi, S.; Sandri, G.; Ferrari, F. Nanoparticle formulations to enhance tumor targeting of poorly soluble polyphenols with potential anticancer properties. Seminars Cancer Bio 2017, 46, 205-214.
  178. Summerlin, N.; Soo, E.; Thakur, S.; Qu, Z.; Jambhrunkar, S.; Popat, A. Resveratrol nanoformulations: Challenges and opportunities. J. Pharm. 2015, 479, 282-290.
  179. Manconi,; Manca, M.L.; Escribano-Ferrer, E.; Coma-Cros, E.M.; Biosca, A.; Lantero, E.; Fernandez-Busquets, X.; Fadda, A.M.; Caddeo, C. Nanoformulation of curcumin-loaded eudragit-nutriosomes to counteract malária infection by a dual strategy: improving anti-oxidant intestinal activity and systemic efficacy. Int. J. Pharm. 2019, 556, 82-88.
  180. Nam, J.S.; Sharma, A.R.; Nguyen, L.T.; Chakraborty, C.; Sharma, G.; Lee, S.S. Application of bioactive quercetin in oncotherapy : from nutrition to nanomedicine. Molecules, 2016, 21, 108. doi: 10.3390/molecules21010108.
  181. Yi, Z.; Chen, G.; Chen, X.; Sun, Z.; Ma, X.; Su, W.; Deng, Z.; Ma, L.; Ran, Y.; Tong, Q.; Li, X. Molecular assembly of versatile nanoparticles with epigallocatechin gallate.ACS Sustainable Chem. Eng. 2020, 8, 9833-9845.
  182. Dutta, D.; Chakraborty, A.; Mukherjee, B.; Gupta, S. Aptamer-conjugated apigenin nanoparticles to target colorectal carcinoma: a promising safe alternative of colorectal cancer chemotherapy. ACS Appl. Bio Mater, 2018, 1, 1538-1556.
  183. Gao, X.; Wang, B.L.; Wei, X. W.; Men, K.; Zheng, F.; Zhou, Y.; Zheng, Y.; Gou, M. L.; Huang, M.; Guo, G.; Huang, N.; Qian, Z. Y.; Wei, Y. Anticancer effect and mechanism of polymer micelle-​encapsulated quercetin on ovarian cancer. Nanoscale 2012, 4, 7021-7030.
  184. Xu,G.; Shi, H.; Ren, L.; Gou, H.; Gong, D.; Gao, X.; Huang, N. Enhancing the anti-colon cancer activity of quercetin by self-assembled micelles. J. Nanomedicine, 2015, 10, 2051-2063.
  185. Coimbra, M.; Isacchi, B.; van Bloois, L.; Torano, J.S.; Ket, A.; Wu, X.; Broere, F.; Metselaar, J.M.; Rijcken, C.J.F.; Storm, G.; Bilia, R.; Schiffelers, R.M. Improving solubility and chemical stability of natural compounds for medicinal use by incorporation into liposomes. J. Pharm. 2011, 416, 433-442.
  186. Feng, M.; Zhong L.X.; Zhan, Z.Y.; Huang Z.H.; Xiong, J.P. Enhanced antitumor efficacy of resveratrol-loaded nanocapsules in colon cancer cells: physicochemical and biological characterization. Rev. Med. Pharmacol. Sci. 2017, 21, 375-382.
  187. Soo, E.; Thakur, S.; Qu, Z.; Jambhrunkar, S.; Parekh, H.S.; Popat, A. Enhancing delivery and cytotoxicity of resveratrol through a dual nanoencapsulation approach. Colloid Interface Sci. 2016, 462, 368-274.
  188. Sudha, T.; El-Far, A.H.; Mousa, D.S.; Mousa, S.A. Resveratrol and its nanoformulation attenuate growth and the angiogenesis of xenograft and orthotopic colon cancer models. Molecules, 2020, 25, 1412.
  189. Wong, K.E.; Ngai, S.C.; Chan, K.G.; Lee, L.H.; Goh, B.H.; Chuah, L.H. Curcumin Nanoformulations for Colorectal Cancer: A Review. Pharmacol. 2019,10, 152. doi: 10.3389/fphar.2019.00152.
  190. Chun, J.; Li, R. J.; Cheng, M. S.; Kim, Y. S. Alantolactone selectively suppresses STAT3 activation and exhibits potent anticancer activity in MDA-MB-231 cells. Cancer Lett. 2015, 357, 393– 403.
  191. Zhang, J.; Shen, L.; Li, X.; Song, W.; Liu, Y.; Huang, L. Nanoformulated codelivery of quercetin and alantolactone promotes an antitumor response through synergistic immunogenic cell death for microsatellite-stable colorectal cancer. ACS NANO 2019, 13, 12511-12524.
  192. Kasala, E. R.; Bodduluru, L. N.; Madana, R. M.; V, Athira K.; Gogoi, R.; Barua, C. C. Chemopreventive and therapeutic potential of chrysin in cancer: mechanistic perspectives. Letters 2015, 233, 214-225.
  193. Lotfi-Attari, J.; Pilehvar-Soltanahmadi, Y.; Dadashpour, M.; Alipour, S.; Farajzadeh, R.; Javidfar, S.; Zarghami, N. Co-delivery of curcumin and chrysin by polymeric nanoparticles inhibit synergistically growth and hTERT gene expression in human colorectal cancer cells. Cancer 2017, 69, 1290-1299.
  194. Yaffe, P.B.; Power Coombs, M.R.; Doucette, C.D.; Walsh, M.; Hoskin, D.W. Piperine, an alkaloid from black pepper, inhibits growth of human colon cancer cells via G1 arrest and apoptosis triggered by endoplasmic reticulum stress. Carcinog. 2015, 54, 1070-1085.
  195. Patial, V.; Mahesh, S.; Sharma, S.; Pratap, K.; Singh, D.; Padwad, Y.S. Synergistic effect of curcumin and piperine in suppression of DENA-​induced hepatocellular carcinoma in rats. Environnemental Toxicol. Pharmacol. 2015, 40, 445-452.
  196. Milincic, D.D.; Popovic, D.A.; Levic, S.M.; Kostic, A.Z.; Tesic, Z.Lj.; Nedovic, V.A.; Pesic, M.B. Application of polyphenol-loaded nanoparticles in food chemistry. Nanomater., 2019, 9,1629.
  197. Decker, E.A.; Phenolics: prooxidants or antioxidants? Rev. 1997, 55, 396-398.
  198. Borowska, S.; Brzoska, M.M.; Tomczyk, M. Complexation of bioelements and toxic metals by polyphenolic compounds – Implications for health. Drug Targets 2018, 19, 1612-1638.
